# Rethinking the Pruning Criteria for Convolutional Neural Network

**Zhongzhan Huang**[1]    **Wenqi Shao**[2,3‡]    **Xinjiang Wang**[3]    **Liang Lin**[1]    **Ping Luo**[4*]

[1]Sun Yat-Sen University, [2]The Chinese University of Hong Kong,
[3]SenseTime Research,[4]The University of Hong Kong

## Abstract

Channel pruning is a popular technique for compressing convolutional neural networks (CNNs), where various pruning criteria have been proposed to remove the redundant filters. From our comprehensive experiments, we found two blind spots of pruning criteria: (1) Similarity: There are some strong similarities among several primary pruning criteria that are widely cited and compared. According to these criteria, the ranks of filters' *Importance Score* are almost identical, resulting in similar pruned structures. (2) Applicability: The filters' *Importance Score* measured by some pruning criteria are too close to distinguish the network redundancy well. In this paper, we analyze the above blind spots on different types of pruning criteria with layer-wise pruning or global pruning. We also break some stereotypes, such as that the results of $\ell_1$ and $\ell_2$ pruning are not always similar. These analyses are based on the empirical experiments and our assumption (*Convolutional Weight Distribution Assumption*) that the well-trained convolutional filters in each layer approximately follow a Gaussian-alike distribution. This assumption has been verified through systematic and extensive statistical tests.

## 1 Introduction

Pruning [1, 2, 3, 4] a trained neural network is commonly seen in network compression. In particular, for CNNs, channel pruning refers to the pruning of the filters in the convolutional layers. There are several critical factors for channel pruning. **Procedures**. One-shot method [5]: Train a network from scratch; Use a certain criterion to calculate filters' *Importance Score*, and prune the filters which have small *Importance Score*; After additional training, the pruned network can recover its accuracy to some extent. Iterative method [1, 6, 7]: Unlike One-shot methods, they prune and fine-tune a network alternately. **Criteria**. The filters' *Importance Score* can be definded by a given criterion. From different ideas, many types of pruning criteria have been proposed, such as Norm-based [5], Activation-based [8, 9], Importance-based [10, 11], BN-based [12] and so on. **Strategy**. Layer-wise pruning: In each layer, we can sort and prune the filters, which have small *Importance Score* measured by a given criterion. Global pruning: Different from layer-wise pruning, global pruning [12, 13] sort the filters from all the layers through their *Importance Score* and prune them.

In this work, we conduct our investigation on a variety of pruning criteria. As one of the simplest and most effective channel pruning criteria, $\ell_1$ pruning [5] is widely used in practice. The core idea of this criterion is to sort the $\ell_1$ norm of filters in one layer and then prune the filters with a small $\ell_1$ norm. Similarly, there is $\ell_2$ pruning which instead leverages the $\ell_2$ norm [7, 6]. $\ell_1$ and $\ell_2$ can be seen as the criteria which use absolute *Importance Score* of filters. Through the study of the distribution of norm, [4] demonstrates that these criteria should satisfy two conditions: (1) the variance of the norm of the filters cannot be too small; (2) the minimum norm of the filters should be small enough. Since

---

*Corresponding author: pluo.lhi@gmail.com;   ‡ co-first author.

35th Conference on Neural Information Processing Systems (NeurIPS 2021).

Table 1: An example to illustrate the phenomenon that different criteria may select the similar sequence of filters for pruning. Taking VGG16 (3$^{\rm rd}$ Conv) and ResNet18 (12$^{\rm th}$ Conv) on Norm-based criteria as examples. The pruned filters' index (the ranks of filters' *Importance Score*) are almost the same, which lead to the similar pruned structures.

| Criteria | Model | Pruned Filters' Index (Top 8) | Model | Pruned Filters' Index (Top 8) |
|---|---|---|---|---|
| $\ell_1$ | ResNet18 | [111, 212, 33, 61, 68, 152, 171, 45] | VGG16 | [102, 28, 9, 88, 66, 109, 86, 45] |
| $\ell_2$ | ResNet18 | [111, 33, 212, 61, 171, 42, 243, 129] | VGG16 | [102, 28, 88, 9, 109, 66, 86, 45] |
| **GM** | ResNet18 | [111, 212, 33, 61, 68, 45, 171, 42] | VGG16 | [102, 28, 9, 88, 109, 66, 45, 86] |
| **Fermat** | ResNet18 | [111, 212, 33, 61, 45, 171, 42, 68] | VGG16 | [102, 28, 88, 9, 109, 66, 45, 86] |

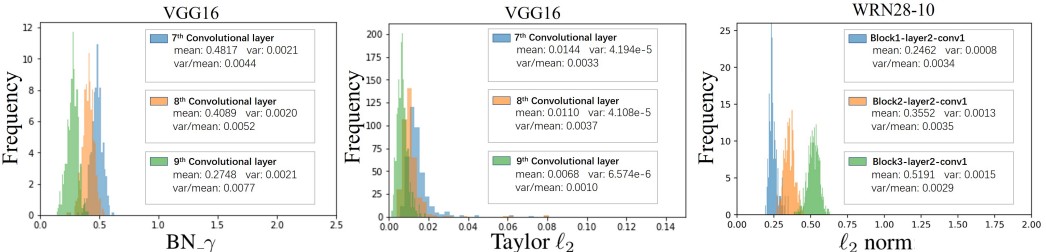

Figure 1: Visualization of Applicability problem, *i.e.,* the histograms of the *Importance Score* measured by different types of pruning criteria (like BN_$\gamma$, Taylor $\ell_2$ and $\ell_2$ norm). The *Importance Score* in each layer are close enough, which implies that it is hard for these criteria to distinguish redundant filters well in layer-wise pruning.

these two conditions do not always hold, a new criterion considering the relative *Importance Score* of the filters is proposed [4]. Since this criterion uses the Fermat point (*i.e.*, geometric median [14]), we call this method **Fermat**. Due to the high calculation cost of Fermat point, [4] further relaxed the **Fermat** and then introduced another criterion denotes as **GM**. To illustrate each of the pruning criteria, let $F_{ij} \in \mathbb{R}^{N_i \times k \times k}$ represent the $j^{\rm th}$ filter of the $i^{\rm th}$ convolutional layer, where $N_i$ is the number of input channels for $i^{\rm th}$ layer and $k$ denotes the kernel size of the convolutional filter. In $i^{\rm th}$ layer, there are $N_{i+1}$ filters. For each criteria, details are shown in Table 2, where $\mathbf{F}$ denotes the Fermat point of $F_{ij}$ in Euclidean space. These four pruning criteria are called Norm-based pruning in this paper as they utilize norm in their design.

Previous works [15, 3, 16, 17, 18], including the criteria mentioned above, the main concerns commonly consist of (a) How much the model was compressed; (b) How much performance was restored; (c) The inference efficiency of the pruned network and (d) The cost of finding the pruned network. However, few works discussed the following two blind spots about the pruning criteria:

**(1) Similarity: What are the actual differences among these pruning criteria?** Taking the VGG16 and ResNet18 on ImageNet as an example, we show the ranks of filters' *Importance Score* under different criteria in Table 1. It is obvious that they have almost the same sequence, leading to similar pruned structures. In this situation, the criteria used absolute *Importance Score* of filters ($\ell_1,\ell_2$) and the criteria used relative *Importance Score* of filters (**Fermat**, **GM**) may not be significantly different.

Table 2: Norm-based pruning criteria.

| Criterion | Details of *Importance Score* |
|---|---|
| $\ell_1$ [5] | $||F_{ij}||_1$ |
| $\ell_2$ [7] | $||F_{ij}||_2$ |
| **Fermat** [4] | $||\mathbf{F} - F_{ij}||_2$ |
| **GM** [4] | $\sum_{k=1}^{N_{i+1}} ||F_{ik} - F_{ij}||_2$ |

**(2) Applicability: What is the applicability of these pruning criteria to prune the CNNs?** There is a toy example w.r.t. $\ell_2$ criterion. If the $\ell_2$ norm of the filters in one layer are 0.9, 0.8, 0.4 and 0.01, according to *smaller-norm-less-informative assumption* [19], it's apparent that we should prune the last filter. However, if the norm are close, such as 0.91, 0.92, 0.93, 0.92, it is hard to determine which filter should be pruned even though the first one is the smallest. In Fig. 1, we demonstrate some real examples, *i.e.,* the visualization of Applicability problem under different networks and criteria.

In this paper, we provide comprehensive observations and in-depth analysis of these two blind spots. Before that, in Section 2, we propose an assumption about the parameters distribution of CNNs, called *Convolution Weight Distribution Assumption* (CWDA), and use it as a theoretical tool to analyze the two blind spots. We explore the Similarity and Applicability problem of pruning criteria in the following order: (1) Norm-based criteria (layer-wise pruning) in Section 3; (2) Other types

of criteria (layer-wise pruning) in Section 4; (3) and different types of criteria (global pruning) in Section 5. Last but not least, we provide further discussion on: (i) the conditions for CWDA to be satisfied, (ii) how our findings help the community in Section 6. In order to focus on the pruning criteria, all the pruning experiments are based on the relatively simple pruning procedure, *i.e.,* one-shot method.

The main **contributions** of this work are two-fold:

**(1)** We analyze the Applicability problem and the Similarity of different types of pruning criteria. These two blind spots can guide and motivate researchers to design more reasonable criteria. We also break some stereotypes, such as that the results of $\ell_1$ and $\ell_2$ pruning are not always similar.

**(2)** We propose and verify an assumption called CWDA, which reveals that the well-trained convolutional filters approximately follow a Gaussian-alike distribution. Using CWDA, we succeeded in explaining the multiple observations about these two blind spots theoretically.

## 2  Weight Distribution Assumption

In this section, we propose and verify an assumption about the parameters distribution of the convolutional filters.

**(Convolution Weight Distribution Assumption)** Let $F_{ij} \in \mathbb{R}^{N_i \times k \times k}$ be the $j^{\text{th}}$ well-trained filter of the $i^{\text{th}}$ convolutional layer. In general[2], in $i^{\text{th}}$ layer, $F_{ij}$ $(j = 1, 2, ..., N_{i+1})$ are i.i.d and follow such a distribution:

$$F_{ij} \sim \mathbf{N}(\mathbf{0}, \mathbf{\Sigma}_{\text{diag}}^i + \epsilon \cdot \mathbf{\Sigma}_{\text{block}}^i), \tag{1}$$

where $\mathbf{\Sigma}_{\text{block}}^i = \text{diag}(K_1, K_2, ..., K_{N_i})$ is a block diagonal matrix and the diagonal elements of $\mathbf{\Sigma}_{\text{block}}^i$ are 0. $\epsilon$ is a small constant. The values of the off-block-diagonal elements are 0 and $K_l \in R^{k^2 \times k^2}, l = 1, 2, ..., N_i$. $\mathbf{\Sigma}_{\text{diag}}^i = \text{diag}(a_1, a_2, ..., a_{N_i \times k \times k})$ is a diagonal matrix and the elements of $\mathbf{\Sigma}_{\text{diag}}^i$ are close enough.

This assumption is based on the observation shown in the Fig. 2. To estimate $\mathbf{\Sigma}_{\text{diag}}^i + \epsilon \cdot \mathbf{\Sigma}_{\text{block}}^i$, we use the correlation matrix $FF^T$ where $F \in \mathbb{R}^{(N_i \times k \times k) \times N_{i+1}}$ denotes all the parameters in $i^{\text{th}}$ layer. Taking a convolutional layer of ResNet18 trained on ImageNet as an example, we find that $FF^T$ is a block diagonal matrix. Specifically, each block is a $k^2 \times k^2$ matrix and the off-diagonal elements are close to 0. We visualize the $j^{\text{th}}$ filter $F_{ij} \in \mathbb{R}^{N_i \times k \times k}$ in $i^{\text{th}}$ layer in Fig. 2(c), and this phenomenon reveals that the parameters in the same channel of $F_{ij}$ tend to be linearly correlated, and the parameters of any two different channels (yellow and green channel in Fig. 2(c)) in $F_{ij}$ only have a low linear correlation.

### 2.1  Statistical test for CWDA

In fact, CWDA is not easy to be verified, *e.g.,* for ResNet164 trained on Cifar100, the number of filters in the first stage is only 16, which is too small to be used to estimate the statistics in CWDA accurately. Thus, We consider verifying four **necessary conditions** of CWDA:

(1) **Gaussian.** Whether the weights of $F_{ij}$ approximately follows a Gaussian-alike distribution; (2) **Variance.** Whether the variance of the diagonal elements of $\Sigma_{\text{diag}}$ are small enough; (3) **Mean.** Whether the mean of weights of $F_{ij}$ is close to 0. (4) **The magnitude of $\epsilon$.** Whether $\epsilon$ is small enough.

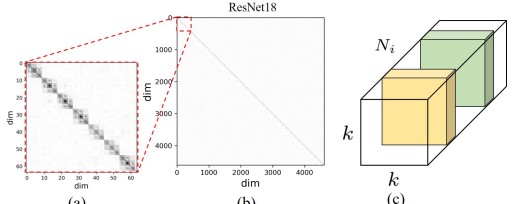

Figure 2: (a-b) Visualization of $FF^T$ in ResNet-18 trained on ImageNet dataset. More experiments can be found in Appendix N. These experiments are based on torchvison model zoo [20], which can guarantee the generality and reproducibility. (c) A convolutional filter. $k$ is the kernel size and $N_i$ denotes the number of input channels.

The results of the tests are shown in Appendix P, where we consider a variety of factors for the statistical tests, including different network structure, optimizer, regularization, initialization, dataset,

---

[2]In Section 6, we make further discussion and analysis on the conditions for CWDA to be satisfied.

training strategy, and other tasks in computer vision (*e.g.*, semantic segmentation, detection and so on). The test results show that CWDA has a great generality for CNNs.

## 3 About the Norm-based criteria

We start from the criteria in Table 2, which are widely cited and compared [21, 22, 23, 24, 25].

### 3.1 Similarity

In this section, we further verify the observation that the Norm-based pruning criteria in Table 2 are highly similar from two perspectives. Empirically, we conducted large amount of experiments on image classification to investigate the similarities. Theoretically, we rigorously prove the similarities of the criteria in Table 2 in layer-wise pruning under CWDA.

**Empirical Analysis**. (1) In Fig. 3, we show the test accuracy of the ResNet56 after pruning and fine-tuning under different pruning ratios and datasets. The test accuracy curves of different pruning criteria at different stages are very close under different pruning ratios. This phenomenon implies that those pruned networks using different Norm-based criteria are very similar, and there are strong similarities among these pruning criteria. The experiments about other commonly used configs of pruning ratio can be found in Appendix L. (2) In Fig. 4, we show the Spearman's rank correlation coefficient[3] (Sp) between different pruning criteria. The Sp in most convolutional layers are more than 0.9, which means the network structures are almost

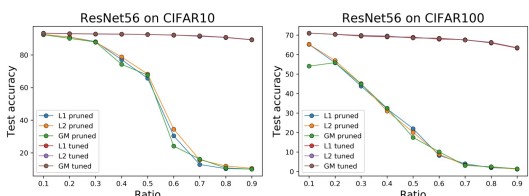

Figure 3: Test accuracy of the ResNet56 on CIFAR10/100 while using different pruning ratios. "L1 pruned" and "L1 tuned" denote the test accuracy of the ResNet56 after $\ell_1$ pruning and fine-tuning, respectively. If ratio is 0.5, we prune 50% filters in all layers.

the same after pruning. Note that the Sp in transition layer are relatively small, and the transition layer refers to the layer where the dimensions of the filter change, like the layer between stage 1 and stage 2 of a ResNet. The reason for this phenomenon may be that the layers in these areas are sensitive. It is interesting but will not greatly impact the structural similarity of the whole pruned network. The similar observations are shown in Fig. 2 in [16], Fig. 6 and Fig. 10 in [5].

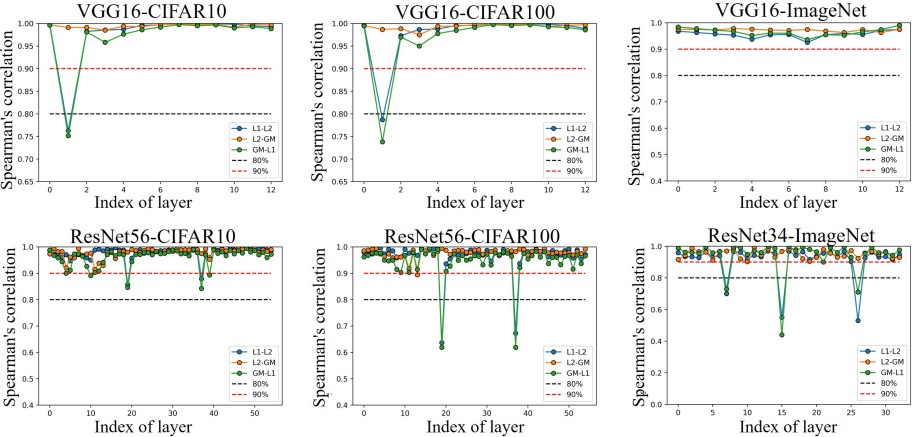

Figure 4: Spearman's rank correlation coefficient (Sp) between different pruning criteria on several networks and datasets (more experiments can be found in Appendix R).

---

[3]Sp is a nonparametric measurement of ranking correlation, and it assesses how well the relationship between two variables can be described using a monotonic function, *i.e.*, filters ranking sequence in the same layer under two criteria in this paper.

**Theoretical Analysis**. Besides the experimental verification, the similarities via using layer-wise pruning among the criteria in Table 2 can also be proved theoretically in this section. Let $C_1$ and $C_2$ be two pruning criteria to calculate the *Importance Score* for all convolutional filters in one layer. If they can produce the similar ranks of *Importance Score*, we define that $C_1$ and $C_2$ are *approximately monotonic* to each other and use $C_1 \cong C_2$ to represent this relationship. In Section 3.1, we use the Sp to describe this relationship but it's hard to be analyzed theoretically. Therefore, we focus on a stronger condition. Let $\mathbf{X} = (x_1, x_2, ..., x_k)$ and $\mathbf{Y} = (y_1, y_2, ..., y_k)$ be two given sequences[4]. we first normalize their magnitude, *i.e.*, let $\widehat{\mathbf{X}} = \mathbf{X}/\mathbb{E}(\mathbf{X})$ and $\widehat{\mathbf{Y}} = \mathbf{Y}/\mathbb{E}(\mathbf{Y})$ . This operation does not change the ranking sequence of the elements of $\mathbf{X}$ and $\mathbf{Y}$, because $\mathbb{E}(\mathbf{X})$ and $\mathbb{E}(\mathbf{Y})$ are constants, *i.e.*, $\widehat{\mathbf{X}} \cong \widehat{\mathbf{Y}} \Leftrightarrow \mathbf{X} \cong \mathbf{Y}$. After that, if both $\mathbf{Var}(\widehat{\mathbf{X}}/\widehat{\mathbf{Y}})$ and $\mathbf{Var}(\widehat{\mathbf{Y}}/\widehat{\mathbf{X}})$ are small enough, then the Sp between $\mathbf{X}$ and $\mathbf{Y}$ is close to 1, where $\widehat{\mathbf{X}}/\widehat{\mathbf{Y}} = (\widehat{x_1}/\widehat{y_1}, .., \widehat{x_k}/\widehat{y_k})$. The reason is that in these situations, the ratio $\widehat{\mathbf{X}}/\widehat{\mathbf{Y}}$ and $\widehat{\mathbf{Y}}/\widehat{\mathbf{X}}$ will be close to two constants $a, b$. For any $1 \leq i \leq k$, $\widehat{x_i} \approx a \cdot \widehat{y_i}$ and $\widehat{y_i} \approx b \cdot \widehat{x_i}$. So, $ab \approx 1$ and $a, b \neq 0$. Therefore, there exists an *approximately monotonic* mapping from $\widehat{y_i}$ to $\widehat{x_i}$ (linear function), which makes the Sp between $\mathbf{X}$ and $\mathbf{Y}$ close to 1. With this basic fact, we propose the Theorem 1, which implies that many Norm-based pruning criteria produces almost the same ranks of *Importance Score*.

**Theorem 1.** *Let $n-$dimension random variable $X$ meet CWDA, and the pair of criteria $(C_1, C_2)$ is one of $(\ell_1, \ell_2)$, $(\ell_2, \mathbf{Fermat})$ or $(\mathbf{Fermat}, \mathbf{GM})$, we have*

$$\max \left\{ \mathbf{Var}_X \left( \frac{\widehat{C}_2(X)}{\widehat{C}_1(X)} \right), \mathbf{Var}_X \left( \frac{\widehat{C}_1(X)}{\widehat{C}_2(X)} \right) \right\} \lesssim B(n), \qquad (2)$$

*where $\widehat{C}_1(X)$ denotes $C_1(X)/\mathbb{E}(C_1(X))$ and $\widehat{C}_2(X)$ denotes $C_2(X)/\mathbb{E}(C_2(X))$. $B(n)$ denotes the upper bound of left-hand side and when $n$ is large enough, $B(n) \to 0$.*

*Proof.* (See Appendix C). □

In specific, for $i^{\text{th}}$ convolutional layer of a CNN, since $F_{ij} \in \mathbb{R}^n$, $j = 1, 2, ...N_{i+1}$, meet CWDA and the dimension $n$ is generally large, we can obtain $\ell_1 \cong \ell_2$, $\ell_2 \cong \mathbf{Fermat}$ and $\mathbf{Fermat} \cong \mathbf{GM}$ according to Theorem 1. Therefore, we have $\ell_1 \cong \ell_2 \cong \mathbf{Fermat} \cong \mathbf{GM}$, which verifies the strong similarities among the criteria shown in Table 2.

## 3.2 Applicability

In this section, we analyze the Applicability problem of the Norm-based criteria. In Fig. 1 (Right), we know that there are some cases where the values of *Importance Score* measured by $\ell_2$ criterion are very close (e.g., the distribution looks sharp), which make $\ell_2$ criterion cannot distinguish the redundant filters well. It's related to the variance of *Importance Score*. [4] argue that a *small norm deviation* (the values of variance of *Importance Score* are small) makes it difficult to find an appropriate threshold to select filters to prune. However, even if the values of the variance are large, it still cannot guarantee to solve this problem. Since the magnitude of these *Importance Score* may be much greater than the values of the variance, we can use the mean of *Importance Score* to represent their magnitude. Therefore, we consider using a relative variance $\mathbf{Var}_r[C(F_A)]$ to describe the Applicability problem. Let $\mathbb{E}[C(F_A)] > 0$ and

$$\mathbf{Var}_r[C(F_A)] = \mathbf{Var}[C(F_A)]/\mathbb{E}[C(F_A)], \qquad (3)$$

where $C$ is a given pruning criterion and $F_A$ denotes the filters in layer $A$. The criterion $C$ for layer $A$ has Applicability problem when $\mathbf{Var}_r[C(F_A)]$ is close to 0. Then we introduce the Proposition 1 to provide the estimation of the mean and variance w.r.t. different criteria when the CWDA is hold:

**Proposition 1.** *If the convolutional filters $F_A$ in layer $A$ meet CWDA, then we have following estimations:*

| Criterion | Mean | Variance |
|---|---|---|
| $\ell_1(F_A)$ | $\sqrt{2/\pi}\sigma_A d_A$ | $(1 - \frac{2}{\pi})\sigma_A^2 d_A$ |
| $\ell_2(F_A)$ | $\sqrt{2}\sigma_A \Gamma(\frac{d_A+1}{2})/\Gamma(\frac{d_A}{2})$ | $\sigma_A^2/2$ |
| $\mathbf{Fermat}(F_A)$ | $\sqrt{2}\sigma_A \Gamma(\frac{d_A+1}{2})/\Gamma(\frac{d_A}{2})$ | $\sigma_A^2/2$ |

---

[4]Since $\mathbf{X}$ is not random variables here, $\mathbb{E}(\mathbf{X})$ and $\mathbf{Var}(\mathbf{X})$ denote the average value $\sum_{i=1}^{k} x_i/k$ and the sample variance $\sum_{i=1}^{k}(x_i - \mathbb{E}(\mathbf{X}))/(k-1)$, respectively.

*where $d_A$ and $\sigma_A^2$ denote the dimension of $F_A$ and the variance of the weights in layer A, respectively.*

*Proof.* (See Appendix A). □

Based on the Proposition 1, we further provide the theoretical analysis for each criteria:

(i) For $\ell_2(F_A)$. From Proposition 1, we can obtain that

$$\mathbf{Var}_r[\ell_2(F_A)] = \frac{\sigma_A^2}{2}/[\sqrt{2}\sigma_A\Gamma(\frac{d_A+1}{2})/\Gamma(\frac{d_A}{2})] = O(\sigma_A/g(d_A)), \qquad (4)$$

where $g(d_A) = \Gamma(\frac{d_A+1}{2})/\Gamma(\frac{d_A}{2})$ is a monotonically increasing function w.r.t $d_A$. From Eq. (4), $\mathbf{Var}_r[\ell_2(F_A)]$ depend on $\sigma_A$ and $d_A$. When $\sigma_A$ is small or $d_A$ is large enough, $\mathbf{Var}_r[\ell_2(F_A)]$ tends to be 0.

(ii) For $\mathbf{Fermat}(F_A)$. From the proof in Appendix D, we know that the Fermat point $\mathbf{F}$ of $F_A$ and the origin $\mathbf{0}$ approximately coincide. From Table 1, $||\mathbf{F} - F_A||_2 \approx ||\mathbf{0} - F_A||_2 = ||F_A||_2$. Therefore, the mean and variance of $\mathbf{Fermat}(F_A)$ are the same as $\ell_2(F_A)$'s in Proposition 1. Hence, a similar conclusion can be obtained for $\mathbf{Fermat}$ criterion. *i.e.,* the *Importance Score* tends to be identical and it's hard to distinguish the network redundancy well when $\sigma_A$ is small or $d_A$ is large enough.

(iii) For $\ell_1(F_A)$. Intuitively, the $\ell_1$ criterion should have the same conclusion as the $\ell_2$ criterion. However, given the Proposition 1, we can obtain that

$$\mathbf{Var}_r[\ell_1(F_A)] = (1 - \frac{2}{\pi})\sigma_A^2 d_A/[\sqrt{2/\pi}\sigma_A d_A] = \epsilon(\pi) \cdot \sigma_A, \qquad (5)$$

where $\epsilon(\pi) < 1$ is a constant w.r.t $\pi$. Note that $\mathbf{Var}_r[\ell_1(F_A)]$ only depend on $\sigma_A$, but not the dimension $n$. Moreover, for the common network structures, like VGG, ResNet shown in Fig. 6 (b) and (d), the dimension of the filters are usually large enough. Therefore, compared with $\ell_2$, $\ell_1$ criterion is relatively not prone to have Applicability problems, unless the $\sigma_A$ is very small.

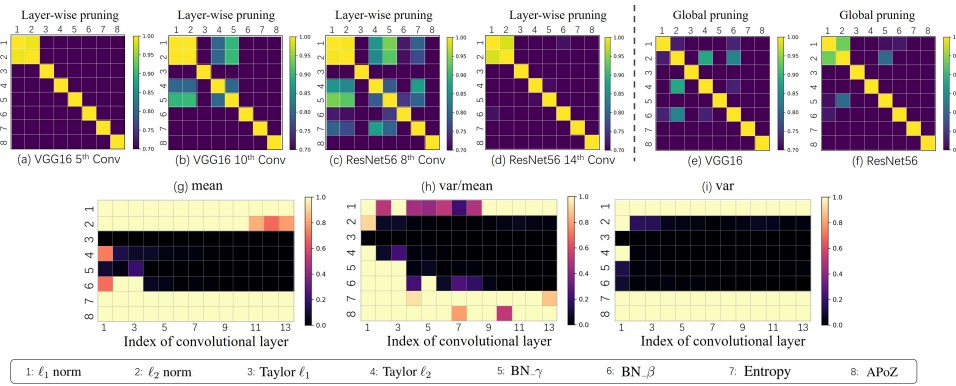

Figure 5: The Similarity and Applicability problem for different types of pruning criteria in layer-wise or global pruning.

## 4 About other types of pruning criteria

In this section, we study the Similarity and Applicability problem in other types of pruning criteria through numerical experiments, such as Activation-based pruning [8, 9], Importance-based pruning [10, 11] and BN-based pruning [12]. For each type, we choose two representative criteria and we call them: (1) Norm-based: $\ell_1$ and $\ell_2$; (2) Importance-based: Taylor $\ell_1$ and Taylor $\ell_2$ [10, 11, 26]; (3) BN-based: BN_$\gamma$[5] and BN_$\beta$ [12]; (4) Activation-based: Entropy [9] and APoZ [8]. The details of these criteria can be found in Appendix K.

**The Similarity for different types of pruning criteria**. In Fig. 5 (a-d), we show the Sp between different types of pruning criteria, and only the Sp greater than 0.7 are shown because if Sp < 0.7, it means that there is no strong similarity between two criteria in the current layer.

---

[5]The empirical result for slimming training [12] is shown in Appendix Q.

According to the Sp shown in Fig. 5 (a-d), we obtain the following observations: (1) As verified in Section 3.1, $\ell_1$ and $\ell_2$ can maintain a strong similarity in each layer; (2) In the layers shown in Fig. 5 (a) and Fig. 5 (d), the Sp between most different pruning criteria are not large in these layers, which indicates that these criteria have great differences in the redundancy measurement of convolutional filters. This may lead to a phenomenon that one criterion considers a convolutional filter to be important, while another considers it redundant. We find a specific example which is shown in Appendix J; (3) Intuitively, the same type of criteria should be similar. However, Fig. 5 (b) and Fig. 5 (c) show that the Sp between Taylor $\ell_1$ and Taylor $\ell_2$ is not large, but Taylor $\ell_2$ has strong similarity with both two Norm-based criteria. Moreover, the Sp between BN_$\gamma$ and each Norm-based criteria exceeds 0.9, but it is not large in other layers (Fig. 5 (a) and Fig. 5 (d)). These phenomena are worthy of further study.

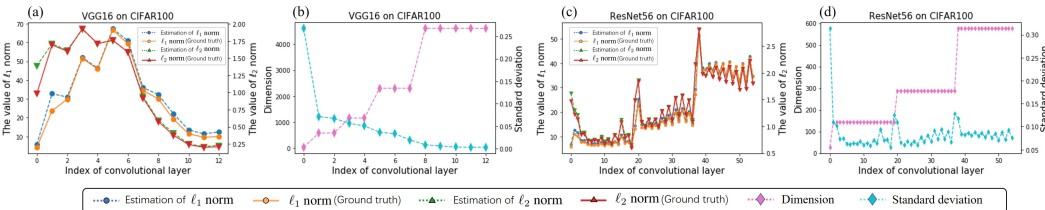

Figure 6: The magnitude of the *Importance Score* measured by $\ell_1$ and $\ell_2$ criteria.

**The Applicability for different types of pruning criteria**. According to the analysis in Section 3.2, the Applicability problem depends on the mean and variance of the *Importance Score*. Fig. 5 (g-i) shows the result of the *Importance Score* measured by different pruning criteria on each layer of VGG16. Due to the difference in the magnitude of *Importance Score* for different criteria, for the convenience of visualization, the value greater than 1 is represented by 1.

First, we analyze the Norm-based criteria. In most layers, the relative variance $\mathbf{Var}_r[\ell_2]$ is much smaller than that of $\mathbf{Var}_r[\ell_1]$, which means that the $\ell_2$ pruning has Applicability problem in VGG16, while the $\ell_1$ does not. This is consistent with our conclusion in Section 3.2. Next, for the Activation-based criteria, the relative variance $\mathbf{Var}_r$ is large in each layer, which means that these two Activation-based criteria can distinguish the network redundancy well from their measured filters' *Importance Score*. However, for the Importance-based and BN-based criteria, their relative variance $\mathbf{Var}_r$ are close to 0. According to Section 3.2, these criteria have Applicability problem, especially in the deeper layers (e.g., from $6^{\text{th}}$ layer to the last layer).

## 5 About global pruning

Compared with layer-wise pruning, global pruning is more widely [27, 10, 12] used in the current research of channel pruning. Therefore, in this section we may also analyze the Similarity and Applicability problem of global pruning.

**Applicability while using global pruning**. In fact, for global pruning, both $\ell_1$ and $\ell_2$ criteria are not prone to Applicability problems. From Proposition 1, we show that the estimations for the mean of *Importance Score* in layer $A$ for $\ell_1$ and $\ell_2$ are $\sigma_A \cdot d_A \sqrt{\frac{2}{\pi}}$ and $\sqrt{2}\sigma_A \cdot \Gamma(\frac{d_A+1}{2})/\Gamma(\frac{d_A}{2})$, respectively. Since $\sigma_A$ and $d_A$ are quite different, shown in Fig. 6 (b) and (d), hence the variance of the *Importance Score* may be large in this situation. Fig. 6 (a) and (c) show such kind of difference of the magnitude

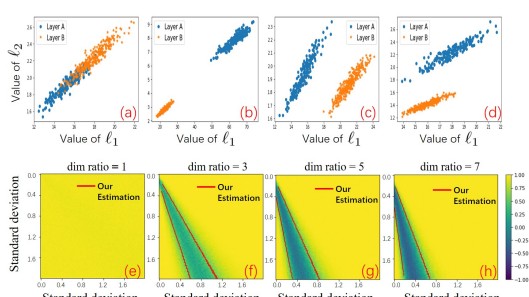

Figure 7: The global pruning simulation for the unpruned network with only two layers.

on different convolutional layers. In addition, from our estimations in Fig. 6 (c), this inconsistent magnitude can be explained for another common problem in practical applications of global pruning: the ResNet is easily pruned off. As shown in Fig. 6 (c), we take ResNet56 as an example. Since the *Importance Score* in first stage is much smaller than the *Importance Score* in the deeper layer,

global pruning will give priority to prune the convolutional filters of the first stage. For problem, we suggest that some normalization tricks should be implemented or a protection mechanism should be established, *e.g.*, a mechanism which can ensure that each layer has at least a certain number of convolutional filters that will not be pruned. Unlike some previous works [13, 28, 29], which make suggestions from qualitative observation, we provide a quantitative view to illustrate that these tricks are necessary.

**Similarity while using global pruning**. In Fig. 5 (e-f), we show the similarity of different types of pruning criteria using global pruning on VGG16 and ResNet56. Comparing to the results from the layer-wise pruning shown in Fig. 5 (a-d), we can find that the similarities of most pruning criteria are quite different in global pruning. In addition, the same criteria may have different results for different network structures in global pruning, *e.g.,* in Fig. 5 (e), we can find $\ell_2 \cong$ Taylor $\ell_2$ and $\mathrm{BN}_\gamma \cong \ell_2$, but this observation does not hold in Fig. 5 (f). In particular, different from the result about ResNet56 in Fig. 5 (f), the similarity between $\ell_1$ and $\ell_2$ is not as strong as the one in the layer-wise case. This phenomenon is counter intuitive.

To understand this phenomenon, we first consider about a simple case, *i.e.,* the unpruned network has only two convolutional layers (layer $A$ and layer $B$). The filters in these two layers are $F_A = (F_A^1, F_A^2, ..., F_A^n)$ and $F_B = (F_B^1, F_B^2, ..., F_B^m)$. According to CWDA, for $1 \le i \le n$ and $1 \le j \le m$, $F_A^i$ and $F_B^j$ can follow $N(\mathbf{0}, \sigma_A^2 \mathbf{I}_{d_A})$ and $N(\mathbf{0}, \sigma_B^2 \mathbf{I}_{d_B})$, respectively. Next, we show Sp between *Importance Score* measured by $\ell_1$ and $\ell_2$ pruning in different dimension ratio $d_A/d_B$, $\sigma_A$ and $\sigma_B$ in Fig. 7 (e-h). Moreover, to analyze this phenomenon concisely, we draw some scatter plots as shown in Fig. 7 (a-d), where the coordinates of each point are given by (value of $\ell_1$, value of $\ell_2$). The set of the points consisting of the filters in layer $A$ is called group-$A$. Then we introduce the Proposition 2.

**Proposition 2.** *If the convolutional filters $F_A$ in layer $A$ meet CWDA, then $\mathbb{E}[\ell_1(F_A)/\ell_2(F_A)]$ and $\mathbb{E}[\ell_2(F_A)/\ell_1(F_A)]$ only depend on their dimension $d_A$.*

*Proof.* (See Appendix A). ☐

Now we analyze the simple case under different situations:

(1) For $d_A/d_B = 1$. If $\sigma_A^2 = \sigma_B^2$, in fact, it's the same situation as layer-wise pruning. From Theorem 1, we know that group-$A$ and group-$B$ coincide and approximately lie on the same line, resulting $\ell_1 \cong \ell_2$ . If $\sigma_A^2 \ne \sigma_B^2$, group-$A$ and group-$B$ lie on two lines, respectively. However, these two lines have the same slope based on Proposition 2, as shown in Fig. 7 (a). For these reasons, we have $\ell_1 \cong \ell_2$ when $d_A/d_B = 1$.

(2) For $d_A/d_B \ne 1$. In Fig. 7 (b-d), there are three main situations about the position relationship between group-$A$ and group-$B$. In Fig. 7 (b), according to Theorem 1, the points in group-$A$ and group-$B$ are monotonic respectively. Moreover, their *Importance Score* measured by $\ell_1$ and $\ell_2$ do not overlap, which make $\ell_1$ and $\ell_2$ are *approximately monotonic* overall. Thus, $\ell_1 \cong \ell_2$. However, for Fig. 7 (c-d), the Sp is small since the points in these two group are not monotonic (the *Importance Score* measured by $\ell_1$ or $\ell_2$ has a large overlap). From Proposition 1 and the approximation $\Gamma(\frac{d_A+1}{2})/\Gamma(\frac{d_A}{2}) \approx \sqrt{d_A/2}$ (Appendix D), these two situations can be described as:

$$\sigma_A d_A \approx \sigma_B d_B \quad or \quad \sigma_A \sqrt{d_A} \approx \sigma_B \sqrt{d_B}, \tag{6}$$

where $d_A \ne d_B$. Through Eq. (6) we can obtain the two red lines shown in Fig. 7 (f-h). It can be seen that the area surrounded by these two red lines is consistent with the area where the Sp is relatively small, which means our analysis is reasonable. Based on the above analysis, we can summarize the conditions about $\ell_1 \cong \ell_2$ in global pruning for two convolutional layers as shown in Table 3.

Next, we go back to the the situation about real neural networks in Fig. 5 (e-f). (1) For ResNet56. As shown in Fig.6 (d), the dimensions of the filters in each stage are almost the same. From Table 3 (1), the pruning results after $\ell_1$ and $\ell_2$ pruning in each stage are similar. And, the magnitudes of the *Importance Score* in each stage are very different, since Table 3 (2), we can obtain that $\ell_1 \cong \ell_2$ for ResNet56.

Table 3: The conditions about $\ell_1 \cong \ell_2$ in global pruning for two layers (layer $A$ and layer $B$)

|  | $d_A = d_B$? | $\frac{\sigma_A}{\sigma_B} \approx \frac{d_B}{d_A}$? | $\frac{\sigma_A}{\sigma_B} \approx \frac{\sqrt{d_B}}{\sqrt{d_A}}$? | $\ell_1 \cong \ell_2$? |
|---|---|---|---|---|
| (1) | ✓ | – | – | ✓ |
| (2) | ✗ | ✗ | ✗ | ✓ |
| (3) | ✗ | ✓ | – | ✗ |
| (4) | ✗ | – | ✓ | ✗ |

(2) For VGG16. As shown in Fig.6 (a-b), compared with ResNet56, VGG16 has some layers with different dimensions but similar *Importance Score* measured by $\ell_1$ or $\ell_2$, such as "layer 2" and "layer

8" for $\ell_2$ criterion in Fig.6 (a). From Table 3 (3-4), these pairs of layers make the Sp small, which explain why the result of $\ell_1$ and $\ell_2$ pruning is not similar in Fig. 5 (e) for VGG16. In Appendix O, more experiments show that we can increase the Sp in global pruning by ignoring part of these pairs of layers, which support our analysis.

## 6 Discussion

### 6.1 Why CWDA sometimes does not hold?

CWDA may not always hold. As shown in Appendix P, a small number of convolutional filters may not pass all statistical tests. In this section, we try to analyze this phenomenon.

(1) **The network is not trained well enough**. The distribution of parameters should be discussed **only when** the network is trained well. If the network does not converge, it is easy to construct a scenario which does not satisfy CWDA, *e.g.*, for a network with uniform initialization, when it is only be trained for a few epochs, the distribution of parameters may be still close to a uniform distribution. At this time, the distribution obviously does not satisfy CWDA. A specific example is in Appendix I.

(2) **The number of filters is insufficient.** In Appendix P, the layers that can not pass the statistical tests are almost those whose position is in the front of the network. A common characteristic of these layers is that they have a few filters, which may not estimate statistics well. Taking the second convolutional layer (64 filters) in VGG16 on CIFAR10 as an example, first, the filters in this layer can not pass all the statistical tests. And then the Sp in this transition layer is relatively small, as shown in Fig. 4. However, in Fig. 8, we change the number of filters in this layer from 64 to 128 or 256. After that, the Sp increases significantly, and the filters can pass all the statistical tests when the number of filters is 256. These observations suggest that the number of filters is a major factor for CWDA to be hold.

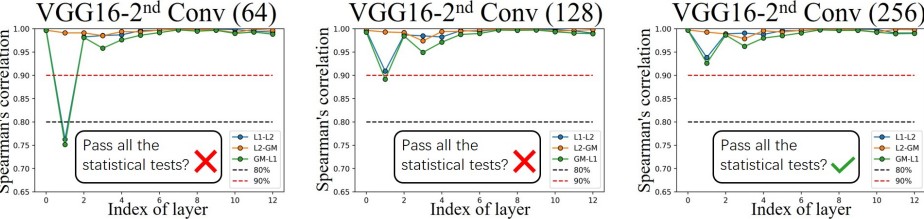

Figure 8: The Sp between different pruning criteria on VGG16 (CIFAR10). The number of filters in the second convolutional layers is changed from 64 to 256. The filters in this layer can pass all the statistical tests when the number of filters is 256.

### 6.2 How our findings help the community?

(1) We propose an assumption about the parameters distribution of the CNNs called CWDA, which is an effective theoretical tool for analyzing convolutional filter. In this paper, CWDA is successfully used to explain many phenomena in the Similarity and Applicability of pruning criteria. In addition, it also explains why the ResNet is easily pruned off in global pruning. In Section 2.1, since CWDA can pass statistical tests in various situations, it can be expected that it can also be used as an effective and concise analysis tool for other CNNs-related areas, **not just** pruning area.

(2) In this paper, we study the Similarity and Applicability problem about pruning criteria, which can guide and motivate the researchers to design more reasonable criteria. For Applicability problem, we suggest that, intuitively, it is reasonable that the *Importance Score* should be distinguishable for the proposed novel criteria. For Similarity, as more and more criteria are proposed, these criteria can be used for ensemble learning to enhance their pruning performance [23]. In this case, the similarity analysis between criteria in this paper is important, because highly similar criteria cannot bring gains to ensemble learning.

(3) In pruning area, $\ell_1$ and $\ell_2$ are usually regarded as the same pruning criteria, which is intuitive. In layer-wise pruning, we do prove that the $\ell_1$ and $\ell_2$ pruning are almost the same. However, in global pruning, the pruning results by these two criteria are sometimes very different. In addition, compared

with $\ell_1$ criterion, $\ell_2$ criterion is prone to Applicability problems. These counter-intuitive phenomena enlighten us that we can't just rely on intuition when analyzing problems.

Table 4: The random pruning results of VGGNet with different criteria which have the Applicability problem. The VGG16 and VGG19 are trained on CIFAR100. The unpruned baseline accuracy of VGG16 and VGG19 are 72.99 and 73.42, respectively.

| Model | criterion | min (r=10%) | max (r=10%) | mean (r=10%) | $\Delta$ | min (r=20%) | max (r=20%) | mean (r=20%) | $\Delta$ |
|---|---|---|---|---|---|---|---|---|---|
| VGG16 | $\ell_2$ | 71.41 | 72.65 | 71.75 | 1.24 | 71.01 | 72.47 | 71.32 | 1.46 |
| | Taylor $\ell_1$ | 71.67 | 72.34 | 71.89 | 0.67 | 71.32 | 72.32 | 71.45 | 1.01 |
| | Taylor $\ell_2$ | 71.87 | 72.37 | 71.91 | 0.5 | 71.66 | 72.27 | 71.65 | 0.61 |
| | $BN_\gamma$ | 71.09 | 71.66 | 71.36 | 0.57 | 71.02 | 71.57 | 71.12 | 0.55 |
| | $BN_\beta$ | 71.15 | 72.58 | 71.43 | 1.43 | 71.06 | 72.11 | 71.87 | 1.05 |
| VGG19 | $\ell_2$ | 71.99 | 73.15 | 72.26 | 1.16 | 71.11 | 73.02 | 72.15 | 1.91 |
| | Taylor $\ell_1$ | 71.67 | 73.04 | 72.23 | 1.37 | 71.6 | 72.98 | 72.24 | 1.38 |
| | Taylor $\ell_2$ | 72.12 | 72.99 | 72.28 | 0.87 | 72.04 | 72.83 | 72.54 | 0.79 |
| | $BN_\gamma$ | 72.01 | 73.23 | 72.25 | 1.22 | 71.98 | 72.32 | 72.12 | 0.34 |
| | $BN_\beta$ | 72.25 | 73.23 | 72.41 | 0.98 | 72.04 | 72.65 | 72.33 | 0.61 |

(4) Similar to the setting in Fig. 5, we can explore the effect of pruning filters with similar *Importance Score* on the performance. First, we find that the criteria ($\ell_2$, Taylor $\ell_1$, Taylor $\ell_2$, $BN_\gamma$ and $BN_\beta$) for VGGNet can cause the Applicability problem in most layers (Fig. 5). As such, we randomly select 10% or 20% filters to be pruned by the uniform distribution $U[0, 1]$ in each layer, and the selective filters will be in similar *Importance Score*. Finally, we finetune the pruned model (there are 20 random repeated experiments). $\Delta$ denotes the difference between max acc. and min acc. (*i.e.* max acc. - min acc.) . Since their *Importance Score* are very similar, when the network is pruned and finetuned, it can be expected that the performance should be similar in these repeated experiments. However, from the results in the above table, although the *Importance Score* of the pruned filters is very close, we can still get pruning results with very different results (*e.g.* the $\Delta$ of VGG16 on $\ell_2$ are more than 1). It means that these criteria may not really represent the importance of convolutional filters. Therefore, it is necessary to re-evaluate the correctness of the existing pruning criteria.

**Acknowledgments**. Z. Huang gratefully acknowledges the technical and writing support from Mingfu Liang (Northwestern University), Senwei Liang (Purdue University) and Wei He (Nanyang Technological University). Moreover, he sincerely thanks Mingfu Liang for offering his self-purchasing GPUs and Qinyi Cai (NetEase, Inc.) for checking part of the proof in this paper. This work was supported in part by the General Research Fund of Hong Kong No.27208720, the National Key R&D Program of China under Grant No. 2020AAA0109700, the National Science Foundation of China under Grant No.61836012 and 61876224, the National High Level Talents Special Support Plan (Ten Thousand Talents Program), and GD-NSF (no.2017A030312006).

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
