# A   Related Proposition

**Proposition 3** (Amoroso distribution). *The Amoroso distribution is a four parameter, continuous, univariate, unimodal probability density, with semi-infinite range [30]. And its probability density function is*

$$\mathbf{Amoroso}(X|a, \theta, \alpha, \beta) = \frac{1}{\Gamma(\alpha)} |\frac{\beta}{\theta}| (\frac{X-a}{\theta})^{\alpha\beta-1} \exp\left\{-(\frac{X-a}{\theta})^{\beta}\right\}, \tag{7}$$

*for $x, a, \theta, \alpha, \beta \in \mathbb{R}, \alpha > 0$ and range $x \geq a$ if $\theta > 0$, $x \leq a$ if $\theta < 0$. The mean and variance of Amoroso distribution are*

$$\mathbb{E}_{X\sim\mathbf{Amoroso}(X|a,\theta,\alpha,\beta)}X = a + \theta \cdot \frac{\Gamma(\alpha + \frac{1}{\beta})}{\Gamma(\alpha)}, \tag{8}$$

*and*

$$\mathbf{Var}_{X\sim\mathbf{Amoroso}(X|a,\theta,\alpha,\beta)}X = \theta^2 \left[\frac{\Gamma(\alpha + \frac{2}{\beta})}{\Gamma(\alpha)} - \frac{\Gamma(\alpha + \frac{1}{\beta})^2}{\Gamma(\alpha)^2}\right]. \tag{9}$$

**Proposition 4** (Half-normal distribution). *Let random variable $X$ follow a normal distribution $N(0, \sigma^2)$, then $Y = |X|$ follows a half-normal distribution [31]. Moreover, $Y$ also follows $\mathbf{Amoroso}(x|0, \sqrt{2}\sigma, \frac{1}{2}, 2)$. By Eq. (8) and Eq. (9), the mean and variance of half-normal distribution are*

$$\mathbb{E}_{X\sim N(0,\sigma^2)}|X| = \sigma\sqrt{2/\pi}, \tag{10}$$

*and*

$$\mathbf{Var}_{X\sim N(0,\sigma^2)}|X| = \sigma^2 \left(1 - \frac{2}{\pi}\right). \tag{11}$$

**Proposition 5** (Scaled Chi distribution). *Let $X = (x_1, x_2, ...x_k)$ and $x_i, i = 1, ..., k$ are $k$ independent, normally distributed random variables with mean 0 and standard deviation $\sigma$. The statistic $\ell_2(X) = \sqrt{\sum_{i=1}^{k} x_i^2}$ follows Scaled Chi distribution [30]. Moreover, $\ell_2(X)$ also follows $\mathbf{Amoroso}(x|0, \sqrt{2}\sigma, \frac{k}{2}, 2)$. By Eq. (8) and Eq. (9), the mean and variance of Scaled Chi distribution are*

$$\mathbb{E}_{X\sim N(\mathbf{0},\sigma^2\cdot\mathbf{I_k})}[\ell_2(X)]^j = 2^{j/2}\sigma^j \cdot \frac{\Gamma(\frac{k+j}{2})}{\Gamma(\frac{k}{2})}, \tag{12}$$

*and*

$$\mathbf{Var}_{X\sim N(\mathbf{0},\sigma^2\cdot\mathbf{I_k})}\ell_2(X) = 2\sigma^2 \left[\frac{\Gamma(\frac{k}{2}+1)}{\Gamma(\frac{k}{2})} - \frac{\Gamma(\frac{k+1}{2})^2}{\Gamma(\frac{k}{2})^2}\right]. \tag{13}$$

**Proposition 6** (Stirling's formula). [6] *For big enough $x$ and $x \in \mathbb{R}^+$, we have an approximation of Gamma function:*

$$\Gamma(x+1) \approx \sqrt{2\pi x}\left(\frac{x}{e}\right)^x. \tag{14}$$

**Proposition 7** (FKG inequality). *If $f$ and $g$ are increasing functions on $\mathbb{R}^n$ [32], we have*

$$\mathbb{E}(f)\mathbb{E}(g) \leq \mathbb{E}(fg). \tag{15}$$

*Say that a function on $\mathbb{R}^n$ is increasing if it is an increasing function in each of its arguments.(i.e., for fixed values of the other arguments).*

---

[6] en.wikipedia.org/wiki/Stirling'sapproximation

**Proposition 8.** *Let $f(X, Y)$ is a two dimensional differentiable function. According to Taylor theorem [33], we have*

$$f(X, Y) = f(\mathbb{E}(X), \mathbb{E}(Y)) + \sum_{cyc}(X - \mathbb{E}(X))\frac{\partial}{\partial X}f(\mathbb{E}(X), \mathbb{E}(Y)) + Remainder1, \quad (16)$$

$$f(X, Y) = f(\mathbb{E}(X), \mathbb{E}(Y)) + \sum_{cyc}(X - \mathbb{E}(X))\frac{\partial}{\partial X}f(\mathbb{E}(X), \mathbb{E}(Y)) +$$
$$\frac{1}{2}\sum_{cyc}(X - \mathbb{E}(X))^T\frac{\partial^2}{\partial X^2}f(\mathbb{E}(X), \mathbb{E}(Y))(X - \mathbb{E}(X)) + Remainder2 \quad (17)$$

**Lemma 1.** *Let $X$ and $Y$ are random variables. Then we have such an estimation*

$$\mathbf{Var}\left(\frac{X}{Y}\right) \approx \left(\frac{\mathbb{E}(X)}{\mathbb{E}(Y)}\right)^2\left(\frac{\mathbf{Var}X}{\mathbb{E}(X)^2} + \frac{\mathbf{Var}Y}{\mathbb{E}(Y)^2} - 2\frac{\mathbf{Cov}(X, Y)}{\mathbb{E}(X)\mathbb{E}(Y)}\right). \quad (18)$$

*Proof.* Let $f(X, Y) = X/Y$, according to the definition of variance, we have

$$\mathbf{Var}f(X, Y) = \mathbb{E}[f(X, Y) - \mathbb{E}(f(X, Y))]^2$$
$$\approx \mathbb{E}[f(X, Y) - \mathbb{E}\left\{f(\mathbb{E}(X), \mathbb{E}(Y)) + \sum_{cyc}(X - \mathbb{E}(X))\frac{\partial}{\partial X}f(\mathbb{E}(X), \mathbb{E}(Y))\right\}]^2$$
$$\hfill \text{from Eq. (16)}$$
$$= \mathbb{E}[f(X, Y) - f(\mathbb{E}(X), \mathbb{E}(Y)) - \sum_{cyc}\mathbb{E}(X - \mathbb{E}(X))\frac{\partial}{\partial X}f(\mathbb{E}(X), \mathbb{E}(Y))]^2$$
$$= \mathbb{E}[f(X, Y) - f(\mathbb{E}(X), \mathbb{E}(Y))]^2$$
$$\approx \mathbb{E}[\sum_{cyc}(X - \mathbb{E}(X))\frac{\partial}{\partial X}f(\mathbb{E}(X), \mathbb{E}(Y))]^2 \hfill \text{from Eq. (16)}$$
$$= 2\mathbf{Cov}(X, Y)\frac{\partial}{\partial X}f(\mathbb{E}(X), \mathbb{E}(Y))\frac{\partial}{\partial Y}f(\mathbb{E}(X), \mathbb{E}(Y)) + \sum_{cyc}[\frac{\partial}{\partial X}f(\mathbb{E}(X), \mathbb{E}(Y))]^2 \cdot \mathbf{Var}X$$
$$= 2\mathbf{Cov}(X, Y) \cdot \frac{1}{\mathbb{E}(Y)} \cdot \left(-\frac{\mathbb{E}(X)}{(\mathbb{E}(Y))^2}\right) + \frac{1}{(\mathbb{E}(Y))^2} \cdot \mathbf{Var}X + \frac{(\mathbb{E}X)^2}{(\mathbb{E}Y)^4} \cdot \mathbf{Var}Y$$
$$= \left(\frac{\mathbb{E}(X)}{\mathbb{E}(Y)}\right)^2\left(\frac{\mathbf{Var}X}{\mathbb{E}(X)^2} + \frac{\mathbf{Var}Y}{\mathbb{E}(Y)^2} - 2\frac{\mathbf{Cov}(X, Y)}{\mathbb{E}(X)\mathbb{E}(Y)}\right).$$

$$\square$$

From Eq.(17) and **Lemma 1**, we also can obtain an estimation of $\mathbb{E}(\mathbf{A}/\mathbf{B})$, where $\mathbf{A}$ and $\mathbf{B}$ are two random variables. *i.e.*,

$$\mathbb{E}\left(\frac{\mathbf{A}}{\mathbf{B}}\right) \approx \frac{\mathbb{E}\mathbf{A}}{\mathbb{E}\mathbf{B}} + \mathbf{Var}(\mathbf{B}) \cdot \frac{\mathbb{E}\mathbf{A}}{(\mathbb{E}\mathbf{B})^3}. \quad (19)$$

**Lemma 2.** *For big enough $x$ and $x \in \mathbb{R}^+$, we have*

$$\lim_{x \to +\infty}\left[\frac{\Gamma(\frac{x+1}{2})}{\Gamma(\frac{x}{2})}\right]^2 \cdot \frac{1}{x} = \frac{1}{2}. \quad (20)$$

*And*

$$\lim_{x \to +\infty}\frac{\Gamma(\frac{x}{2} + 1)}{\Gamma(\frac{x}{2})} - \left[\frac{\Gamma(\frac{x+1}{2})}{\Gamma(\frac{x}{2})}\right]^2 = \frac{1}{4}. \quad (21)$$

*Proof.*

$$\lim_{x \to +\infty} \left[ \frac{\Gamma(\frac{x+1}{2})}{\Gamma(\frac{x}{2})} \right]^2 \cdot \frac{1}{x} \approx \lim_{x \to +\infty} \left( \frac{\sqrt{2\pi(\frac{x-1}{2})} \cdot (\frac{x-1}{2e})^{\frac{x-1}{2}}}{\sqrt{2\pi(\frac{x-2}{2})} \cdot (\frac{x-2}{2e})^{\frac{x-2}{2}}} \right)^2 \cdot \frac{1}{x} \qquad \text{from Proposition. 6}$$

$$= \lim_{x \to +\infty} \left( \frac{x-1}{x-2} \right) \cdot \frac{(\frac{x-1}{2e})^{x-2}}{(\frac{x-2}{2e})^{x-2}} \cdot \left( \frac{x-1}{2e} \right) \cdot \frac{1}{x}$$

$$= \lim_{x \to +\infty} \left( 1 + \frac{1}{x-2} \right)^{x-2} \cdot \frac{x-1}{x-2} \cdot \frac{x-1}{2e} \cdot \frac{1}{x}$$

$$= \frac{1}{2}$$

on the other hand, we have

$$\lim_{x \to +\infty} \frac{\Gamma(\frac{x}{2}+1)}{\Gamma(\frac{x}{2})} - \left[ \frac{\Gamma(\frac{x+1}{2})}{\Gamma(\frac{x}{2})} \right]^2 = \lim_{x \to +\infty} \frac{x}{2} - \left( 1 + \frac{1}{x-2} \right)^{x-2} \cdot \frac{x-1}{x-2} \cdot \frac{x-1}{2e}$$

$$= \lim_{x \to +\infty} \frac{x}{2e} \left( e - (1 + \frac{1}{x})^x \right)$$

$$= \frac{1}{2} \left( -\frac{\frac{1}{e}(-e)}{2} \right)$$

$$= \frac{1}{4}$$

$\square$

**Proposition 9.** *KL divergence between two distributions $P$ and $Q$ of a continuous random variable is given by $D_{KL}(p\|q) = \int_x p(x) \log \frac{p(x)}{q(x)}$. And probabilty density function of multivariate Normal distribution is given by $p(\mathbf{x}) = \frac{1}{(2\pi)^{k/2}|\Sigma|^{1/2}} \exp\left( -\frac{1}{2}(\mathbf{x} - \boldsymbol{\mu})^T \Sigma^{-1}(\mathbf{x} - \boldsymbol{\mu}) \right)$. Let our two Normal distributions be $\mathcal{N}\left( \boldsymbol{\mu_p}, \Sigma_p \right)$ and $\mathcal{N}\left( \boldsymbol{\mu_q}, \Sigma_q \right)$, both $k$ dimensional. we have*

$$D_{KL}(p\|q) = \frac{1}{2} \left[ \log \frac{|\Sigma_q|}{|\Sigma_p|} - k + \left( \boldsymbol{\mu_p} - \boldsymbol{\mu_q} \right)^T \Sigma_q^{-1} \left( \boldsymbol{\mu_p} - \boldsymbol{\mu_q} \right) + \text{tr} \left\{ \Sigma_q^{-1} \Sigma_p \right\} \right]. \tag{22}$$

**Proposition 10** (Jacobi's formula). *If $A$ is a differentiable map from the real numbers to $n \times n$ matrices,*

$$\frac{d}{dt} \det A(t) = \text{tr} \left( \text{adj}(A(t)) \frac{dA(t)}{dt} \right). \tag{23}$$

**Proposition 11.** *For random variable $X$ with $\mu$ and $\sigma^2$ as mean and variance, then we can use Taylor expansion to obtain:*

$$\begin{cases} \mathbb{E}(\log X) \approx \log \mu - \frac{\sigma^2}{2\mu^2} \\ \mathbf{Var}(\log X) \approx \frac{\sigma^2}{\mu^2} \end{cases}. \tag{24}$$

**Proposition 12.** *Given $n$ normal distributions $N(0, \sigma_i^2), 1 \le i \le n$ and $(X_{i1}, X_{i2}, ..., X_{im})$ are sample from $N(0, \sigma_i^2), 1 \le j \le m$. then*

$$\mathbf{Var}_{1 \le i \le n, 1 \le j \le m}(X_{ij}) = \frac{1}{n} \sum_{i=1}^{n} \sigma_i^2. \tag{25}$$

*Proof.*

$$\mathbf{Var}_{1 \le i \le n, 1 \le j \le m}(X_{ij}) = \frac{1}{mn} \sum_{i=1}^{n} \sum_{j=1}^{m} [X_{ij} - \mathbb{E}(X_{ij})]^2 \tag{26}$$

$$= \frac{1}{n} \{ \frac{1}{m} \sum_{j=1}^{m} [X_{ij} - \mathbb{E}(X_{1j})]^2 + ... + \frac{1}{m} \sum_{j=1}^{m} [X_{nj} - \mathbb{E}(X_{nj})]^2 \}$$

$$\text{Since } \mathbb{E}(X_{ij}) = 0, 1 \le i \le n, 1 \le j \le m$$

$$= \frac{1}{n} \{ \sigma_1^2 + ... + \sigma_n^2 \} \tag{27}$$

$\square$

**Lemma 3.** *For a matrix* $\mathbf{B} \in R^{n \times n}$ *and a small constant* $\epsilon$, *we have:*

$$det(\mathbf{I}_n + \epsilon \mathbf{B}) = 1 + \epsilon \operatorname{tr}(\mathbf{B}) + O(\epsilon^2). \tag{28}$$

*Proof.* First, we regard $det(\mathbf{I}_n + \epsilon \mathbf{B})$ as a function w.r.t $\epsilon$. Since Proposition 10, we have:

$$\frac{d}{d\epsilon} det(\mathbf{I}_n + \epsilon \mathbf{B})|_{\epsilon=0} = \operatorname{tr}\{\operatorname{adj}(\mathbf{I}_n + \epsilon \mathbf{B})\mathbf{B}\}|_{\epsilon=0} \tag{29}$$

$$= \operatorname{tr}\{det(\mathbf{I}_n + \epsilon \mathbf{B}) \cdot (\mathbf{I}_n + \epsilon \mathbf{B})^{-1} \mathbf{B}\}|_{\epsilon=0} \tag{30}$$

$$= det(\mathbf{I}_n + \epsilon \mathbf{B}) \cdot \operatorname{tr}\{(\mathbf{I}_n + \epsilon \mathbf{B})^{-1} \mathbf{B}\}|_{\epsilon=0} \tag{31}$$

$$= \operatorname{tr}(\mathbf{B}) \tag{32}$$

Using Taylor expansion for $det(\mathbf{I}_n + \epsilon \mathbf{B})$, we have $\frac{d}{d\epsilon} det(\mathbf{I}_n + \epsilon \mathbf{B}) = det(\mathbf{I}_n) + \frac{d}{d\epsilon} det(\mathbf{I}_n + \epsilon \mathbf{B})|_{\epsilon=0} \cdot \epsilon + O(\epsilon^2)$. In other words, $det(\mathbf{I}_n + \epsilon \mathbf{B}) = 1 + \epsilon \operatorname{tr}(\mathbf{B}) + O(\epsilon^2)$.

$\square$

## A.1 The proof of Proposition 1

(**Proposition** 1) If the convolutional filters $F_A$ in layer $A$ meet CWDA, then we have following estimations:

| Criterion | Mean | Variance |
|---|---|---|
| $\ell_1(F_A)$ | $\sqrt{2/\pi}\sigma_A d_A$ | $(1 - \frac{2}{\pi})\sigma_A^2 d_A$ |
| $\ell_2(F_A)$ | $\sqrt{2}\sigma_A \Gamma(\frac{d_A+1}{2})/\Gamma(\frac{d_A}{2})$ | $\sigma_A^2/2$ |
| **Fermat**$(F_A)$ | $\sqrt{2}\sigma_A \Gamma(\frac{d_A+1}{2})/\Gamma(\frac{d_A}{2})$ | $\sigma_A^2/2$ |

where $d_A$ and $\sigma_A^2$ denote the dimension of $F_A$ and the variance of the weights in layer $A$, respectively.

*Proof.* According to Appendix B, Eq. (21), Proposition 4 and Proposition 5, we can obtain the mean and variance of $\ell_1(F_A)$ and $\ell_2(F_A)$. Moreover, From the Theorem 3, we know that the Fermat point $\mathbf{F}$ of $F_A$ and the origin $\mathbf{0}$ approximately coincide. According to Table 1, $||\mathbf{F} - F_A||_2 \approx ||\mathbf{0} - F_A||_2 = ||F_A||_2$. Therefore, the mean and variance of **Fermat**$(F_A)$ are the same as $\ell_2(F_A)$'s in Proposition 1.

$\square$

## A.2 The proof of Proposition 2

(**Proposition** 2) If the convolutional filters $F_A$ in layer $A$ meet CWDA, then $\mathbb{E}[\ell_1(F_A)/\ell_2(F_A)]$ and $\mathbb{E}[\ell_2(F_A)/\ell_1(F_A)]$ only depend on their dimension $d_A$.

*Proof.* From Eq. (19), we have:

$$
\begin{aligned}
\mathbb{E}[\frac{\ell_1(F_A)}{\ell_2(F_A)}] &\approx \frac{\mathbb{E}[\ell_1(F_A)]}{\mathbb{E}[\ell_2(F_A)]} + \mathbf{Var}[\ell_2(F_A)] \cdot \frac{\mathbb{E}[\ell_1(F_A)]}{\mathbb{E}[\ell_2(F_A)]^3} \\
&= \frac{\sqrt{2/\pi}\sigma_A d_A}{\sqrt{2}\sigma_A \Gamma(\frac{d_A+1}{2})/\Gamma(\frac{d_A}{2})} + \sigma_A^2/2 \cdot \frac{\sqrt{2/\pi}\sigma_A d_A}{[\sqrt{2}\sigma_A \Gamma(\frac{d_A+1}{2})/\Gamma(\frac{d_A}{2})]^3} \quad \text{from Proposition. 1} \\
&\approx O(\sqrt{d_A}) + O(\frac{1}{\sqrt{d_A}}) \qquad\qquad\qquad\qquad\qquad\qquad \text{from Eq. (20)}
\end{aligned}
$$

Similarly, we can prove that $\mathbb{E}[\ell_2(F_A)/\ell_1(F_A)]$ also only depend on their dimension $d_A$.

$$
\begin{aligned}
\mathbb{E}[\frac{\ell_2(F_A)}{\ell_1(F_A)}] &\approx \frac{\mathbb{E}[\ell_2(F_A)]}{\mathbb{E}[\ell_1(F_A)]} + \mathbf{Var}[\ell_1(F_A)] \cdot \frac{\mathbb{E}[\ell_2(F_A)]}{\mathbb{E}[\ell_1(F_A)]^3} \\
&= \frac{\sqrt{2}\sigma_A \Gamma(\frac{d_A+1}{2})/\Gamma(\frac{d_A}{2})}{\sqrt{2/\pi}\sigma_A d_A} + (1 - \frac{2}{\pi})\sigma_A^2 d_A \cdot \frac{\sqrt{2}\sigma_A \Gamma(\frac{d_A+1}{2})/\Gamma(\frac{d_A}{2})}{[\sqrt{2/\pi}\sigma_A d_A]^3} \\
&\qquad\qquad\qquad\qquad\qquad\qquad\qquad\qquad\qquad\qquad \text{from Proposition. 1} \\
&\approx O(\frac{1}{\sqrt{d_A}}) + O(\frac{1}{d_A^{1.5}}) \qquad\qquad\qquad\qquad\qquad\qquad \text{from Eq. (20)}
\end{aligned}
$$

$\square$

# B  The relaxation for CWDA

(**Convolution Weight Distribution Assumption**) Let $F_{ij} \in \mathbb{R}^{N_i \times k \times k}$ be the $j^{\text{th}}$ well-trained filter of the $i^{\text{th}}$ convolutional layer. In general[7], in $i^{\text{th}}$ layer, $F_{ij}$ ($j = 1, 2, ..., N_{i+1}$) are i.i.d and follow such a distribution:

$$F_{ij} \sim \mathbf{N}(\mathbf{0}, \mathbf{\Sigma}_{\text{diag}}^i + \epsilon \cdot \mathbf{\Sigma}_{\text{block}}^i), \tag{33}$$

where $\mathbf{\Sigma}_{\text{block}}^i = \text{diag}(K_1, K_2, ..., K_{N_i})$ is a block diagonal matrix and the diagonal elements of $\mathbf{\Sigma}_{\text{block}}^i$ are 0. $\epsilon$ is a small constant. The values of the off-block-diagonal elements are 0 and $K_l \in R^{k^2 \times k^2}, l = 1, 2, ..., N_i$. $\mathbf{\Sigma}_{\text{diag}}^i = \text{diag}(a_1, a_2, ..., a_{N_i \times k \times k})$ is a diagonal matrix and the elements of $\mathbf{\Sigma}_{\text{diag}}^i$ are close enough.

In Section 2, we propose CWDA. In order to use this assumption conveniently, we give the following relaxation of CWDA:

(**Convolution Weight Distribution Assumption-Relaxation**) Let $F_{ij} \in \mathbb{R}^{N_i \times k \times k}$ be the $j^{\text{th}}$ well-trained filter of the $i^{\text{th}}$ convolutional layer. In general, in $i^{\text{th}}$ layer, $F_{ij}$ ($j = 1, 2, ..., N_{i+1}$) are i.i.d and follow such a distribution:

$$F_{ij} \sim \mathbf{N}(\mathbf{0}, \sigma_{\text{layer}}^2 \cdot \mathbf{I}_{N_i \times k \times k}), \tag{34}$$

where $\sigma_{\text{layer}}^2$ is the variance of the weights in $i^{\text{th}}$ convolutional layer.

Next, we analyze the gap between CWDA and CWDA-Relaxation, *i.e.,* the difference between $\mathbf{N}(\mathbf{0}, \mathbf{\Sigma}_{\text{diag}}^i + \epsilon \cdot \mathbf{\Sigma}_{\text{block}}^i)$ and $\mathbf{N}(\mathbf{0}, \sigma_{\text{layer}}^2 \cdot \mathbf{I}_{N_i \times k \times k})$.

**Lemma 4.** *Given two n-dimension Gaussian distributions* $\mathbf{N}(\mathbf{0}, \mathbf{\Sigma}_{diag} + \epsilon \cdot \mathbf{\Sigma}_{block})$ *and* $\mathbf{N}(\mathbf{0}, \mathbf{\Sigma}_{diag})$, *we can estimate the KL divergence of them:*

$$\text{KL}[\mathbf{N}(\mathbf{0}, \mathbf{\Sigma}_{diag} + \epsilon \cdot \mathbf{\Sigma}_{block}) || \mathbf{N}(\mathbf{0}, \mathbf{\Sigma}_{diag})] \approx \frac{1}{2}\log[\frac{1}{1 + O(\epsilon^2)}] \tag{35}$$

---

[7]In Section 6, we make further discussion and analysis on the conditions for CWDA to be satisfied.

where $\mathbf{\Sigma}_{block} = \mathrm{diag}(K_1, K_2, ..., K_{N_i})$ is a block diagonal matrix and the diagonal elements of $\mathbf{\Sigma}_{block}$ are 0. $\epsilon$ is a small constant. The values of the off-block-diagonal elements are 0 and $K_l \in R^{k^2 \times k^2}, l = 1, 2, ..., N_i$. $\mathbf{\Sigma}_{diag} = \mathrm{diag}(a_1, a_2, ..., a_{N_i \times k \times k})$ is a diagonal matrix and the elements of $\mathbf{\Sigma}_{diag}$ are close enough. $n = N_i \times k \times k$.

*Proof.* Since Proposition 9, we have:

$$2\,\mathrm{KL} = \log \frac{det[\mathbf{\Sigma}_{\mathrm{diag}}]}{det[\mathbf{\Sigma}_{\mathrm{diag}} + \epsilon \cdot \mathbf{\Sigma}_{\mathrm{block}}]} - n + 0 + \mathrm{tr}\{\mathbf{\Sigma}_{\mathrm{diag}}^{-1}(\mathbf{\Sigma}_{\mathrm{diag}} + \epsilon \cdot \mathbf{\Sigma}_{\mathrm{block}})\} \tag{36}$$

$$= \log \frac{det[\mathbf{\Sigma}_{\mathrm{diag}}]}{det[\mathbf{\Sigma}_{\mathrm{diag}} + \epsilon \cdot \mathbf{\Sigma}_{\mathrm{block}}]} - n + \mathrm{tr}\{\mathbf{I}_k + \epsilon \mathbf{\Sigma}_{\mathrm{diag}}^{-1} \mathbf{\Sigma}_{\mathrm{block}}\} \tag{37}$$

$$= \log \frac{det[\mathbf{\Sigma}_{\mathrm{diag}}]}{det[\mathbf{\Sigma}_{\mathrm{diag}} + \epsilon \cdot \mathbf{\Sigma}_{\mathrm{block}}]} \qquad \text{Since the diagonal elements of } \mathbf{\Sigma}_{\mathrm{block}} \text{ are } 0 \tag{38}$$

Let $\mathbf{\Sigma}_{\mathrm{diag}} = \mathrm{diag}(S_1, S_2, ..., S_{N_i})$, where $S_j = \mathrm{diag}(a_{(j-1)k^2+1}, a_{(j-1)k^2+2}, ..., a_{(j-1)k^2+k^2}), j = 1, 2, ..., N_i$.

$$2\,\mathrm{KL} = \log \frac{det[\mathbf{\Sigma}_{\mathrm{diag}}]}{det[\mathbf{\Sigma}_{\mathrm{diag}} + \epsilon \cdot \mathbf{\Sigma}_{\mathrm{block}}]} \tag{39}$$

$$= \log \prod_{j=1}^{n} a_k - \log\{\prod_{h=1}^{N_i} det[S_h + \epsilon K_h]\} \tag{40}$$

$$= \log \prod_{j=1}^{n} a_k - \log\{\prod_{h=1}^{N_i} det[S_h] det[\mathbf{I}_{k^2} + \epsilon S_h^{-1} K_h]\} \qquad \text{Since } S_h \succeq 0 \tag{41}$$

Note that $S_h$ is a diagonal matrix and the diagonal elements of $K_h$ are all zero. Therefore

$$\mathrm{tr}(S_h^{-1} K_h) = 0. \tag{42}$$

Next,

$$2\,\mathrm{KL} = \log \prod_{j=1}^{n} a_k - \log\{\prod_{h=1}^{N_i} det[S_h] det[\mathbf{I}_{k^2} + \epsilon S_h^{-1} K_h]\} \tag{43}$$

$$= \log \prod_{j=1}^{n} a_k - \log\{\prod_{h=1}^{N_i} det[S_h] \cdot (1 + \epsilon\,\mathrm{tr}(S_h^{-1} K_h) + O(\epsilon^2))\} \qquad \text{Since Lemma 3}$$

$$= \log \prod_{j=1}^{n} a_k - \log\{\prod_{h=1}^{N_i} det[S_h] \cdot (1 + O(\epsilon^2))\} \qquad \text{Since Eq. (42)}$$

$$= \log \prod_{j=1}^{n} a_k - \log \prod_{j=1}^{n} a_k(1 + O(\epsilon^2)) \tag{44}$$

$$= \log[\frac{1}{1 + O(\epsilon^2)}] \tag{45}$$

$\square$

According to Statistical test (2) in Section 2.1, $\mathbf{N}(\mathbf{0}, \mathbf{\Sigma}_{\mathrm{diag}})$ can be approximate to $\mathbf{N}(\mathbf{0}, \frac{1}{n}\,\mathrm{tr}(\mathbf{\Sigma}_{\mathrm{diag}})\mathbf{I}_n)$. In addition, from Propsition 12 and Lemma 4, while $\epsilon$ is small enough, the distribution $\mathbf{N}(\mathbf{0}, \mathbf{\Sigma}_{\mathrm{diag}} + \epsilon \cdot \mathbf{\Sigma}_{\mathrm{block}})$ can be approximate to $\mathbf{N}(\mathbf{0}, \sigma_{\mathrm{layer}}^2 \cdot \mathbf{I}_{N_i \times k \times k})$. The analysis in this paper are based on *Convolution Weight Distribution Assumption-Relaxation* and we use it to explain successfully many phenomena in the Similarity and Applicability problem of pruning criteria.

# C  Proof of Theorem 1

**Theorem 1.** Let $n-$dimension random variable $X$ meet CWDA, and the pair of criteria $(C_1, C_2)$ is one of $(\ell_1, \ell_2)$, $(\ell_2, \textbf{Fermat})$ or $(\textbf{Fermat}, \textbf{GM})$, we have

$$\max \left\{ \textbf{Var}_X \left( \frac{\widehat{C}_2(X)}{\widehat{C}_1(X)} \right), \textbf{Var}_X \left( \frac{\widehat{C}_1(X)}{\widehat{C}_2(X)} \right) \right\} \lesssim B(n). \tag{46}$$

where $\widehat{C}_1(X)$ denotes $C_1(X)/\mathbb{E}(C_1(X))$ and $\widehat{C}_2(X)$ denotes $C_2(X)/\mathbb{E}(C_2(X))$. $B(n)$ denotes the upper bound of left-hand side and when $n$ is large enough, $B(n) \to 0$.

For $i^{\text{th}}$ layer, we use $v_j$ to represent $F_{ij}$, $j = 1, 2, ...N$. And $v_j$ meets CWDA. Since Appendix B, we use the following three points to prove Theorem 1.

**(1) For** $(\ell_2, \ell_1)$. In fact, $\ell_2 \cong \ell_1$ (their importance rankings are similar) is not trivial. Generally speaking, for convolutional filters, $\textbf{dim}(v_j)$ is large enough. Since $v_i$ satisfies CWDA, from Theorem 2, we know that the variance of ratio between $\widehat{\ell}_1$ and $\widehat{\ell}_2$ have a bound $O(\textbf{dim}(v_j)^{-1})$, which means $\ell_2$ and $\ell_1$ are *appropriate monotonic*. Specific numerical validation is shown in Fig. 9 of Appendix D).

**Theorem 2.** *Let* $X \sim N(\mathbf{0}, c^2 \cdot \mathbf{I}_n)$, *we have*

$$\max \left\{ \textbf{Var}_X \left( \frac{\widehat{\ell}_2(X)}{\widehat{\ell}_1(X)} \right), \textbf{Var}_X \left( \frac{\widehat{\ell}_1(X)}{\widehat{\ell}_2(X)} \right) \right\} \lesssim \frac{1}{n}. \tag{47}$$

*where* $\widehat{\ell}_1(X)$ *denotes* $\ell_1(X)/\mathbb{E}(\ell_1(X))$ *and* $\widehat{\ell}_2(X)$ *denotes* $\ell_2(X)/\mathbb{E}(\ell_2(X))$. $c$ *is a constant.*

*Proof.* (See Appendix D). $\qquad\qquad\square$

**(2) For** $(\ell_2, \textbf{Fermat})$. Since $v_i$ satisfies CWDA, from Theorem 3, we know that the Fermat point of $v_i$ and the origin $\mathbf{0}$ approximately coincide. According to Table 2, $||\textbf{Fermat} - v_i||_2 \approx ||\mathbf{0} - v_i||_2 = ||v_i||_2$. Therefore, from Theorem 2, the bound $B(n)$ for the $(\ell_1, \textbf{Fermat})$ and $(\ell_2, \textbf{Fermat})$ are $\frac{1}{n}$ and 0, respectively. Moreover, since CWDA, the centroid of $v_i$ is $\mathbf{G} = \frac{1}{n} \sum_{i=1}^{N} v_i = \mathbf{0}$. Hence,

$$\mathbf{G} = \mathbf{0} \approx \textbf{Fermat}. \tag{48}$$

**Theorem 3.** *Let random variable* $v_i \in \mathbb{R}^k$ *and they are i.i.d and follow normal distribution* $N(\mathbf{0}, \sigma^2 \mathbf{I}_k)$. *For* $F \in \mathbb{R}^k$, *we have* $\textbf{argmin}_F \left\{ \mathbb{E}_{v_i \sim N(\mathbf{0}, \sigma^2 \mathbf{I}_k)} \sum_{i=1}^{n} ||F - v_i||_2 \right\} = \mathbf{0}$.

*Proof.* (See Appendix E). $\qquad\qquad\square$

**(3) For** $(\textbf{GM}, \textbf{Fermat})$. First, we show the following two theorems:

**Theorem 4.** *For $n$ random variables* $a_i \in \mathbb{R}^k$ *follow* $N(\mathbf{0}, c^2 \cdot \mathbf{I}_k)$. *When $k$ is large enough, we have such an estimation:*

$$\textbf{Var}_{a_i} \frac{F_1(a_i)}{F_2(a_i)} \approx \frac{1}{2nk}, \quad \textbf{Var}_{a_i} \frac{F_2(a_i)}{F_1(a_i)} \approx \frac{1}{2nk}, \tag{49}$$

*where* $F_1(a_i) = \sum_{i=1}^{n} ||a_i||_2 / \mathbb{E}(\sum_{i=1}^{n} ||a_i||_2)$ *and* $F_2(a_i) = \sum_{i=1}^{n} ||a_i||_2^2 / \mathbb{E}(\sum_{i=1}^{n} ||a_i||_2^2)$.

*Proof.* (See Appendix F). $\qquad\qquad\square$

**Theorem 5.** *Let* $v_0, v_1, ..., v_k$ *be the $k + 1$ vectors in $n$ dimensional Euclidean space* $\mathbb{E}^n$. *For all $P$ in* $\mathbb{E}^n$,

$$\sum_{i=0}^{k} ||P - v_i||_2^2 = \sum_{i=0}^{k} ||G - v_i||_2^2 + (k+1)||P - G||_2^2, \tag{50}$$

*where $G$ is the centroid of $v_i$, will hold if it satisfies one of the following conditions:*

*(1)if $k \geq n$ and* $\textbf{rank}(v_1 - v_0, v_2 - v_0, ..., v_k - v_0) = n$.

*(2)if $k < n$ and* $(v_1 - v_0, v_2 - v_0, ..., v_k - v_0)$ *are linearly independent.*

*(3)if* $v_i \sim N(\mathbf{0}, c^2 \cdot \mathbf{I}_n)$, *Eq.(50) holds with probability 1.*

*Proof.* (See Appendix G). □

Let $P \in \{v_1, v_2, ..., v_N\}$. Since $v_i \sim N(\mathbf{0}, c^2 \cdot \mathbf{I})$, we can obtain that $a_i = P - v_i \sim N(\mathbf{0}, 2c^2 \cdot \mathbf{I})$ if $P \neq v_i$. According to the analysis in Section 3.1 and Theorem 4, we have

$$\sum_{i=1}^{n} ||a_i||_2 \cong \sum_{i=1}^{n} ||a_i||_2^2, \tag{51}$$

Next, we can prove $(k+1)||P - F||_2^2$ (**Fermat**) and $\sum_{i=1}^{N} ||P - v_i||_2$ (**GM**) are *approximately monotonic*, where $P \in \{v_1, v_2, ..., v_N\}$.

$$
\begin{aligned}
(k+1)||P - F||_2^2 &\cong (k+1)||P - G||_2^2 && \text{Since Eq. (48)} \\
&= \sum_{i=1}^{N} ||P - v_i||_2^2 - \sum_{i=1}^{N} ||G - v_i||_2^2 && \text{Since Theorem 5} \\
&\cong \sum_{i=1}^{N} ||P - v_i||_2 - \sum_{i=1}^{N} ||G - v_i||_2^2 && \text{Since Eq. (51)} \\
&\cong \sum_{i=1}^{N} ||P - v_i||_2 && \text{(52)}
\end{aligned}
$$

The reason for the last equation is that $\sum_{i=1}^{N} ||G - v_i||_2^2$ is a constant for given $v_i$.

## D   Proof of Theorem 2

**Theorem 2** Let $X \sim N(\mathbf{0}, c^2 \cdot \mathbf{I}_n)$, we have

$$\mathbf{max}\left\{\mathbf{Var}_X\left(\frac{\widehat{\ell}_2(X)}{\widehat{\ell}_1(X)}\right), \mathbf{Var}_X\left(\frac{\widehat{\ell}_1(X)}{\widehat{\ell}_2(X)}\right)\right\} \lesssim \frac{1}{n}.$$

where $\widehat{\ell}_1(X)$ denotes $\ell_1(X)/\mathbb{E}(\ell_1(X))$ and $\widehat{\ell}_2(X)$ denotes $\ell_2(X)/\mathbb{E}(\ell_2(X))$.

*Proof.* For the ratio $\widehat{\ell}_2(X)/\widehat{\ell}_1(X)$, we have

$$
\begin{aligned}
\mathbf{Var}\left(\frac{\widehat{\ell}_2(X)}{\widehat{\ell}_1(X)}\right) &= \left(\frac{\mathbb{E}(\ell_1(X))}{\mathbb{E}(\ell_2(X))}\right)^2 \mathbf{Var}\left(\frac{\ell_2(X)}{\ell_1(X)}\right) \\
&\approx \left(\frac{\mathbb{E}(\ell_1(X))}{\mathbb{E}(\ell_2(X))}\right)^2 \left(\frac{\mathbb{E}(\ell_2(X))}{\mathbb{E}(\ell_1(X))}\right)^2 \left(\frac{\mathbf{Var}\ell_2(X)}{\mathbb{E}(\ell_2(X))^2} + \frac{\mathbf{Var}\ell_1(X)}{\mathbb{E}(\ell_1(X))^2} - 2\frac{\mathbf{Cov}(\ell_2(X), \ell_1(X))}{\mathbb{E}(\ell_2(X))\mathbb{E}(\ell_1(X))}\right) \\
&&\text{from Lemma. 1} \\
&\leq \left(\frac{\mathbf{Var}\ell_2(X)}{\mathbb{E}(\ell_2(X))^2} + \frac{\mathbf{Var}\ell_1(X)}{\mathbb{E}(\ell_1(X))^2}\right). &&\text{from Proposition. 7}
\end{aligned}
$$

similarly, we also have

$$\mathbf{Var}\left(\frac{\widehat{\ell}_1(X)}{\widehat{\ell}_2(X)}\right) \leq \left(\frac{\mathbf{Var}\ell_2(X)}{\mathbb{E}(\ell_2(X))^2} + \frac{\mathbf{Var}\ell_1(X)}{\mathbb{E}(\ell_1(X))^2}\right). \tag{53}$$

Therefore,

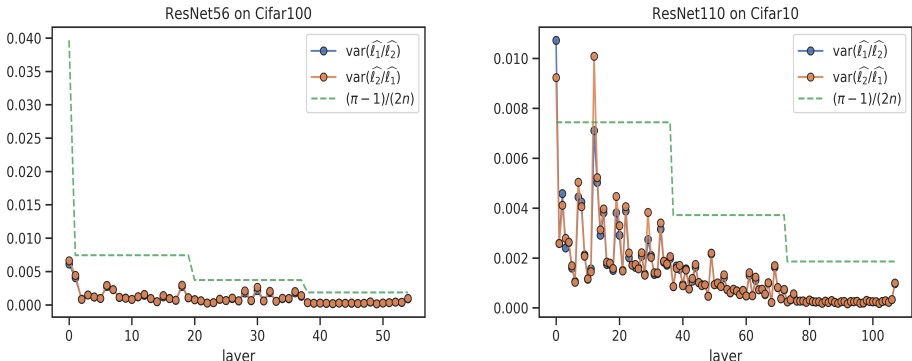

Figure 9: The approximation of **Theorem 2**: (Left) the example about ResNet56; (Right) the example about ResNet110.

$$\mathbf{max}\left\{\mathbf{Var}_X\left(\frac{\widehat{\ell}_2(X)}{\widehat{\ell}_1(X)}\right),\mathbf{Var}_X\left(\frac{\widehat{\ell}_1(X)}{\widehat{\ell}_2(X)}\right)\right\}\leq\left(\frac{\mathbf{Var}\ell_2(X)}{\mathbb{E}(\ell_2(X))^2}+\frac{\mathbf{Var}\ell_1(X)}{\mathbb{E}(\ell_1(X))^2}\right)$$

$$=\frac{2\sigma^2\left[\frac{\Gamma(\frac{n}{2}+1)}{\Gamma(\frac{n}{2})}-\frac{\Gamma(\frac{n+1}{2})^2}{\Gamma(\frac{n}{2})^2}\right]}{(\sqrt{2}\sigma\cdot\frac{\Gamma(\frac{n+1}{2})}{\Gamma(\frac{n}{2})})^2}+\frac{\sigma^2\left(1-\frac{2}{\pi}\right)n}{(n\cdot\sigma\sqrt{2/\pi})^2}$$

$$\text{from Proposition. 5 and 4}$$

$$\approx\left(\frac{1}{2n}+(\frac{\pi}{2}-1)\frac{1}{n}\right)\qquad\text{from Lemma 2}$$

$$=\frac{\pi-1}{2n}$$

$$\square$$

Because the approximation is widely used in the proof of Theorem 1, it is necessary to verify it numerically. As shown in Fig. 9, we use ResNet56 on Cifar100 and ResNet110 on Cifar10 respectively to verify Theorem 1. From Fig. 9, we find that the estimationn of Theorem 1 is reliable, *i.e.*, the estimation $O(\frac{1}{n})$ for $\mathbf{max}\left\{\mathbf{Var}_X\left(\frac{\widehat{\ell}_2(X)}{\widehat{\ell}_1(X)}\right),\mathbf{Var}_X\left(\frac{\widehat{\ell}_1(X)}{\widehat{\ell}_2(X)}\right)\right\}$ is appropriate.

## E    Proof of Theorem 3

**Proposition 13.** *Let $L_p^{(\alpha)}(x)$ denotes generalized Laguerre function, and it have following properties:*

$$\frac{\partial^n}{\partial x^n}L_p^{(\alpha)}=(-1)^nL_{p-n}^{(\alpha+n)}(x),\tag{54}$$

*and for $\alpha>0$,*

$$L_{-\frac{1}{2}}^{(\alpha)}(x)>0.\tag{55}$$

**Theorem 3.** Let random variable $v_i\in\mathbb{R}^k$. They are i.i.d and follow normal distribution $N(\mathbf{0},\sigma^2\mathbf{I}_k)$. For $F$ in $\mathbb{R}^k$, we have

$$\mathbf{argmin}_F\left\{\mathbb{E}_{v_i\sim N(\mathbf{0},\sigma^2\mathbf{I}_k)}\sum_{i=1}^n||F-v_i||_2\right\}=\mathbf{0}.$$

*Proof.* Let $w_i = F - v_i$ and we have $w_i \sim N(F, \sigma^2 \mathbf{I}_k)$, then

$$\mathbb{E}_{v_i \sim N(\mathbf{0}, \sigma^2 \mathbf{I}_k)} \sum_{i=1}^{n} ||F - v_i||_2 = \sum_{i=1}^{n} \mathbb{E}_{v_i \sim N(\mathbf{0}, \sigma^2 \mathbf{I}_k)} ||F - v_i||_2$$

$$= \sum_{i=1}^{n} \mathbb{E}_{w_i \sim N(F, \sigma^2 \mathbf{I}_k)} ||w_i||_2$$

$$= n \cdot \sigma^2 \sqrt{\frac{\pi}{2}} \cdot L_{\frac{1}{2}}^{(\frac{k}{2}-1)} \left( -\frac{||F||_2^2}{2\sigma^2} \right)$$

The reason for the last equation is that $||w_i||_2$ follows scaled noncentral chi distribution[8] when $w_i \sim N(F, \sigma^2 \mathbf{I}_k)$. Let $T(x) = L_{\frac{1}{2}}^{(\frac{k}{2}-1)} \left( -\frac{x^2}{2\sigma^2} \right)$, we calculate the minimum of $T(x)$. From Eq. (54),

$$\frac{d}{dx} T(x) = \frac{x}{\sigma^2} \cdot L_{-\frac{1}{2}}^{(\frac{k}{2})} \left( -\frac{x^2}{2\sigma^2} \right). \tag{56}$$

Since Eq. (55), we find that $\frac{d}{dx} T(x) > 0$ when $x > 0$ and if $x \leq 0$, then $\frac{d}{dx} T(x) \leq 0$. It means that $T(x)$ gets the minimizer at $||F||_2 = 0$, *i.e.*, $F = \mathbf{0}$.

$\square$

# F  Proof of Theorem 4

**Lemma 5.** *For two random variables $X, Y \in \mathbb{R}^k$ follow $N(\mathbf{0}, c^2 \cdot \mathbf{I}_k)$ and they are i.i.d. When $k$ is large enough, we have:*

$$\mathbb{E} \left( \frac{(||X||_2^2 - ||Y||_2^2)^2}{2||X||_2 \cdot ||Y||_2} \right) \approx 2c^2 + \frac{4c^2 k + 1}{2k^2}, \tag{57}$$

*and*

$$\mathbf{Var} \left( \frac{(||X||_2^2 - ||Y||_2^2)^2}{2||X||_2 \cdot ||Y||_2} \right) \lesssim 8c^4 + \frac{16c^4 k + c^2}{k^2}, \tag{58}$$

*Proof.* According to **Proposition 3** and **Lemma 2**, it is easy to know, when $k$ is large enough, that

$$\mathbb{E} \left( 2||X||_2 \cdot ||Y||_2 \right) = 2c^2 k, \quad \mathbf{Var} \left( 2||X||_2 \cdot ||Y||_2 \right) = c^2 + 4c^4 k, \tag{59}$$

and

$$\mathbb{E} \left( (||X||_2^2 - ||Y||_2^2)^2 \right) = 4c^4 k, \quad \mathbf{Var} \left( (||X||_2^2 - ||Y||_2^2)^2 \right) = 16c^8 (2k^2 + 3k). \tag{60}$$

Since Lemma 1, we have an estimation

$$\mathbf{Var} \left( \frac{(||X||_2^2 - ||Y||_2^2)^2}{2||X||_2 \cdot ||Y||_2} \right) \leq \left( \frac{\mathbb{E}(||X||_2^2 - ||Y||_2^2)^2}{\mathbb{E} 2||X||_2 \cdot ||Y||_2} \right)^2 \left( \frac{\mathbf{Var}(||X||_2^2 - ||Y||_2^2)^2}{\mathbb{E}(||X||_2^2 - ||Y||_2^2)^2} + \frac{\mathbf{Var}(2||X||_2 \cdot ||Y||_2)^2)}{\mathbb{E}(2||X||_2 \cdot ||Y||_2)^2} \right)$$

$$\approx \left( \frac{4c^4 k}{2c^2 k} \right)^2 \cdot \left( \frac{c^2 + 4c^4 k}{4c^4 k} + \frac{16c^8 (2k^2 + 3k)}{16c^8 k^2} \right) \qquad \text{Since Eq.(59) and Eq.(60)}$$

$$= 8c^4 + \frac{16c^4 k + c^2}{k^2}.$$

Therefore,

$$\mathbb{E} \left( \frac{(||X||_2^2 - ||Y||_2^2)^2}{2||X||_2 \cdot ||Y||_2} \right) \approx \frac{\mathbb{E}(||X||_2^2 - ||Y||_2^2)^2}{\mathbb{E} 2||X||_2 \cdot ||Y||_2} + \mathbf{Var}(2||X||_2 \cdot ||Y||_2) \cdot \frac{\mathbb{E}(||X||_2^2 - ||Y||_2^2)^2}{(\mathbb{E} 2||X||_2 \cdot ||Y||_2)^3}$$

$$\text{Since Eq.(19)}$$

$$\approx \frac{4c^4 k}{2c^2 k} + \frac{4c^4 k}{8c^6 k^3} \cdot (c^2 + 4c^4 k) \qquad \text{Since Eq.(59) and Eq.(60)}$$

$$= 2c^2 + \frac{4c^2 k + 1}{2k^2}.$$

$\square$

---

[8]Survey of simple,continuous,uniariate probability distribution and Wikipredia.

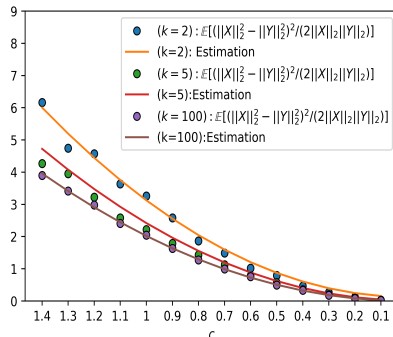 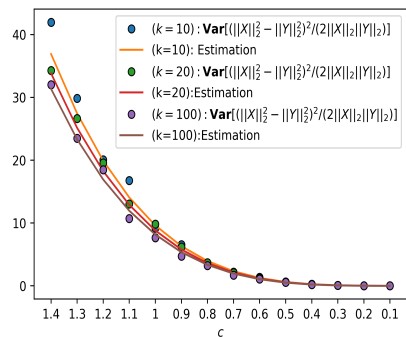

Figure 10: (Left) The numerical verification of Eq.(57) and (Right) The numerical verification of Eq.(58). $X$ and $Y$ follow $N(\mathbf{0}, c^2 \cdot I_k)$.

Note that, the approximation is widely used in the proof of Eq.(57) and Eq.(58). Hence, it is also necessary to verify it numerically. As shown in Fig. 10, the estimation is appropriate. According to **Lemma** 5, the mathematical expectation and variance of the ratio of $(||X||_2^2 - ||Y||_2^2)^2$ and $2||X||_2 \cdot ||Y||_2$ are both close to 0 when $k$ is large enough and $c$ is small enough. that is,

$$2(||X||_2 \cdot ||Y||_2) \gg (||X||_2^2 - ||Y||_2^2)^2. \tag{61}$$

By the way, the convolutional filters easily meet the condition that $k$ is large enough.

**Theorem 4.** For $n$ random variables $a_i \in \mathbb{R}^k$ follow $N(\mathbf{0}, c^2 \cdot \mathbf{I}_k)$.When $k$ is large enough, we have such an estimation:

$$\mathbf{Var}_{a_i} \frac{F_1(a_i)}{F_2(a_i)} \approx \frac{1}{2nk}, \quad \mathbf{Var}_{a_i} \frac{F_2(a_i)}{F_1(a_i)} \approx \frac{1}{2nk}.$$

where $F_1(a_i) = \sum_{i=1}^{n} ||a_i||_2 / \mathbb{E}(\sum_{i=1}^{n} ||a_i||_2)$ and $F_2(a_i) = \sum_{i=1}^{n} ||a_i||_2^2 / \mathbb{E}(\sum_{i=1}^{n} ||a_i||_2^2)$.

*Proof.* Since Eq. (12) and Eq. (13), we have

$$\mathbf{Var}_{a_i} \frac{F_1(a_i)}{F_2(a_i)} = \left(\frac{nc^2 k}{nc\sqrt{k}}\right)^2 \cdot \mathbf{Var}_{a_i} \left(\frac{\sum_{i=1}^{n} ||a_i||_2}{\sum_{i=1}^{n} ||a_i||_2^2}\right). \tag{62}$$

and

$$\mathbf{Var}_{a_i} \frac{F_2(a_i)}{F_1(a_i)} = \left(\frac{nc\sqrt{k}}{nc^2 k}\right)^2 \cdot \mathbf{Var}_{a_i} \left(\frac{\sum_{i=1}^{n} ||a_i||_2^2}{\sum_{i=1}^{n} ||a_i||_2}\right). \tag{63}$$

According to Lagrange's identity, we have

$$\left(\sum_{i=1}^{n} ||a_i||_2^2\right) \left(\sum_{i=1}^{n} 1\right) = \left(\sum_{i=1}^{n} ||a_i||_2\right)^2 + \sum_{1 \le i < j \le n} (||a_i||_2^2 - ||a_j||_2^2)^2$$

$$= \sum_{i=1}^{n} ||a_i||_2^2 + \sum_{1 \le i < j \le n} (||a_i||_2 \cdot ||a_j||_2) + 2 \sum_{1 \le i < j \le n} (||a_i||_2^2 - ||a_j||_2^2)^2$$

$$\approx \sum_{i=1}^{n} ||a_i||_2^2 + 2 \sum_{1 \le i < j \le n} (||a_i||_2 \cdot ||a_j||_2) \qquad \text{Since Eq. (61)}$$

$$= \left(\sum_{i=1}^{n} ||a_i||_2\right)^2$$

so we have

$$\mathbf{Var}_{a_i \sim N(\mathbf{0}, c^2 \cdot \mathbf{I}_k)} \frac{\sum_{i=1}^{n} ||a_i||_2}{\sum_{i=1}^{n} ||a_i||_2^2} \approx \mathbf{Var}_{a_i \sim N(\mathbf{0}, c^2 \cdot \mathbf{I}_k)} \frac{n}{\sum_{i=1}^{n} ||a_i||_2} \tag{64}$$

By central limit theorem, we have $\sqrt{n}(\frac{1}{n}\sum_{i=1}^{n}||a_i||_2 - \mu) \sim N(\mathbf{0}, \sigma^2)$. And let $g(x) = \frac{1}{x}$, we can use Delta method[9] to find the distribution of $g(\frac{1}{n}\sum_{i=1}^{n}||a_i||_2)$:

$$\sqrt{n}\left(g(\frac{\sum_{i=1}^{n}||a_i||_2}{n}) - g(\mu))\right) \sim N(0, \sigma^2 \cdot [g\prime(\mu)]^2) = N(0, \sigma^2 \cdot \frac{1}{\mu^4}). \tag{65}$$

where $\mu$ and $\sigma^2$ denote the mean and variance of $||a_i||_2$ respectively. From Eq. (64), we have

$$\mathbf{Var}_{a_i \sim N(\mathbf{0}, c^2 \cdot \mathbf{I}_k)} \frac{\sum_{i=1}^{n}||a_i||_2}{\sum_{i=1}^{n}||a_i||_2^2} \approx \mathbf{Var}_{a_i \sim N(\mathbf{0}, c^2 \cdot \mathbf{I}_k)} \frac{n}{\sum_{i=1}^{n}||a_i||_2}$$

$$= \sigma^2 \cdot \frac{1}{\mu^4 \cdot n} \qquad\qquad \text{Since Eq. (65)}$$

$$= 2c^2 \left[ \frac{\Gamma(\frac{k}{2}+1)}{\Gamma(\frac{k}{2})} - \frac{\Gamma(\frac{k+1}{2})^2}{\Gamma(\frac{k}{2})^2} \right] \cdot \frac{1}{(\sqrt{2}c \cdot \frac{\Gamma(\frac{k+1}{2})}{\Gamma(\frac{k}{2})})^4 \cdot n}$$
$$\text{Since Eq. (12) and Eq. (13)}$$

$$= \frac{1}{2c^2 \cdot nk^2} \qquad\qquad \text{Since Lemma. 2}$$

Since Eq. (62), we have

$$\mathbf{Var}_{a_i} \frac{F_1(a_i)}{F_2(a_i)} = \left( \frac{nc^2 k}{nc\sqrt{k}} \right)^2 \cdot \mathbf{Var}_{a_i} \left( \frac{\sum_{i=1}^{n}||a_i||_2}{\sum_{i=1}^{n}||a_i||_2^2} \right) \approx \frac{1}{2nk}. \tag{66}$$

Similar to Eq. (64),

$$\mathbf{Var}_{a_i \sim N(\mathbf{0}, c^2 \cdot \mathbf{I}_k)} \frac{\sum_{i=1}^{n}||a_i||_2^2}{\sum_{i=1}^{n}||a_i||_2} \approx \mathbf{Var}_{a_i \sim N(\mathbf{0}, c^2 \cdot \mathbf{I}_k)} \frac{\sum_{i=1}^{n}||a_i||_2}{n} \tag{67}$$

$$\mathbf{Var}_{a_i \sim N(\mathbf{0}, c^2 \cdot \mathbf{I}_k)} \frac{\sum_{i=1}^{n}||a_i||_2^2}{\sum_{i=1}^{n}||a_i||_2} \approx \mathbf{Var}_{a_i \sim N(\mathbf{0}, c^2 \cdot \mathbf{I}_k)} \frac{\sum_{i=1}^{n}||a_i||_2}{n} \qquad \text{Similar to Eq. (64)}$$

$$= \sigma^2 \cdot \frac{1}{n} \qquad\qquad \text{Since central limit theorem}$$

$$= 2c^2 \left[ \frac{\Gamma(\frac{k}{2}+1)}{\Gamma(\frac{k}{2})} - \frac{\Gamma(\frac{k+1}{2})^2}{\Gamma(\frac{k}{2})^2} \right] \cdot \frac{1}{n} \qquad \text{Since Eq. (13)}$$

$$= \frac{c^2}{2n} \qquad\qquad \text{Since Lemma. 2}$$

Since Eq. (63), we have

$$\mathbf{Var}_{a_i} \frac{F_2(a_i)}{F_1(a_i)} = \left( \frac{nc\sqrt{k}}{nc^2 k} \right)^2 \cdot \mathbf{Var}_{a_i} \left( \frac{\sum_{i=1}^{n}||a_i||_2^2}{\sum_{i=1}^{n}||a_i||_2} \right) \approx \frac{1}{2nk}. \tag{68}$$

From Eq.(66) and Eq.(68), **Theorem 4** holds.

$\square$

In Fig. 11, we also show a numerical verification of **Theorem 4**.

---

[9]https://en.wikipedia.org/wiki/Delta_method

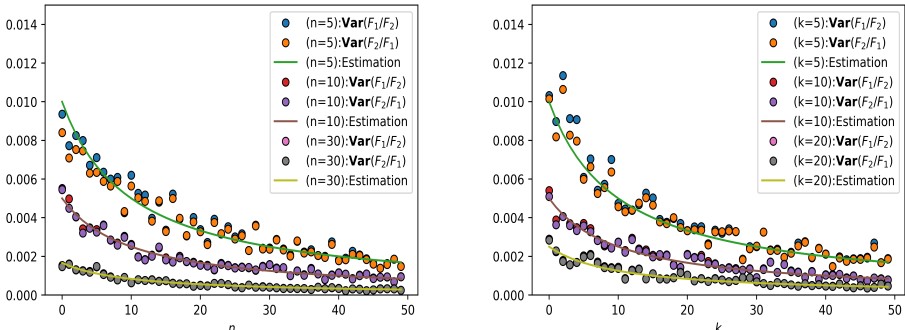

Figure 11: A numerical verification of **Theorem 4**, where $F_1 = \sum_{i=1}^{n} ||a_i||_2 / \mathbb{E}(\sum_{i=1}^{n} ||a_i||_2)$ and $F_2 = \sum_{i=1}^{n} ||a_i||_2^2 / \mathbb{E}(\sum_{i=1}^{n} ||a_i||_2^2)$. $a_i$ follow $N(\mathbf{0}, 0.01^2 \cdot I_k)$.

## G Proof of Theorem 5

**Proposition 14.** *For a $n \times m$ random matrix $(a_{ij})_{n \times m}$, where $a_{ij} \sim N(0, \sigma^2)$. And Eq. (14) holds with probability 1.*

$$\mathbf{rank}((a_{ij})_{n \times m}) = \mathbf{min}(m, n). \tag{69}$$

**Lemma 6.** *Let $v_0, v_1, ..., v_k$ be the $k+1$ vectors in $n$ dimensional Euclidean space $V$ and $k \leq n$. If $\mathbf{rank}(v_1 - v_0, v_2 - v_0, ..., v_k - v_0) = n$, then $\forall x \in V$, $\exists \lambda_i (0 \leq i \leq k)$, s.t.*

$$x = \sum_{i=0}^{k} \lambda_i \cdot v_i, \tag{70}$$

*and $\sum_{i=0}^{k} \lambda_i = 1$. We call $\lambda = (\lambda_0, \lambda_1, ..., \lambda_k)$ the generalized barycentric coordinate with respect to $(v_0, v_1, ..., v_k)$. (In general, barycentric coordinate is a concept in Polytope)*

*Proof.* Note that $v_i$ is the element of $n$ dimensional linear space $V$ and $\mathbf{rank}(v_1 - v_0, v_2 - v_0, ..., v_k - v_0) = n$. It means $(v_1 - v_0, v_2 - v_0, ..., v_k - v_0)$ form a set of basis in the linear space $V$. $\forall x \in V$, $x - v_0$ can be expressed linearly by them, *i.e.*, $\exists t_i (1 \leq i \leq k)$ s.t.

$$x = v_0 + \sum_{i=1}^{k} t_i (v_i - v_0)$$

$$= (1 - \sum_{i=1}^{k} t_i) v_0 + \sum_{i=1}^{k} t_i v_i.$$

Let $\lambda_0 = (1 - \sum_{i=1}^{k} t_i)$ and $\lambda_i = t_i (1 \leq i \leq k)$, Lemma 6 holds. $\square$

**Lemma 7.** *Let $v_0, v_1, ..., v_k$ be the $k+1$ vectors in $n$ dimensional Euclidean space $V$. $\forall a, b \in V$, and the generalized barycentric coordinate of $a, b$ with respect to $(v_0, v_1, ..., v_k)$ are $\lambda = (\lambda_0, \lambda_1, ..., \lambda_k)^T$ and $\mu = (\mu_0, \mu_1, ..., \mu_k)^T$, respectively. Then*

$$||a - b||_2^2 = (\lambda - \mu)^T D (\lambda - \mu), \tag{71}$$

*where $D = (-\frac{1}{2} d_{ij})_{(k+1) \times (k+1)}$, and $d_{ij} = ||v_i - v_j||_2^2$.*

*Proof.* Since Lemma 6, let $R = [v_0, v_1, ..., v_k]_{n \times (k+1)}$, and we have $a = R\lambda$ and $b = R\mu$. Moreover,

$$||a - b||_2^2 = (a - b)^T (a - b) \tag{72}$$

$$= [R(\lambda - \mu)]^T [R(\lambda - \mu)] \tag{73}$$

$$= (\lambda - \mu)^T R^T R (\lambda - \mu). \tag{74}$$

Note that, for $D = (-\frac{1}{2}d_{ij})_{(k+1)\times(k+1)}$,

$$-\frac{1}{2}d_{ij} = -\frac{1}{2}(v_i - v_j)^T(v_i - v_j) \tag{75}$$

$$= v_i^T v_j - \frac{1}{2}(v_i^T v_i + v_j^T v_j). \tag{76}$$

So we have $D = R^T R - \frac{1}{2}\left((v_i^T v_i + v_j^T v_j)_{(k+1)\times(k+1)}\right)$. It can be further simplified to $D = R^T R - \frac{1}{2}(V\alpha^T + \alpha V^T)$, where $V = (v_0^T v_0, ..., v_k^T v_k)^T$ and $\alpha = (1, ..., 1)^T$. So

$$||a - b||_2^2 = (\lambda - \mu)^T R^T R(\lambda - \mu) \tag{77}$$

$$= (\lambda - \mu)^T(D + \frac{1}{2}(V\alpha^T + \alpha V^T))(\lambda - \mu) \tag{78}$$

$$= (\lambda - \mu)^T D(\lambda - \mu) + \frac{1}{2}(\lambda - \mu)^T(V\alpha^T + \alpha V^T)(\lambda - \mu), \tag{79}$$

therefore, we only need to prove $(\lambda - \mu)^T(V\alpha^T + \alpha V^T)(\lambda - \mu) = 0$. From Lemma 6, we have $\alpha^T(\lambda - \mu) = (\lambda - \mu)^T\alpha = 0$ and the Lemma 7 holds.

$\square$

**Definition 1** (Ultra dimension). *For a set $U$ composed of vectors in a $n$ dimensional linear space $V$, we define $\widehat{\dim}(U)$ as the Ultra dimension of $U$. The definition is that if $U$ has $k$ linearly independent vectors and there are no more, then $\widehat{\dim}(U) = k$.*

In fact, if $U$ is a linear subspace in $V$, then the Ultra dimension and the dimensions of the linear subspace are equivalent. If $U$ is a linear manifold, $U = \{x + v_0 | x \in W\}$, where $v_0$ and $W$ are non-zero vectors and linear subspaces in $V$, respectively. And $\dim(W) = r$. Then

$$\widehat{\dim}(U) = \begin{cases} r, & v_0 \in W \\ r + 1, & v_0 \notin W \end{cases} \tag{80}$$

In other words, $\widehat{\dim}(U) \geq \widehat{\dim}(W)$ always holds.

**Lemma 8.** *For arbitrary $k$ ($1 \leq k \leq n - 1$), let $a_1, a_2, ..., a_k$ be $k$ linearly independent vectors in $n$ dimensional linear space $V$. Consider one $n - 1$ dimensional linear subspace $W$ in $V$ and a non-zero vector $v_0$ in $V$. They form a linear manifold $P = \{v_0 + \alpha | \alpha \in W\}$. If $a_1, a_2, ..., a_k$ do not all belong to $P$, then there must exist $n - k$ vectors $p_1, p_2, ..., p_{n-k}$ from $P$, s.t $(a_1, a_2, ..., a_k, p_1, p_2, ..., p_{n-k})$ are a set of basis for the linear space $V$.*

*Proof.* we use mathematical induction. First, show that the Lemma 8 holds for $n - k = 1$. it means we need to find a vector $p_1 \in P$ s.t. $a_1, a_2, ..., a_k, p_1$ linearly independent. If $p_1$ does not exist, then $\forall p \in P$ would be linearly represented by $a_1, a_2, ..., a_k$. In other word,

$$P \subset L = \mathbf{span}(a_1, a_2, ..., a_k), \tag{81}$$

① For the linear manifold $P$, if $v_0 \in W$. This means that $P$ is equal to the linear subspace $W$. Since Eq. (81), we have $W \subset L$ and $\widehat{\dim}(W) = \widehat{\dim}(L)$. Hence, $P = W = L$. However, $a_1, a_2, ..., a_k$ do not all belong to $P$, a contradiction.

② For the linear manifold $P$, if $v_0 \notin W$, then $\widehat{\dim}(P) = n$. Because $v_0 \notin W$, that is, $v_0$ cannot be represented by a set of basis of $W$. In other words, $v_0$ and a set of basis of $W$ are linearly independent. However, the dimension of $W$ is $n - 1$, hence $\widehat{\dim}(P) = n$. From Eq. (81), we have $P \subset L$, so

$$n = \widehat{\dim}(P) \leq \widehat{\dim}(L) = k = n - 1, \tag{82}$$

a contradiction. Therefore, Lemma 8 holds for $n - k = 1$. Assume the induction hypothesis that Lemma 8 is true when $n - k = l$, where $1 \leq l$. when $n - k = l + 1$, *i.e.*, $k = n - (l + 1)$, we also can find a vector $p_1 \in P$ s.t. $a_1, a_2, ..., a_k, p_1$ linearly independent. Otherwise, $\forall p \in P$ would be linearly represented by $a_1, a_2, ..., a_k$. Similarly, we have Eq. (81). Note that, from Definition 1, $\widehat{\dim}(P) \geq n - 1$, hence

$$n - 1 \leq \widehat{\dim}(P) \leq \widehat{\dim}(L) = k = n - (l + 1). \tag{83}$$

a contradiction. At this time, we have $k + 1 = n - (l + 1) + 1 = n - l$ vectors $a_1, a_2, ..., a_k, p_1$ which are not all on $P$. Note that $n - (n - l) = l$, using the induction hypothesis, the Lemma 8 also holds for $n - k = l$. In summary, Lemma 8 holds.

$\square$

**Theorem 5.** Let $v_0, v_1, ..., v_k$ be the $k + 1$ vectors in $n$ dimensional Euclidean space $\mathbb{E}^n$. For all $P$ in $\mathbb{E}^n$,

$$\sum_{i=0}^{k} ||P - v_i||_2^2 = \sum_{i=0}^{k} ||G - v_i||_2^2 + (k+1)||P - G||_2^2.$$

where $G$ is the centroid of $v_i$, will hold if it satisfies one of the following conditions:

(1)if $k \geq n$ and $\mathbf{rank}(v_1 - v_0, v_2 - v_0, ..., v_k - v_0) = n$.

(2)if $k < n$ and $(v_1 - v_0, v_2 - v_0, ..., v_k - v_0)$ are linearly independent.

(3)if $v_i \sim N(\mathbf{0}, c \cdot \mathbf{I}_n)$, Eq.(50) holds with probability 1 where $c$ is a constant.

*Proof.* **For Theorem 5 (1)**. From Lemma 6, $\forall P \in E^n$ ,$\exists \gamma = (\gamma_0, ..., \gamma_k)$, s.t. $P$ can be represented by $\sum_{i=0}^{k} \gamma_i v_i$, where $\sum_{i=0}^{k} \gamma_i = 1$. In fact, for each $v_i$, it also can be respresented by $\sum_{j=0}^{k} \beta_{ij} v_i$, where $\sum_{i=0}^{k} \beta_{ij} = 1$. We just take $(\beta_{i0}, \beta_{i1}, ..., \beta_{ik})$ as one of the standard orthogonal basis $\epsilon_i = (0, 0, ..., 1_i, ...0)$. According to lemma 7,

$$||P - v_i||_2^2 = (\gamma - \epsilon_i)^T D(\gamma - \epsilon_i) \tag{84}$$

$$= \gamma^T D\gamma - 2\gamma^T D\epsilon_i + \epsilon_i^T D\epsilon_i \tag{85}$$

$$= \gamma^T D\gamma - 2\gamma^T D\epsilon_i. \tag{86}$$

The final equation is because the diagonal elements of the matrix are all 0. On the other hand, we have

$$||G - v_i||_2^2 = (\frac{1}{k+1} \sum_{i=0}^{k} \epsilon_i - \epsilon_i)^T D(\frac{1}{k+1} \sum_{i=0}^{k} \epsilon_i - \epsilon_i) \tag{87}$$

$$= \frac{1}{(k+1)^2} \alpha^T D\alpha - \frac{2}{k+1} \alpha^T D\epsilon_i + \epsilon_i^T D\epsilon_i \tag{88}$$

$$= \frac{1}{(k+1)^2} \alpha^T D\alpha - \frac{2}{k+1} \alpha^T D\epsilon_i, \tag{89}$$

where $\alpha = \sum_{i=0}^{k} \epsilon_i$, *i.e.*,$\alpha = (1, 1, ..., 1)$. Next, we consider $||P - G||_2^2$.

$$||P - G||_2^2 = (\gamma - \frac{1}{k+1} \alpha)^T D(\gamma - \frac{1}{k+1} \alpha) \tag{90}$$

$$= \gamma^T D\gamma + \frac{1}{(k+1)^2} \alpha^T D\alpha - \frac{2}{k+1} \gamma^T D\alpha. \tag{91}$$

In summary, we have

$$\sum_{i=0}^{k} ||P - v_i||_2^2 - ||G - v_i||_2^2 = (k+1)\gamma^T D\gamma - 2\gamma^T D\alpha + \frac{1}{k+1} \alpha^T D\alpha \tag{92}$$

$$= (k+1)||P - G||_2^2 \tag{93}$$

Therefore, Theorem 5 (1) holds.

**For Theorem 5 (2)**. Next, we prove the case of $k < n$. Obviously, Lemma 6 does not hold. We consider about such a linear space $W_1 = \mathbf{span}(P - G)$, *i.e.*, a linear space expanded by $P - G$, and its orthogonal complement $W_1^{\perp}$ (in $E^n$). Since dimension formula from linear space, it is easy to konw that $\mathbf{dim}(W_1^{\perp}) = n - 1$.

Two linear manifolds $T_1$ and $T_2$ are constructed as follows,

$$T_1 = \{x + G | x \in W_1^\perp\} \tag{94}$$

$$T_2 = \{x + G - v_0 | x \in W_1^\perp\} \tag{95}$$

$\forall v_i \in T_1$, we have $(v_i - G)^T (P - G) = 0$, Furthermore,

$$||P - v_i||_2^2 = ||v_i - G||_2^2 + ||P - G||_2^2. \tag{96}$$

It is easy to know that $G - v_0$ is not 0. If $v_1 - v_0, ..., v_k - v_0$ are all belong to $T_2$, it means $v_1, .., v_k$ are all in $T_1$. Hence, we have Eq. (96). By summing both sides of Eq. (96) for $i$, it is obvious find that Theorem 5 (2) holds. If $v_1 - v_0, ..., v_k - v_0$ are not all belong to $T_2$, since Lemma 8, there are $n - k$ vectors $p_1 - v_0, p_2 - v_0, .., p_{n-k} - v_0$ from $T_2$ s.t. they and $v_1 - v_0, ..., v_k - v_0$ are linearly independent, where $p_i$ obviously belongs to manifold $T_1$.

At the same time, we have $2G - p_i \in T_1$, we can also construct $n - k$ new vectors $2G - p_i - v_0 \in T_2$ and calculate the rank that

$$\mathbf{rank}(v_1 - v_0, ..., v_k - v_0, p_1 - v_0, ..., p_{n-k} - v_0, 2G - p_1 - v_0, ..., 2G - p_{n-k} - v_0)$$

$$= \mathbf{rank}(v_1 - v_0, ..., v_k - v_0, p_1 - v_0, ..., p_{n-k} - v_0, 2(G - v_0), ..., 2(G - v_0)) \tag{97}$$

$$= \mathbf{rank}(v_1 - v_0, ..., v_k - v_0, p_1 - v_0, ..., p_{n-k} - v_0, 0, ..., 0) \tag{98}$$

$$= n \tag{99}$$

The reason of the final equation is that $\sum_{i=1}^{k}(v_i - v_0) = (k+1)(G - v_0)$. Note that there are a total of $k + (n - k) + (n - k) = n + (n - k) \geq n$ vectors, meets the lemma 6 condition. For the convenience of description, we define

$$L_i^{(1)} = v_i, (0 \leq i \leq k), \tag{100}$$

$$L_i^{(2)} = p_i, (1 \leq i \leq n - k), \tag{101}$$

$$L_i^{(3)} = 2G - p_i, (1 \leq i \leq n - k). \tag{102}$$

And their centroid is

$$G' = \frac{1}{2n - k + 1}\left(\sum_{i=0}^{k} v_i + \sum_{i=1}^{n-k}(L_i^{(2)} + L_i^{(3)})\right) \tag{103}$$

$$= \frac{1}{2n - k + 1}((k+1)G + 2(n - k)G) \tag{104}$$

$$= G \tag{105}$$

That is, the newly added vector does not change the centroid of $v_i$. On the other hand, since both $L_i^{(2)}$ and $L_i^{(3)}$ are in the linear manifold $T_1$, and it meets the conditions of the Eq.(96). Similar to the derivation in the Theorem 5 (1), we have

$$(2n - k + 1)||P - G||_2^2 = \sum_{t=L_i^{(1)}, L_i^{(2)}, L_i^{(3)}} \left(||P - t||_2^2 - ||G - t||_2^2\right) \tag{106}$$

$$= \sum_{i=0}^{k} \left(||P - v_i||_2^2 - ||G - v_i||_2^2\right) + \sum_{t=L_i^{(2)}, L_i^{(3)}} \left(||P - t||_2^2 - ||G - t||_2^2\right)$$

$$\tag{107}$$

$$= \sum_{i=0}^{k} \left(||P - v_i||_2^2 - ||G - v_i||_2^2\right) + 2(n - k)||P - G||_2^2 \tag{108}$$

The final equation is because both $L_i^{(2)}$ and $L_i^{(3)}$ are in the linear manifold $T_1$ and satisfy Eq. (96). To simplify Eq. (108), we obtain $\sum_{i=0}^{k} \left(||P - v_i||_2^2 - ||G - v_i||_2^2\right) = (k+1)||P - G||_2^2$. Therefore, Theorem 5 (2) holds.

**For Theorem 5 (3)**. When $k \geq n$, from Proposition 14, we know that $\mathbf{rank}(v_1 - v_0, v_2 - v_0, ..., v_k - v_0) = n$ holds with probability 1. Hence, if we use the similar deduction from Theorem 5 (1), we can find that Theorem 5 (3) holds when $k \geq n$. On the other hand, when $k < n$, we can get the same result also according to Proposition 14. The reason is that $(v_1 - v_0, v_2 - v_0, ..., v_k - v_0)$ are linearly independent with probability 1.

$\square$

# H  The result of Sp

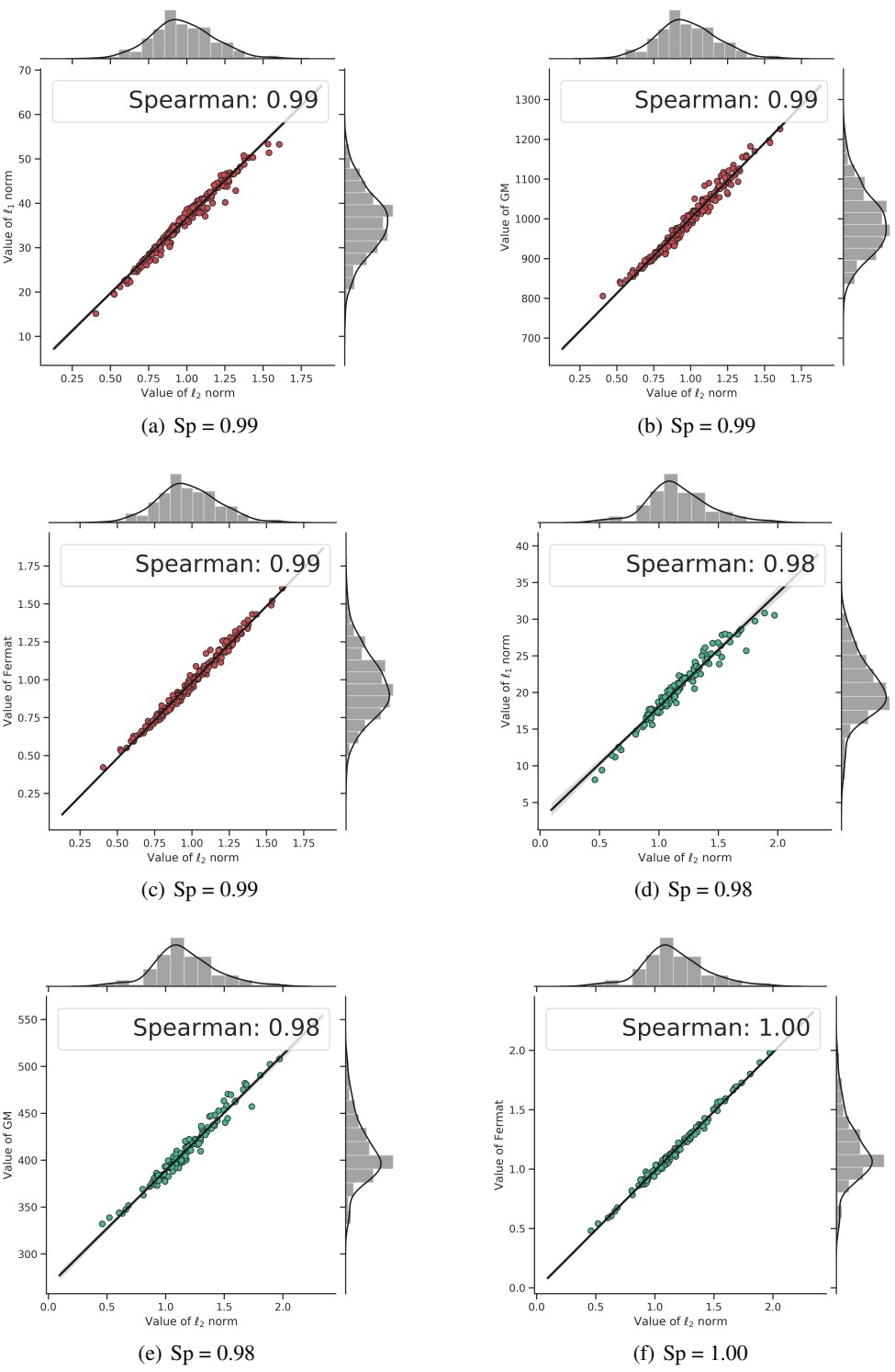

Figure 12: The Spearman's rank correlation coefficient (Sp) for different criteria. (a-c) are Sp between $\ell_1$ and $\ell_2$, **GM** and $\ell_2$, **Fermat** and $\ell_2$ from ResNet18 (12$^{\text{th}}$ Conv), respectively. The results of VGG16 (3$^{\text{rd}}$ Conv) are shown in (d-f). If the Sp of two pruning criteria is close to 1, then the sequence of their pruned filters may have strong similarity.

# I  Other result

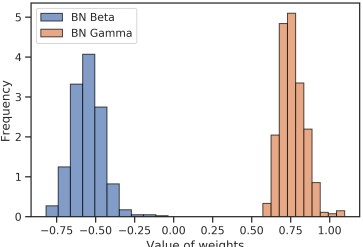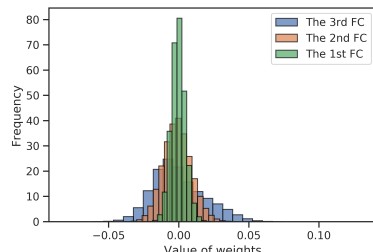

Figure 13: The distribution about other learnable parameters. (Left): The disrtibution about the learnable parameters of batch normalization. (Rihgt): The parameters distribution of the fully-connected layers (FC). For FC, the Sp between the criteria in Table2 are greater than 0.9.

In Fig 13, we show the other learnable parameters (*i.e.* Batch normalization (BN) and fully connected neural network (FC)) in VGG16-BN. For BN, the distribution of its parameters does not satisfy CWDA, and similar results are shown in [34, 35]. Moreover, the learnable parameters of fully-connected layers also do not follow a Gaussian-alike distribution, which is consistent with the conclusion in previous work [36, 37, 38].

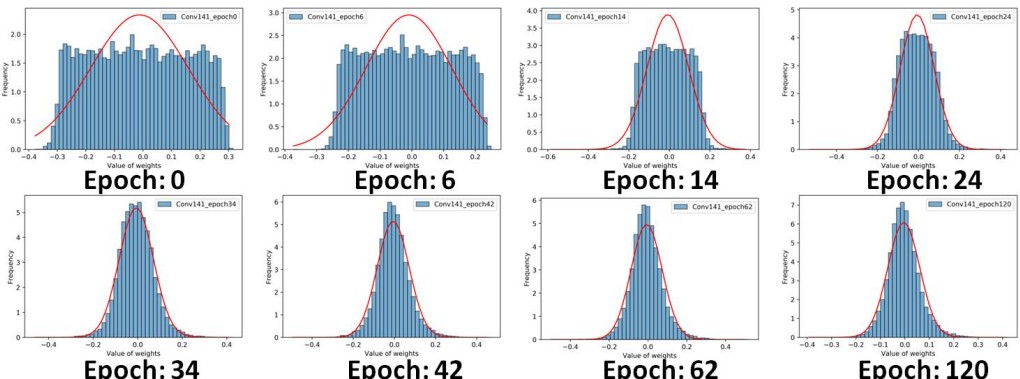

Figure 14: The distribution of the convolutional filter (141[th] Conv) with kaiming-uniform initialization for each epoch.

# J An interesting case for *Importance Score* measured by different criteria

The following results are the index of pruned filters obtained by the filters' *Importance Score* from different types of pruning criteria. We take VGG16 ($2^{nd}$) as an example. The $5^{th}$ filter in this layer is regarded as a redundant convolutional filter for APoZ criterion, but other criteria consider it to be almost the most important.

Taylor $\ell_1$: [27, 36, 25, 11, 6, 23, 24, 16, 0, 57, 48, 53, 1, 61, 18, 55, 34, 15, 51, 58, 31, 3, 12, 21, 59, 30, 7, 38, 41, 50, 10, 33, 17, 46, 62, 13, 49, 43, 42, 47, 2, 32, 44, 20, 39, 52, 56, 40, 9, 26, 37, 22, 29, 54, 60, 8, 14, 45, 4, 63, 19, 35, 28, **5**]

Taylor $\ell_2$: [23, 32, 36, 11, 62, 16, 30, 59, 10, 13, 2, 50, 38, 0, 46, 43, 21, 26, 15, 22, 7, 51, 39, 33, 14, 58, 9, 40, 57, 6, 61, 44, 20, 48, 3, 53, 41, 56, 17, 12, 18, 31, 4, 1, 25, 19, 63, 24, 54, 45, 52, 37, 55, 47, 34, 35, 8, 29, 42, 27, 49, 28, 60, **5**]

BN_$\beta$: [52, 46, 32, 21, 14, 29, 17, 0, 19, 36, 1, 51, 44, 40, 41, 60, 57, 27, 22, 53, 63, 8, 30, 26, 23, 58, 39, 18, 9, 47, 31, 35, 11, 37, 55, 45, 3, 61, 6, 4, 33, 25, 15, 48, 43, 28, 56, 2, 13, 16, 34, 20, 59, 10, 7, 24, 50, 62, 12, 49, 38, 42, **5**, 54]

APoZ: [**5**, 10, 38, 42, 62, 24, 13, 12, 7, 28, 59, 15, 23, 11, 16, 56, 34, 35, 57, 19, 2, 49, 43, 25, 6, 63, 61, 36, 9, 27, 33, 20, 48, 58, 55, 18, 51, 31, 1, 0, 53, 37, 26, 29, 47, 60, 8, 44, 41, 46, 21, 17, 14, 32, 52, 22, 39, 3, 40, 30, 4, 45, 50, 54]

# K   The details of other pruning criteria

For notation, we denote $i^{\text{th}}$ convolutional filter in layer $l$ as $F_i^l$ and the input feature maps in layer $l$ as $\mathbf{I}^l \in \mathbb{R}^{N \times I^l \times H^l \times W^l}$, where $N, I^l, H^l, W_l$ mean the train set size, number of channels, height and width respectively, $i = 1, 2, \cdots, \lambda_l$, and $l = 1, 2, \cdots, L$. The formulation of the filters' *Importance Score* under each pruning criteria are illustrated as follows:

**Norm-based criteria:**

- $\ell_1$-Norm [5]: $||F_i^l||_1$;
- $\ell_2$-Norm [7]: $||F_i^l||_2$;

**BN-based criteria [12]:**

- BN_$\gamma$: $|\gamma_i^l|$, where $\gamma_i^l$ is the scaling factor in the Batch Normalization layer $l$;
- BN_$\beta$: $|\beta_i^l|$, where $\beta_i^l$ is the shifting factor in the Batch Normalization layer $l$.

**Activation-based criteria:**

- APoZ [8]: $\frac{\sum_{p,q} \mathbb{1}\left((|\mathbf{I}^l * F_i^l|)_{p,q} > \sigma\right)}{N \times I^l \times H^l \times W^l}$, where we set $\sigma = 0.0001$ same as [9], and $\mathbb{1}(\cdot)$ is the indicator function, $*$ is convolution operator and $\mathbf{I}^l * F_i^l$ is the $i$-th output feature map;
- Entropy [9]: we first prepare $\mathbf{G}_i^l = GAP(\mathbf{I}^l * F_i^l)$, where $\mathbf{G}_i^l \in \mathbb{R}^{N \times 1}$ and $GAP(\cdot)$ is the Global Average Pooling. Then, we estimate statistical distribution for $\mathbf{G}_i^l$ by dividing all elements in $\mathbf{G}_i^l$ into $m$ bins. Let $p_j$ is the probability of bin $j$, and the the *Importance Score* score is $-\sum_{j=1}^m p_j \log p_j$.

**First order Taylor based criteria [10, 11, 26]:**

- Taylor $\ell_1$-Norm: $||\frac{\partial loss}{\partial F_i^l} \cdot F_i^l||_1$;
- Taylor $\ell_2$-Norm: $||\frac{\partial loss}{\partial F_i^l} \cdot F_i^l||_2$;

The $loss$ is the Cross Entropy Loss on the split training set from the original training set.

# L  Additional experiments about image clasification

Table 5: The accuracy(%) of several networks and datasets using different pruning criteria.

| | | Experiment (1) | | | Experiment (2) | | | Experiment (3) | | |
|---|---|---|---|---|---|---|---|---|---|---|
| | | Trained | Pruned | Fine-tuned | Trained | Pruned | Fine-tuned | Trained | Pruned | Fine-tuned |
| CIFAR10 | $\ell_1$ | 93.61 | 61.21 | 93.51 | 93.21 | 54.31 | 93.22 | 93.26 | 57.74 | 93.32 |
| VGG16 | $\ell_2$ | 93.61 | 63.41 | 93.32 | 93.21 | 54.61 | 93.42 | 93.26 | 57.42 | 93.29 |
| | **GM** | 93.61 | 61.22 | 93.41 | 93.21 | 53.71 | 93.25 | 93.26 | 57.46 | 93.36 |
| CIFAR100 | $\ell_1$ | 72.67 | 25.91 | 71.50 | 72.99 | 20.43 | 71.36 | 72.56 | 24.01 | 71.07 |
| VGG16 | $\ell_2$ | 72.67 | 27.07 | 71.28 | 72.99 | 22.31 | 71.12 | 72.56 | 24.45 | 70.92 |
| | **GM** | 72.67 | 26.37 | 71.27 | 72.99 | 21.67 | 71.26 | 72.56 | 24.26 | 70.78 |
| ImageNet | $\ell_1$ | 71.58 | 30.33 | 71.02 | 71.33 | 40.33 | 70.12 | 72.01 | 28.07 | 70.93 |
| VGG16 | $\ell_2$ | 71.58 | 29.47 | 70.83 | 71.33 | 40.45 | 70.13 | 72.01 | 27.89 | 71.02 |
| | **GM** | 71.58 | 30.76 | 70.95 | 71.33 | 39.86 | 70.33 | 72.01 | 28.01 | 70.74 |
| CIFAR10 | $\ell_1$ | 92.98 | 77.73 | 93.08 | 92.97 | 76.02 | 92.82 | 93.01 | 79.93 | 92.81 |
| ResNet56 | $\ell_2$ | 92.98 | 79.02 | 92.83 | 92.97 | 77.91 | 92.72 | 93.01 | 82.43 | 92.81 |
| | **GM** | 92.98 | 74.26 | 92.77 | 93.2 | 73.93 | 92.61 | 93.01 | 80.48 | 92.84 |
| CIFAR100 | $\ell_1$ | 71.36 | 50.64 | 70.15 | 70.02 | 52.41 | 69.19 | 70.48 | 52.19 | 69.77 |
| ResNet56 | $\ell_2$ | 71.36 | 53.44 | 70.16 | 70.02 | 52.73 | 69.31 | 70.48 | 52.16 | 69.62 |
| | **GM** | 71.36 | 45.12 | 70.22 | 70.02 | 52.62 | 69.54 | 70.48 | 50.74 | 69.69 |
| ImageNet | $\ell_1$ | 73.31 | 62.22 | 73.06 | 73.16 | 54.24 | 72.99 | 73.21 | 63.12 | 73.02 |
| ResNet34 | $\ell_2$ | 73.31 | 62.02 | 72.91 | 73.16 | 53.64 | 72.78 | 73.21 | 62.98 | 72.86 |
| | **GM** | 73.31 | 61.88 | 72.96 | 73.16 | 53.48 | 72.94 | 73.21 | 62.36 | 73.04 |

All the setting of these experiments are under can be found in `https://github.com/bearpaw/pytorch-classification`. Specifically, for pruning ratio:

VGG16 on CIFAR10, CIFAR100 and ImageNet:

`https://github.com/Eric-mingjie/rethinking-network-pruning/blob/master/cifar/l1-norm-pruning/vggprune.py#L84`

ResNet56 on CIFAR10 and CIFAR100:

`https://github.com/Eric-mingjie/rethinking-network-pruning/blob/master/cifar/l1-norm-pruning/res56prune.py#L94`

ResNet34 on ImageNet:

`https://github.com/Eric-mingjie/rethinking-network-pruning/blob/master/imagenet/l1-norm-pruning/prune.py#L138`

# M  About weight decay

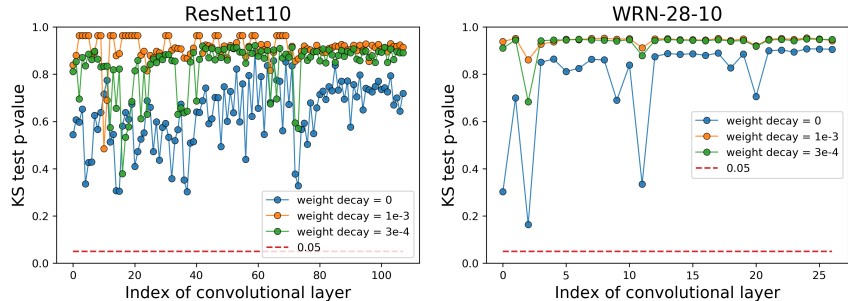

Figure 15: KS test [39] while using different settings of weight decay.

We train the ResNet110 and WRN-28-10 on CIFAR100 with different weight decay (1e-3, 3e-4 and 0) and use KS test to verify whether the parameters of different layers follow a normal distribution. In Fig. 15, we can find

(1) When weight decay (wd) is non-zero, the normality is higher than that when weight decay is 0.

(2) If weight decay is 0, the p-value can still be much greater than 0.05, which means that the regularization of weight decay may not be the key reason for CWDA. The distribution of the parameters in these two networks (weight decay is 0) are shown in Fig. 17 and Fig. 16.

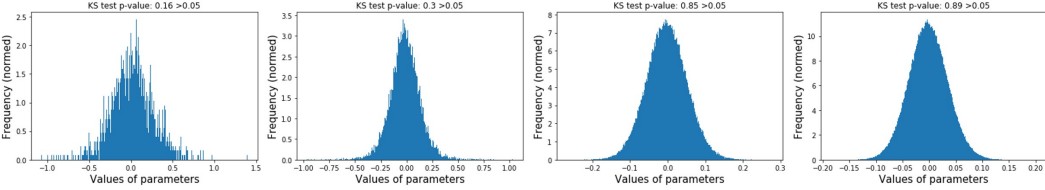

Figure 16: The distribution of parameters in different convolutional filters (WRN-28-10, wd = 0).

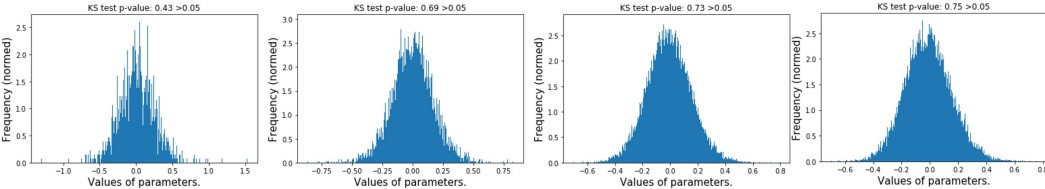

Figure 17: The distribution of parameters in different convolutional filters (ResNet110, wd = 0).

# N More visualizations of correlation matrix

## N.1 VGG16

### VGG16-kernel size = 3

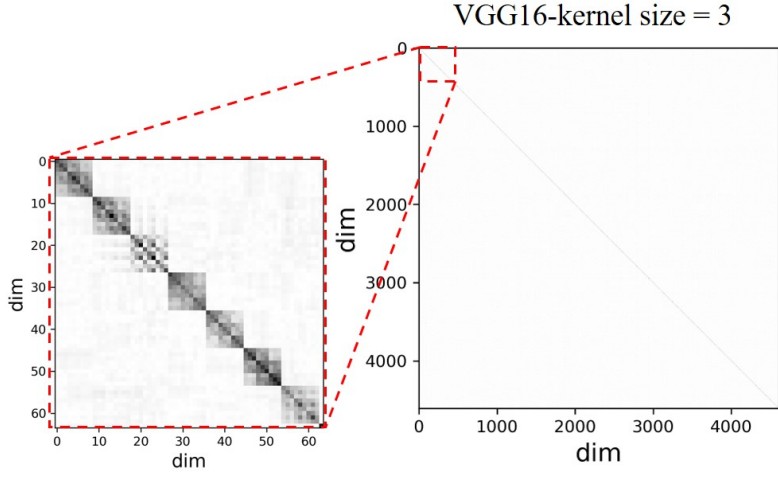

### VGG16-kernel size = 3

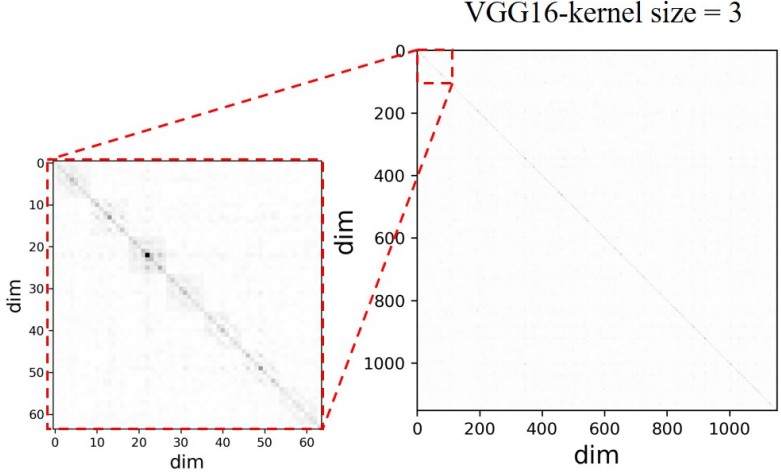

## N.2   VGG19

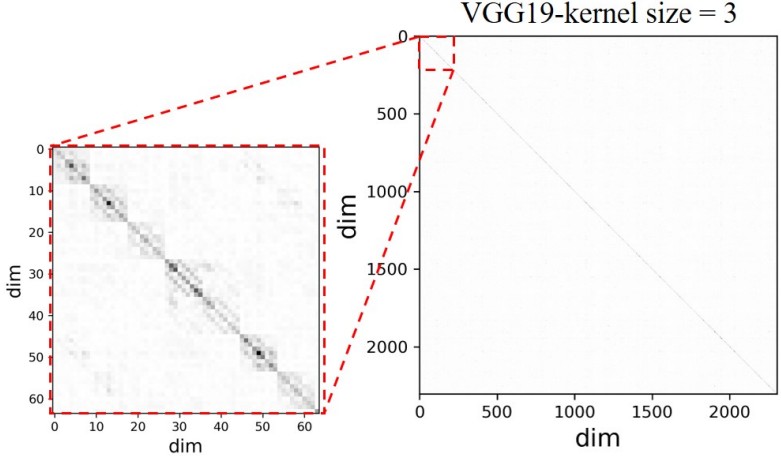

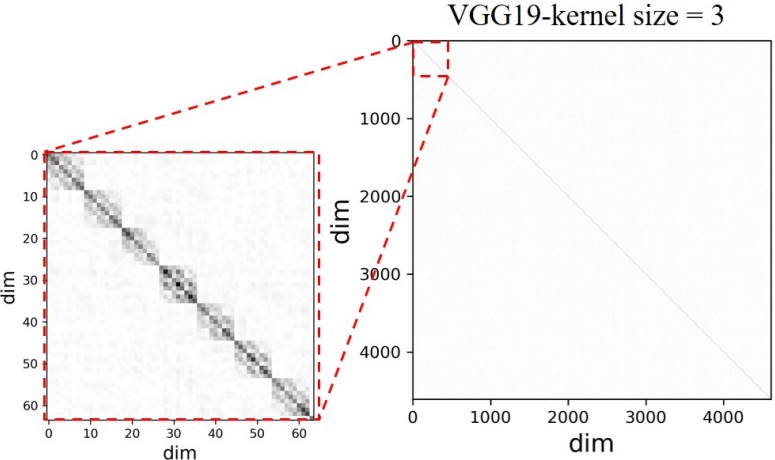

**N.3 ResNet18**

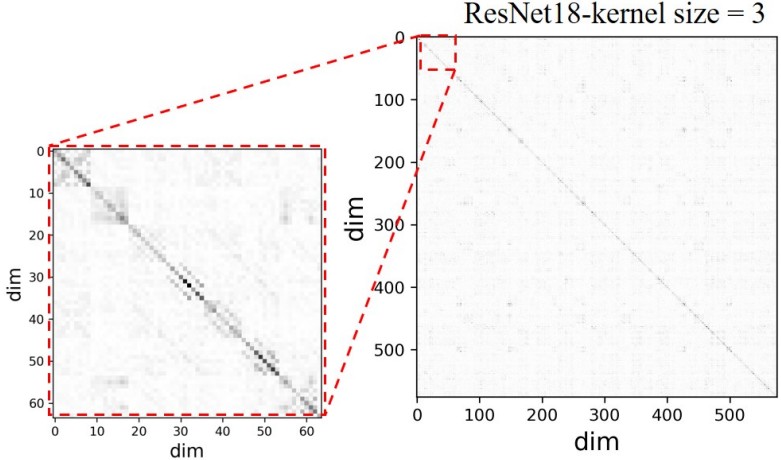

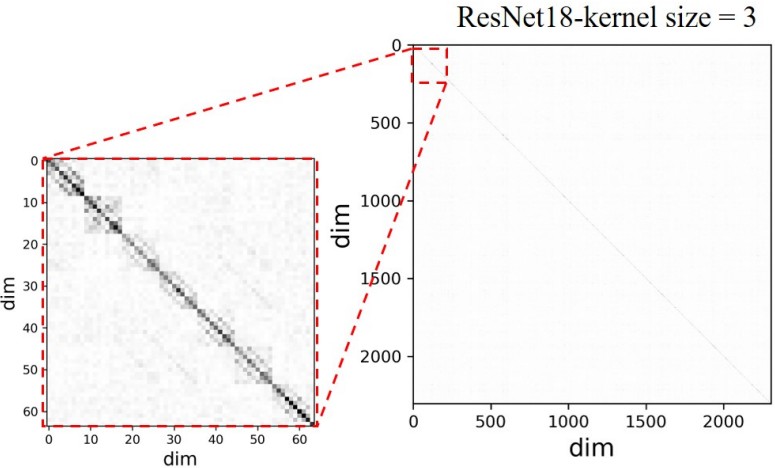

**N.4 ResNet50**

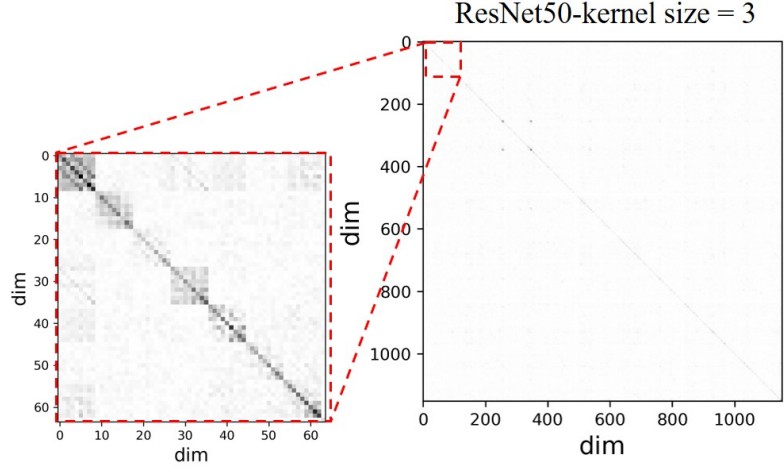

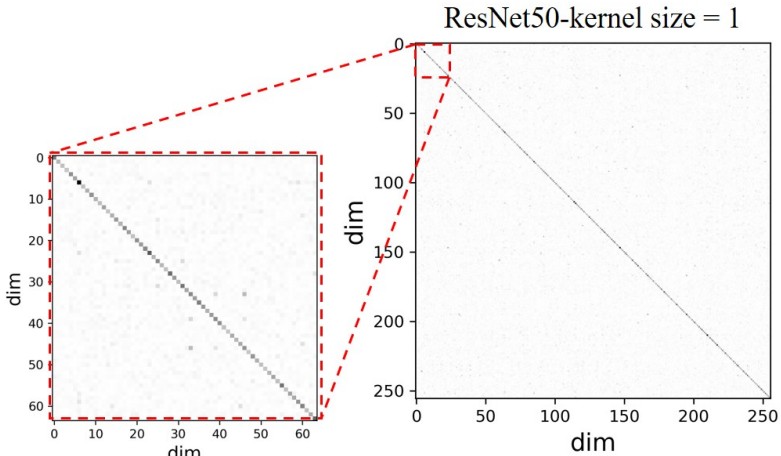

## N.5   AlexNet

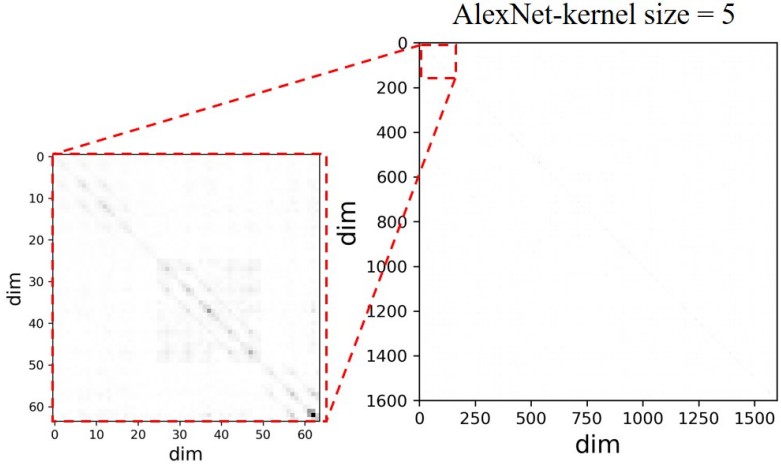

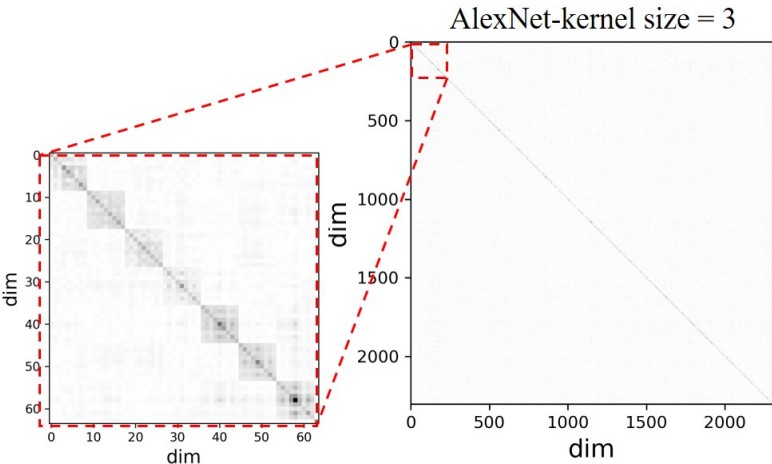

## N.6 DenseNet

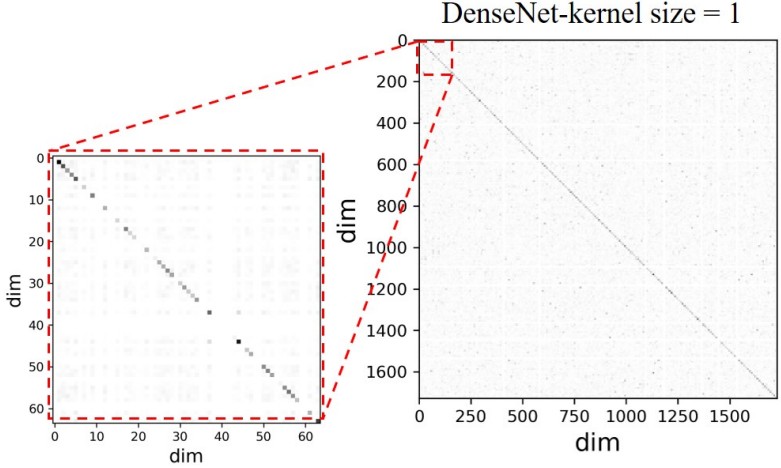

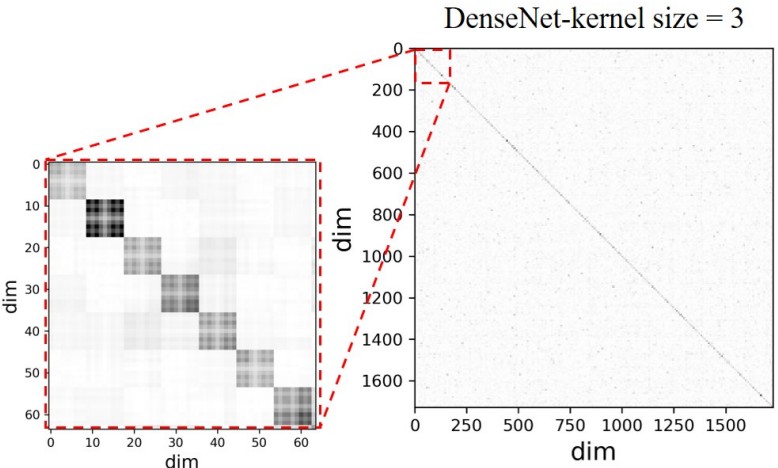

**N.7  ResNext**

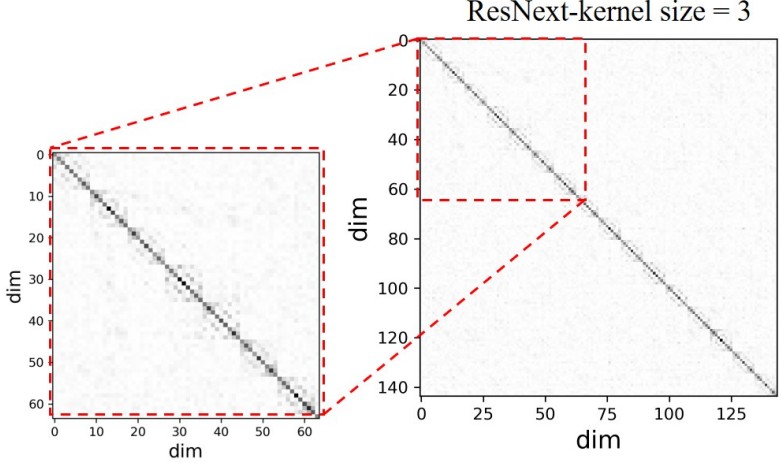

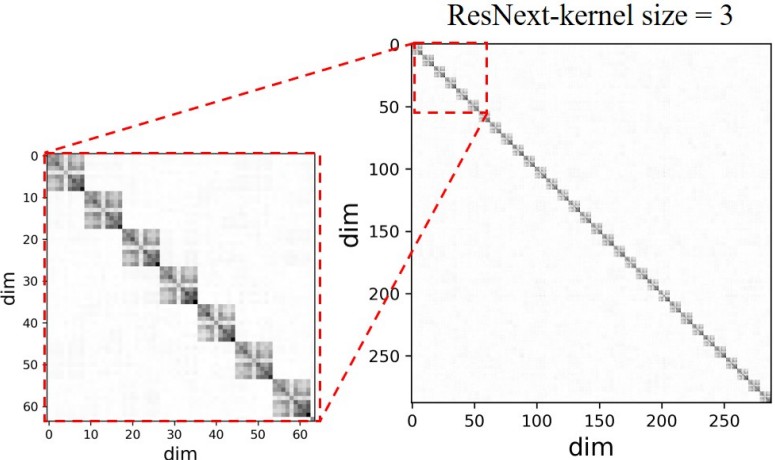

**N.8 MobileNet**

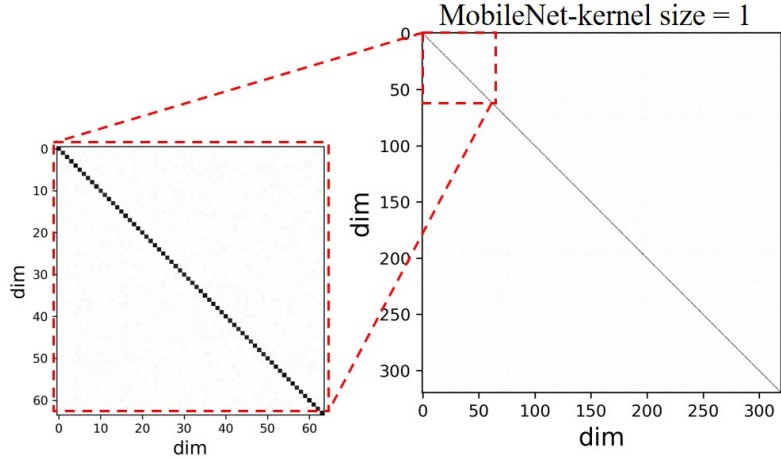

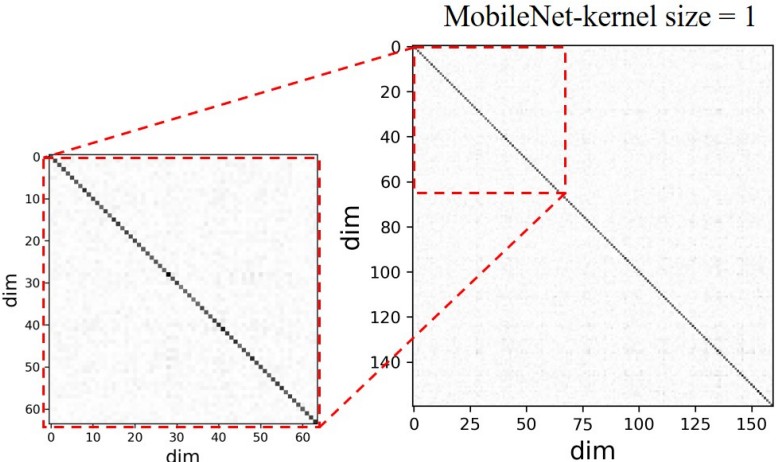

## O   More experiments for supporting our analysis in global pruning

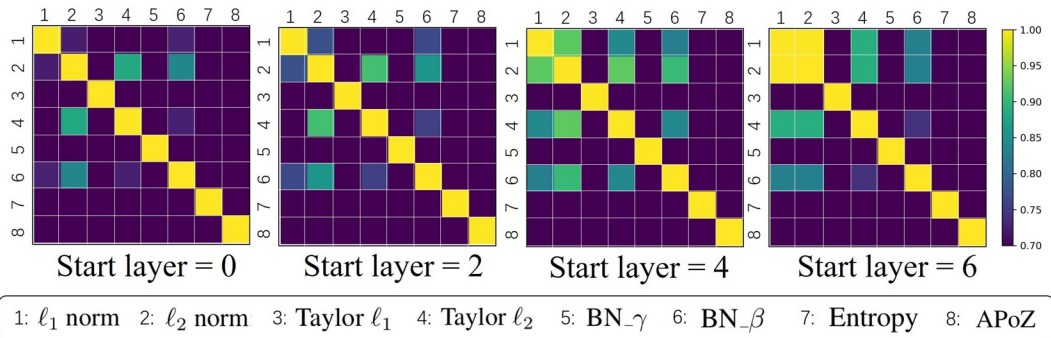

Figure 18: Global pruning with different start layer.

For VGG16. As shown in Fig.6 (a-b), compared with ResNet56, VGG16 has some layers with different dimensions but similar *Importance Score* measured by $\ell_1$ or $\ell_2$, such as "layer 2" and "layer 8" for $\ell_2$ criterion in Fig.6 (a). From Table 3 (3-4), these pairs of layers make the Sp small, which explain why the result of $\ell_1$ and $\ell_2$ pruning is not similar in Fig. 5 (e) for VGG16. We consider a special class of global pruning, *i.e.,* the convolutional filters from one middle layer (called "Start layer") to the last layer are pruned globally. According to our analysis and Fig.6 (a-b), we can deduce that when "Start layer" $\geq 4$, the Sp between $\ell_1$ and $\ell_2$ is large enough. The experiments in Fig.18 are consistent with our analysis, which imply our analysis is reasonable.

# P  Statistical Test

In this section, according to Section 2.1, we have a series of statistical tests for the necessary conditions of CWDA. let $F_{ij} \in \mathbb{R}^{N_i \times k \times k}$ represent the $j^{\text{th}}$ filter of the $i^{\text{th}}$ convolutional layer.[10]

(1) **Gaussian**. We verify whether $F_{ij}$ approximatively follow a Gaussian-alike distribution. In $i^{\text{th}}$ layer, we use Kolmogorov–Smirnov (KS) test [39] to check if all the weights in the same layer follow a normal distribution.

(2) **Variance**. We verify whether the variance of the diagonal elements of $\Sigma_{\text{diag}}$ are small enough. Since Appendix B, Let $\sigma_j$ denotes the standard deviation of all the weights of filter $F_{ij}$ in $i^{\text{th}}$ layer. We use Student's t test [40] to check if the variance of these $\sigma_j$ is small enough. The null hypothesis $H_0$ and the alternative hypothesis $H_1$ are:

$$H_0 : \mathbf{Var}(\sigma_1^2, \sigma_2^2, .., \sigma_{N_i}^2) \leq \sigma_0^2, \qquad H_1 : \mathbf{Var}(\sigma_1^2, \sigma_2^2, .., \sigma_{N_i}^2) > \sigma_0^2.$$

where $N_i$ denotes the number of the filters in $i^{\text{th}}$ layer and $\sigma_0$ is a given real number which is small enough, like $\sigma_0^2 = 0.0001$.

(3) **Mean**. We verify whether the mean of $F_{ij}$ is 0. Let the mean of all the weights in the same layer is $\mu$. We use Student's t test [40] to check if $\mu$ is close to 0. First, we check the upper bound (Mean-Left) of $\mu$, *i.e.*,

$$H_0 : \mu \leq \epsilon_0, \qquad H_1 : \mu > \epsilon_0.$$

where $\epsilon_0$ is a small constant, like $\epsilon_0 = 0.01$. Next, we check the lower bound (Mean-Right) and the null hypothesis $H_0$ and the alternative hypothesis $H_1$ are:

$$H_0 : \mu \geq -\epsilon_0, \qquad H_1 : \mu < -\epsilon_0.$$

(4) **Magnitude**. We verify whether $\epsilon$ is small enough. Let $h$ denote the mean of the off-diagonal elements of $\Sigma_{\text{diag}} + \epsilon \cdot \Sigma_{\text{block}}$.

$$H_0 : h \leq \epsilon_0, \qquad H_1 : h > \epsilon_0.$$

Table 6: The experiments for having the comprehensive statistical tests on CWDA.

| NETWORK STRUCTURE | OPTIMIZER | REGULARIZATION |
|---|---|---|
| ResNet [41] | SGD [42] | L1 norm |
| VGG [43] | ASGD [44] | L2 norm |
| AlexNet [45] | Adam [46] | RReLu [47] |
| DenseNet [48] | Adagrad [49] | Dropact [50] |
| PreResNet [51] | Adamax [46] | Autoaug [52] |
| WRN [53] | Adadelta [54] | Cutout [55] |
| ResNext [56] | | Cutmix [57] |
| **ATTENTION MECHANISM** | **INITIALIZATION** | **DATASET** |
| SENet [58] | Kaiming-normal [59] | CIFAR10 [60] |
| DIANet [61] | Kaiming-uniform [59] | CIFAR100 [60] |
| SRMNet [62] | Xavier-normal [63] | ImageNet [64] |
| CBAM [65] | Xavier-uniform [63] | MNIST [66] |
| IEBN [67] | Orthogonal [68] | |
| SGENet [69] | | |
| **SEGMENTATION** | **DETECTION** | **BATCH NORMALIZATION** |
| SegNet [70] | Faster RCNN [71] | VGG |
| PSPNet [72] | | VGG-bn |
| **PYTORCH PRETRAIN** | **MATTING** | **LEARNING RATE** |
| ResNet18/34/50 | Deep image matting [73] | Schedule150-225 |
| VGG11/16/19 | AlphaGAN matting [74] | Schedule82-164 |
| **STYLE TRANSFER** | **GAN** | Schedule60-120 |
| Fast neural style [75] | DCGAN [76] | Cos-lr [77] |

---

[10] The statistical tests about the situation with or without weight decay can be found in Appendix M.

Next, we show the passing rate about the statistical tests for different situations. "in the front of network" denotes whether all the failed cases are the layers whose position is in the front of the network.

For Network structure: `https://github.com/bearpaw/pytorch-classification`.

Table 7: **Network structure**.

| Experiments | Remark | Gaussian | Variance | Mean | Magnitude | in the front of network? |
|---|---|---|---|---|---|---|
| ResNet164 | CIFAR100 | 98.77% | 97.55% | 100% | 97.55% | ✓ |
| VGG16 | CIFAR100 | 100% | 93.75% | 100% | 100% | ✓ |
| AlexNet | CIFAR100 | 100% | 100% | 100% | 100% | ✓ |
| DenseNet-BC-100-12 | CIFAR100 | 100% | 98.99% | 100% | 98.99% | ✓ |
| PreResNet110 | CIFAR100 | 100% | 99.08% | 100% | 100% | ✓ |
| WRN28-10 | CIFAR100 | 100% | 100% | 100% | 100% | ✓ |
| ResNext-16x64d | CIFAR100 | 100% | 100% | 100% | 100% | ✓ |
| ResNet164 | CIFAR10 | 100.00% | 97.55% | 100% | 97.55% | ✓ |
| VGG16 | CIFAR10 | 100% | 93.75% | 100% | 93.75% | ✓ |
| AlexNet | CIFAR10 | 100% | 100% | 100% | 100% | ✓ |
| DenseNet-BC-100-12 | CIFAR10 | 100% | 100% | 100% | 98.99% | ✓ |
| PreResNet110 | CIFAR10 | 100% | 99.08% | 100% | 100% | ✓ |
| WRN28-10 | CIFAR10 | 100% | 100% | 100% | 100% | ✓ |
| ResNext-16x64d | CIFAR10 | 100% | 100% | 100% | 100% | ✓ |

For Optimizer: `https://pytorch.org/docs/master/optim.html#torch-optim`.

Table 8: **Optimizer**

| Experiments | Remark | Gaussian | Variance | Mean | Magnitude | in the front of network? |
|---|---|---|---|---|---|---|
| ASGD | ResNet164 | 100% | 99.39% | 99.39% | 100% | ✓ |
| Adam | ResNet164 | 99.39% | 90.18% | 100% | 99.39% | ✗ |
| Adagrad | ResNet164 | 100% | 99.39% | 100% | 100% | ✓ |
| Adamax | ResNet164 | 100% | 96.93% | 100% | 99.39% | ✗ |
| Adadelta | ResNet164 | 100% | 100% | 100% | 100% | ✓ |
| SGD | ResNet164 | 98.77% | 97.55% | 100% | 97.53% | ✓ |
| ASGD | VGG16 | 100% | 100% | 93.75% | 100% | ✓ |
| Adam | VGG16 | 93.75% | 93.75% | 100% | 100.00% | ✓ |
| Adagrad | VGG16 | 100% | 100% | 100% | 100% | ✓ |
| Adamax | VGG16 | 100% | 100% | 100% | 93.75% | ✗ |
| Adadelta | VGG16 | 100% | 100% | 100% | 100% | ✓ |
| SGD | VGG16 | 100% | 93.75% | 100% | 100% | ✓ |
| ASGD | AlexNet | 100% | 100% | 100% | 100% | ✓ |
| Adam | AlexNet | 100% | 100% | 100% | 100% | ✓ |
| Adagrad | AlexNet | 100% | 100% | 100% | 100% | ✓ |
| Adamax | AlexNet | 100% | 100% | 100% | 100% | ✓ |
| Adadelta | AlexNet | 100% | 100% | 100% | 100% | ✓ |
| SGD | AlexNet | 100% | 100% | 100% | 100% | ✓ |

For Regularization:`https://github.com/LeungSamWai/Drop-Activation`

`https://github.com/uoguelph-mlrg/Cutout`

`https://github.com/clovaai/CutMix-PyTorch`

`https://github.com/DeepVoltaire/AutoAugment`

For Attention:`https://github.com/moskomule/senet.pytorch`

`https://github.com/gbup-group/DIANet`

`https://github.com/EvgenyKashin/SRMnet`

Table 9: **Regularization**

| Experiments | Remark | Gaussian | Variance | Mean | Magnitude | in the front of network? |
|---|---|---|---|---|---|---|
| L1 norm | ResNet164 | 100% | 99.39% | 99.39% | 100% | ✓ |
| L2 norm | ResNet164 | 98.77% | 97.53% | 100% | 97.53% | ✓ |
| RReLU | ResNet164 | 100% | 99.39% | 100% | 100% | ✓ |
| Dropact | ResNet164 | 100% | 96.93% | 100% | 99.39% | ✓ |
| Autoaugment | ResNet164 | 100% | 96.93% | 100% | 99.39% | ✓ |
| Cutout | ResNet164 | 100% | 100% | 100% | 100% | ✓ |
| Cutmix | ResNet164 | 98.77% | 97.53% | 100% | 97.53% | ✓ |
| L1 norm | WRN28-10 | 100% | 96.43% | 100% | 96.43% | ✓ |
| L2 norm | WRN28-10 | 100% | 100% | 100% | 100% | ✓ |
| RReLU | WRN28-10 | 100% | 96.43% | 100% | 100% | ✓ |
| Dropact | WRN28-10 | 100% | 96.43% | 100% | 100% | ✓ |
| Autoaugment | WRN28-10 | 100% | 96.43% | 100% | 100% | ✓ |
| Cutout | WRN28-10 | 100% | 96.43% | 100% | 100% | ✓ |
| Cutmix | WRN28-10 | 100% | 100% | 100% | 100% | ✓ |
| L1 norm | VGG16 | 100% | 93.75% | 100% | 100% | ✓ |
| L2 norm | VGG16 | 100% | 93.75% | 100% | 100% | ✓ |
| RReLU | VGG16 | 100% | 93.75% | 100% | 93.75% | ✓ |
| Dropact | VGG16 | 100% | 93.75% | 100% | 100% | ✓ |
| Autoaugment | VGG16 | 100% | 93.75% | 100% | 100% | ✓ |
| Cutout | VGG16 | 100% | 93.75% | 93.75% | 93.75% | ✓ |
| Cutmix | VGG16 | 100% | 93.75% | 100% | 100% | ✓ |
| L1 norm | PreResNet110 | 100% | 99.08% | 100% | 100% | ✓ |
| L2 norm | PreResNet110 | 100% | 99.08% | 100% | 100% | ✓ |
| RReLU | PreResNet110 | 100% | 100% | 100% | 100% | ✓ |
| Dropact | PreResNet110 | 100% | 99.08% | 100% | 100% | ✓ |
| Autoaugment | PreResNet110 | 100% | 100% | 100% | 100% | ✓ |
| Cutout | PreResNet110 | 100% | 99.08% | 99.08% | 99.08% | ✓ |
| Cutmix | PreResNet110 | 100% | 99.08% | 100% | 100% | ✓ |
| L1 norm | AlexNet | 100% | 100% | 100% | 100% | ✓ |
| L2 norm | AlexNet | 100% | 100% | 100% | 100% | ✓ |
| RReLU | AlexNet | 100% | 100% | 100% | 100% | ✓ |
| Dropact | AlexNet | 100% | 100% | 100% | 100% | ✓ |
| Autoaugment | AlexNet | 100% | 100% | 100% | 100% | ✓ |
| Cutout | AlexNet | 100% | 100% | 100% | 100% | ✓ |
| Cutmix | AlexNet | 100% | 100% | 100% | 100% | ✓ |
| L1 norm | DenseNet-BC-100-12 | 100% | 98.99% | 100% | 98.99% | ✓ |
| L2 norm | DenseNet-BC-100-12 | 100% | 98.99% | 100% | 98.99% | ✓ |
| RReLU | DenseNet-BC-100-12 | 100% | 98.99% | 100% | 98.99% | ✓ |
| Dropact | DenseNet-BC-100-12 | 98.99% | 98.99% | 98.99% | 98.99% | ✓ |
| Autoaugment | DenseNet-BC-100-12 | 100% | 98.99% | 100% | 98.99% | ✓ |
| Cutout | DenseNet-BC-100-12 | 100% | 98.99% | 98.99% | 98.99% | ✓ |
| Cutmix | DenseNet-BC-100-12 | 100% | 98.99% | 100% | 98.99% | ✓ |

```
https://github.com/luuuyi/CBAM.PyTorch
https://github.com/gbup-group/IEBN
https://github.com/implus/PytorchInsight
```

Table 10: **Attention**

| Experiments | Remark | Gaussian | Variance | Mean | Magnitude | in the front of network? |
|---|---|---|---|---|---|---|
| SENet | ResNet164 | 99.39% | 99.39% | 100% | 100% | ✓ |
| DIANet | ResNet164 | 99.39% | 99.39% | 100% | 100% | ✓ |
| SRMNet | ResNet164 | 99.39% | 97.55% | 100% | 99.39% | ✓ |
| CBAM | ResNet164 | 99.39% | 99.39% | 100% | 100% | ✓ |
| IEBN | ResNet164 | 99.39% | 99.39% | 99.39% | 99.39% | ✓ |
| SGENet | ResNet164 | 99.39% | 98.77% | 100% | 100% | ✓ |
| SENet | VGG16 | 100% | 93.75% | 100% | 100% | ✓ |
| DIANet | VGG16 | 100% | 93.75% | 100% | 93.75% | ✓ |
| SRMNet | VGG16 | 100% | 100% | 100% | 100% | ✓ |
| CBAM | VGG16 | 100% | 93.75% | 100% | 100% | ✓ |
| IEBN | VGG16 | 100% | 93.75% | 93.75% | 93.75% | ✓ |
| SGENet | VGG16 | 100% | 93.75% | 100% | 100% | ✓ |
| SENet | PreResNet110 | 99.08% | 100% | 100% | 100% | ✓ |
| DIANet | PreResNet110 | 100% | 99.08% | 100% | 100% | ✓ |
| SRMNet | PreResNet110 | 100% | 99.08% | 99.08% | 100% | ✓ |
| CBAM | PreResNet110 | 100% | 100% | 100% | 100% | - |
| IEBN | PreResNet110 | 100% | 99.08% | 100% | 99.08% | ✓ |
| SGENet | PreResNet110 | 100% | 100% | 100% | 99.08% | ✓ |
| SENet | DenseNet-BC-100-12 | 100% | 100% | 100% | 100% | ✓ |
| DIANet | DenseNet-BC-100-12 | 98.99% | 98.99% | 100% | 100% | ✓ |
| SRMNet | DenseNet-BC-100-12 | 100% | 98.99% | 98.99% | 98.99% | ✓ |
| CBAM | DenseNet-BC-100-12 | 100% | 100% | 100% | 98.99% | ✓ |
| IEBN | DenseNet-BC-100-12 | 100% | 98.99% | 100% | 100% | ✓ |
| SGENet | DenseNet-BC-100-12 | 100% | 100% | 98.99% | 100% | ✓ |
| SENet | WRN28-10 | 100% | 96.43% | 100% | 100% | ✓ |
| DIANet | WRN28-10 | 100% | 96.43% | 100% | 100% | ✓ |
| SRMNet | WRN28-10 | 100% | 96.43% | 100% | 100% | ✓ |
| CBAM | WRN28-10 | 100% | 96.43% | 100% | 100% | ✓ |
| IEBN | WRN28-10 | 100% | 96.43% | 100% | 100% | ✓ |
| SGENet | WRN28-10 | 100% | 96.43% | 100% | 100% | ✓ |

For initialization:

```
https://pytorch.org/docs/master/nn.init.html#nn-init-doc.
```

For dataset:

For other tasks:

```
https://github.com/meetshah1995/pytorch-semse
https://github.com/jwyang/faster-rcnn.pytorch
https://github.com/speedinghzl/pytorch-segmentation-toolbox
https://github.com/foamliu/Deep-Image-Matting-PyTorch
https://github.com/CDOTAD/AlphaGAN-Matting
https://github.com/abhiskk/fast-neural-style
```

Table 11: **Initialization**

| Experiments | Remark | Gaussian | Variance | Mean | Magnitude | in the front of network? |
|---|---|---|---|---|---|---|
| Kaiming-uniform | ResNet164 | 98.77% | 97.55% | 100% | 100% | ✓ |
| Kaiming-normal | ResNet164 | 98.77% | 97.53% | 100% | 97.55% | ✓ |
| Xavier-normal | ResNet164 | 98.77% | 96.32% | 100% | 97.55% | ✓ |
| Xarier-uniform | ResNet164 | 98.16% | 96.32% | 100% | 99.39% | ✓ |
| Orthogonal | ResNet164 | 97.55% | 96.32% | 100% | 100% | ✓ |
| Kaiming-uniform | VGG16 | 100% | 93.75% | 100% | 100% | ✓ |
| Kaiming-normal | VGG16 | 100% | 93.75% | 100% | 100% | ✓ |
| Xavier-normal | VGG16 | 100% | 93.75% | 100% | 93.75% | ✓ |
| Xarier-uniform | VGG16 | 100% | 93.75% | 100% | 93.75% | ✓ |
| Orthogonal | VGG16 | 100% | 93.75% | 93.75% | 93.75% | ✓ |
| Kaiming-uniform | WRN28-10 | 100% | 96.43% | 100% | 100% | ✓ |
| Kaiming-normal | WRN28-10 | 100% | 100% | 100% | 100% | ✓ |
| Xavier-normal | WRN28-10 | 100% | 96.43% | 100% | 100% | ✓ |
| Xarier-uniform | WRN28-10 | 100% | 96.43% | 100% | 100% | ✓ |
| Orthogonal | WRN28-10 | 100% | 96.43% | 100% | 100% | ✓ |
| Kaiming-uniform | PreResNet110 | 100% | 99.08% | 100% | 100% | ✓ |
| Kaiming-normal | PreResNet110 | 100% | 99.08% | 100% | 100% | ✓ |
| Xavier-normal | PreResNet110 | 100% | 100% | 100% | 100% | ✓ |
| Xarier-uniform | PreResNet110 | 100% | 99.08% | 100% | 100% | ✓ |
| Orthogonal | PreResNet110 | 100% | 100% | 100% | 100% | ✓ |
| Kaiming-uniform | AlexNet | 100% | 100% | 100% | 100% | ✓ |
| Kaiming-normal | AlexNet | 100% | 100% | 100% | 100% | ✓ |
| Xavier-normal | AlexNet | 100% | 100% | 100% | 100% | ✓ |
| Xarier-uniform | AlexNet | 100% | 100% | 100% | 100% | ✓ |
| Orthogonal | AlexNet | 100% | 100% | 100% | 100% | ✓ |
| Kaiming-uniform | DenseNet-BC-100-12 | 100% | 98.99% | 100% | 98.99% | ✓ |
| Kaiming-normal | DenseNet-BC-100-12 | 100% | 98.99% | 100% | 98.99% | ✓ |
| Xavier-normal | DenseNet-BC-100-12 | 100% | 98.99% | 100% | 98.99% | ✓ |
| Xarier-uniform | DenseNet-BC-100-12 | 98.99% | 98.99% | 98.99% | 98.99% | ✓ |
| Orthogonal | DenseNet-BC-100-12 | 100% | 98.99% | 100% | 98.99% | ✓ |

Table 12: **Dataset**

| Experiments | Remark | Gaussian | Variance | Mean | Magnitude | in the front of network? |
|---|---|---|---|---|---|---|
| CIFAR10 | WRN28-10 | 100% | 96.43% | 100% | 100% | ✓ |
| CIFAR100 | WRN28-10 | 100% | 100% | 100% | 100% | ✓ |
| ImageNet | WRN28-10 | 100% | 96.43% | 100% | 100% | ✓ |
| MINIST | WRN28-10 | 100% | 96.43% | 100% | 96% | ✓ |

`https://github.com/csinva/gan-pretrained-pytorch`

Table 13: **Other tasks**

| Experiments | Remark | Gaussian | Variance | Mean | Magnitude | in the front of network? |
|---|---|---|---|---|---|---|
| SgeNet(Cityscapes) | Segmentation | 100% | 100% | 100% | 100% | ✓ |
| PSPNet(Cityscapes) | Segmentation | 100% | 99.12% | 100% | 99.12% | ✓ |
| ResNet101(COCO) | Faster RCNN | 100% | 99.05% | 100% | 100% | ✗ |
| ResNet101(VOC2007) | Faster RCNN | 100% | 99.05% | 100% | 100% | ✗ |
| VGG16(Visual Genome) | Faster RCNN | 100% | 93.75% | 100% | 100% | ✓ |
| AlphaGAN | Image matting | 100% | 95.00% | 100% | 95.00% | ✓ |
| Deep image matting | Image matting | 100% | 100% | 100% | 100% | ✓ |
| Fast neural style | candy | 86.67% | 100% | 100% | 100% | ✗ |
| Fast neural style | mosaic | 93.33% | 100% | 100% | 100% | ✓ |
| Fast neural style | starry night | 86.67% | 100% | 100% | 100% | ✗ |
| Fast neural style | udnie | 66.67% | 100% | 100% | 100% | ✗ |
| DCGAN(MNIST) | GAN | 100% | 100% | 100% | 100% | ✓ |
| DCGAN(CIFAR10) | GAN | 100% | 100% | 100% | 100% | ✓ |
| DCGAN(CIFAR100) | GAN | 100% | 100% | 100% | 100% | ✓ |
| VGG19(CIFAR10) | without BN | 100% | 100% | 100% | 100% | ✓ |
| VGG19(CIFAR10) | with BN | 93.75% | 100% | 100% | 100% | ✓ |
| VGG19(CIFAR10-lr) | schedule(82-164) | 93.75% | 100% | 100% | 100% | ✓ |
| VGG19(CIFAR10-lr) | schedule(60-120) | 93.75% | 100% | 100% | 100% | ✓ |
| VGG19(CIFAR10-lr) | coslr | 93.75% | 100% | 100% | 100% | ✓ |

For pytorch pretrain:`http://pytorch.org/docs/master/torchvision/index.html`.

Table 14: **Pytorch pretrian**

| Experiments | Remark | Gaussian | Variance | Mean | Magnitude | in the front of network? |
|---|---|---|---|---|---|---|
| VGG11 | ImageNet | 100% | 75.00% | 100% | 75.00% | ✓ |
| VGG16 | ImageNet | 100% | 84.62% | 100% | 100% | ✓ |
| VGG19 | ImageNet | 100% | 87.50% | 100% | 100% | ✓ |
| ResNet18 | ImageNet | 100% | 88.24% | 100% | 100% | ✓ |
| ResNet34 | ImageNet | 100% | 88.24% | 100% | 96.97% | ✓ |
| ResNet50 | ImageNet | 100% | 83.67% | 100% | 100% | ✗ |

## Q    Training through slimming

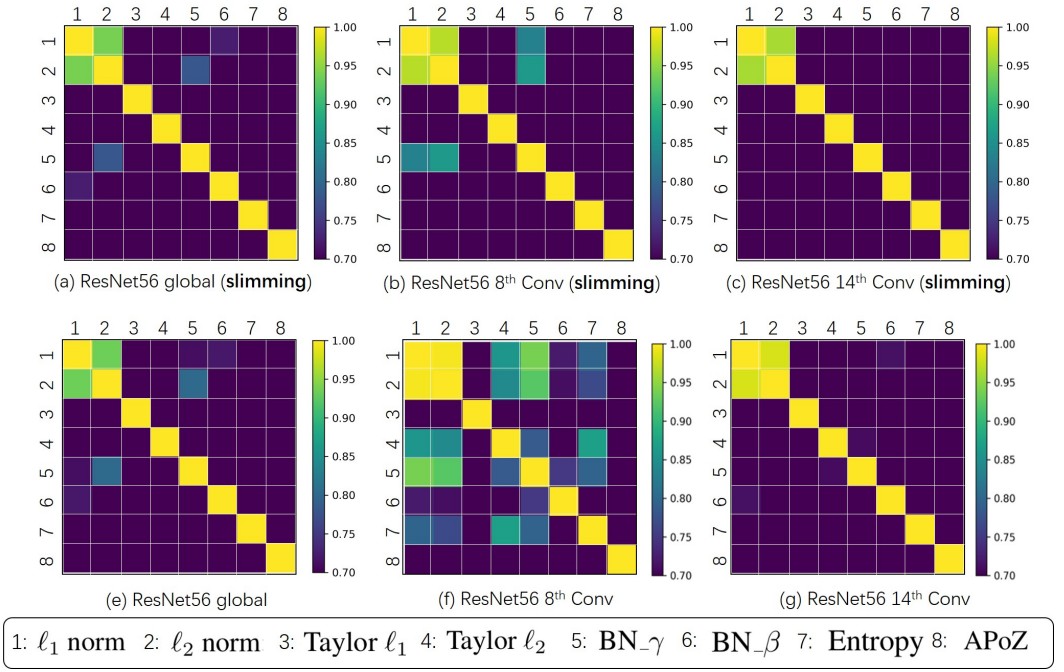

Figure 19: The Similarity for different criteria with/without slimming [34].

As a representative of the BN-based pruning method, slimming pruning[34] can not be directly compared with the criteria mentioned in the paper because it adopts a special training method. Therefore, we use the training method in [34] to train another ResNet56 on cifar100. Then, the analysis of similarities between 8 different pruning criteria on such a model is shown in Fig. 19.

In this situation, the fifth criterion $BN\_\gamma$ is the method introduced in [34]. From Fig. 19, there is no significant difference in the result of the similarity between ResNet56 obtained by slimming method and resnet56 trained in general.

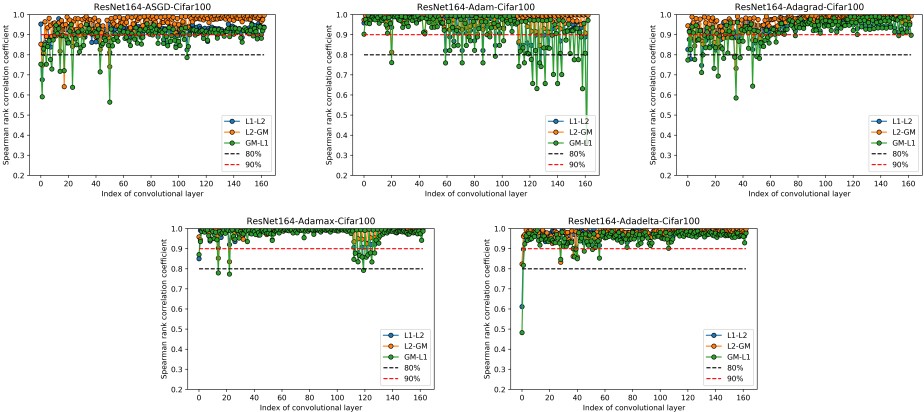

Figure 21: Optimizer

# R  More experiments of Sp in Norm-based criteria

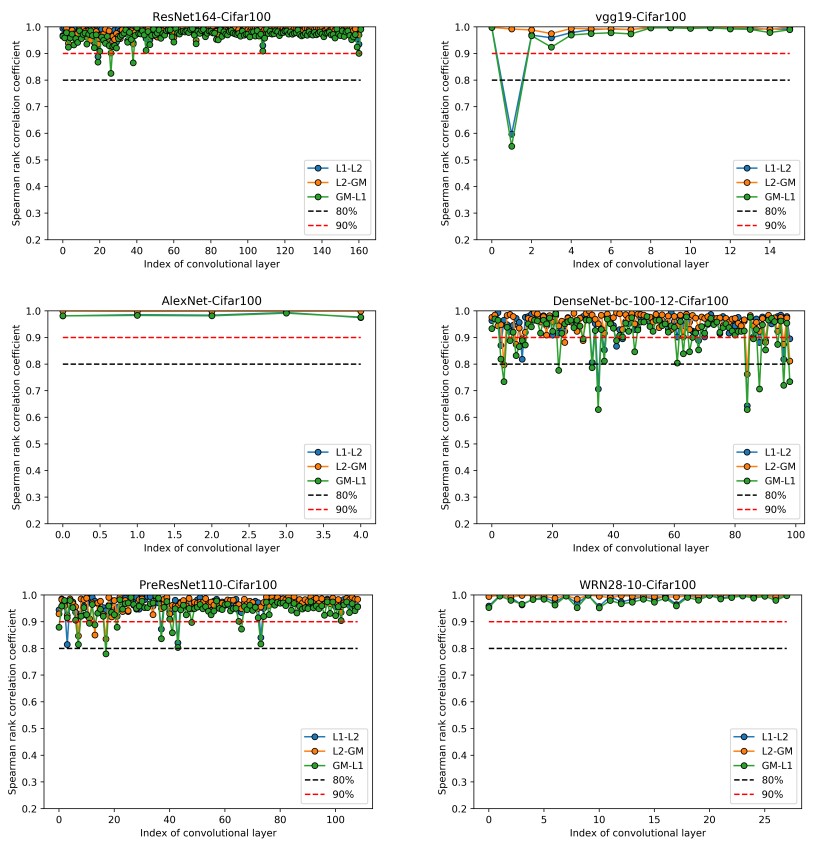

Figure 20: Network Structure

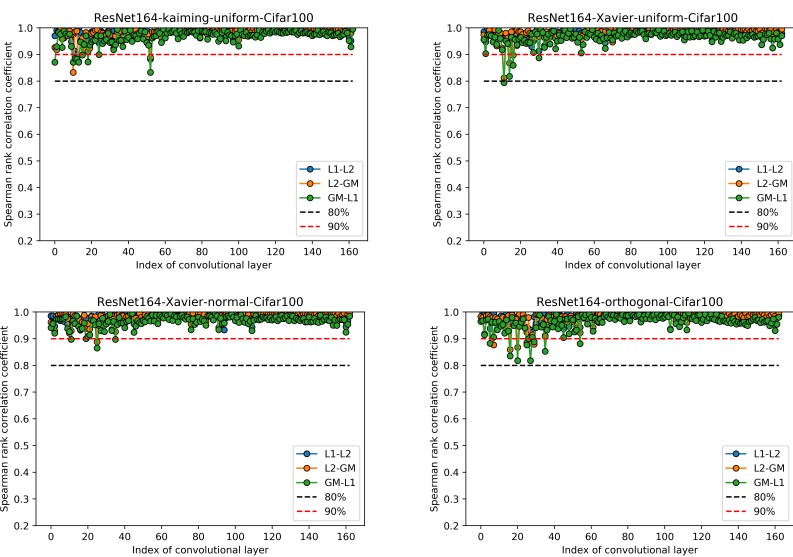

Figure 22: Initialization

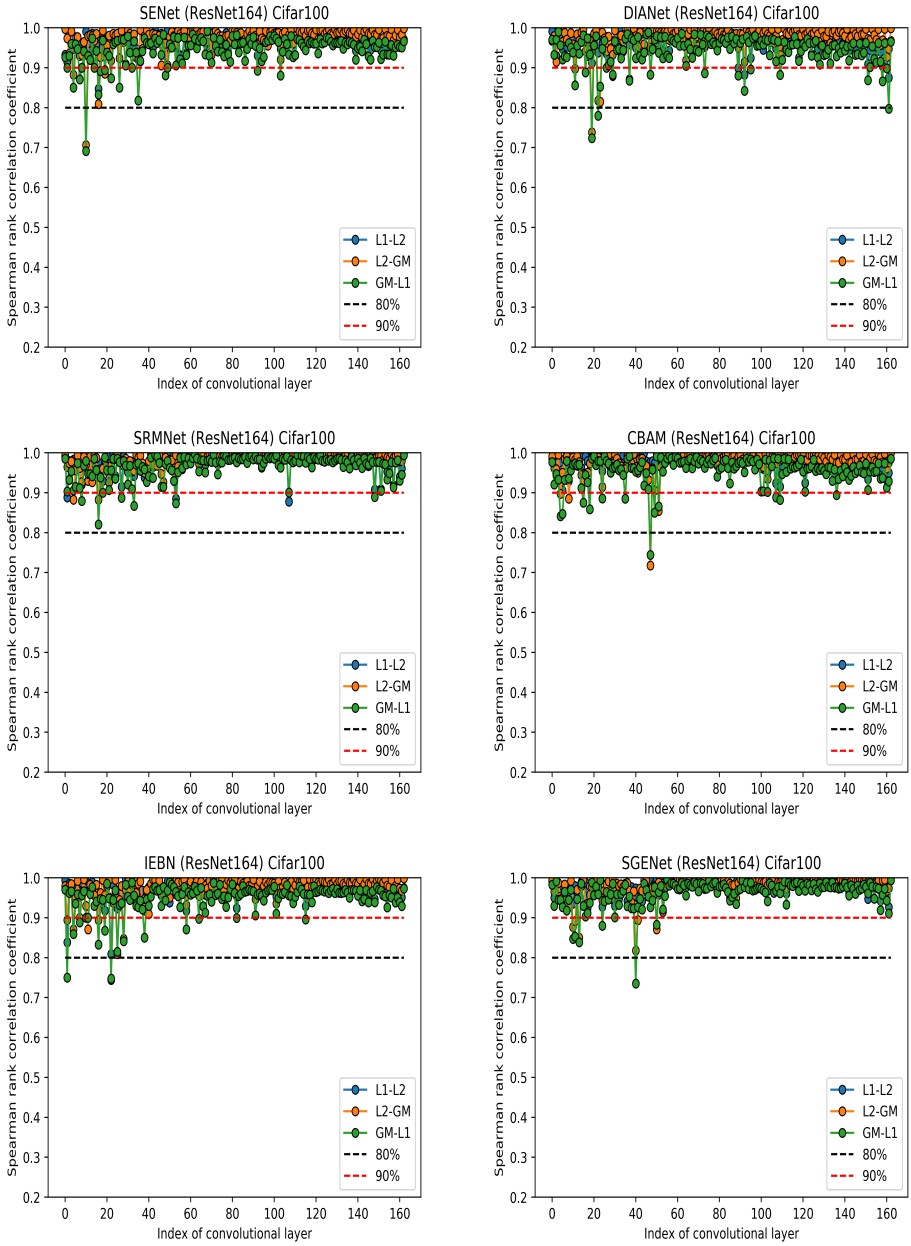

Figure 23: Attention mechanism

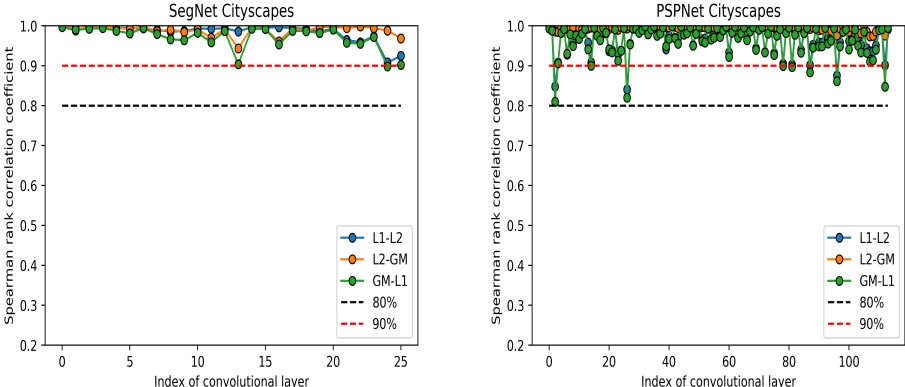

Figure 24: Other task: segmentation

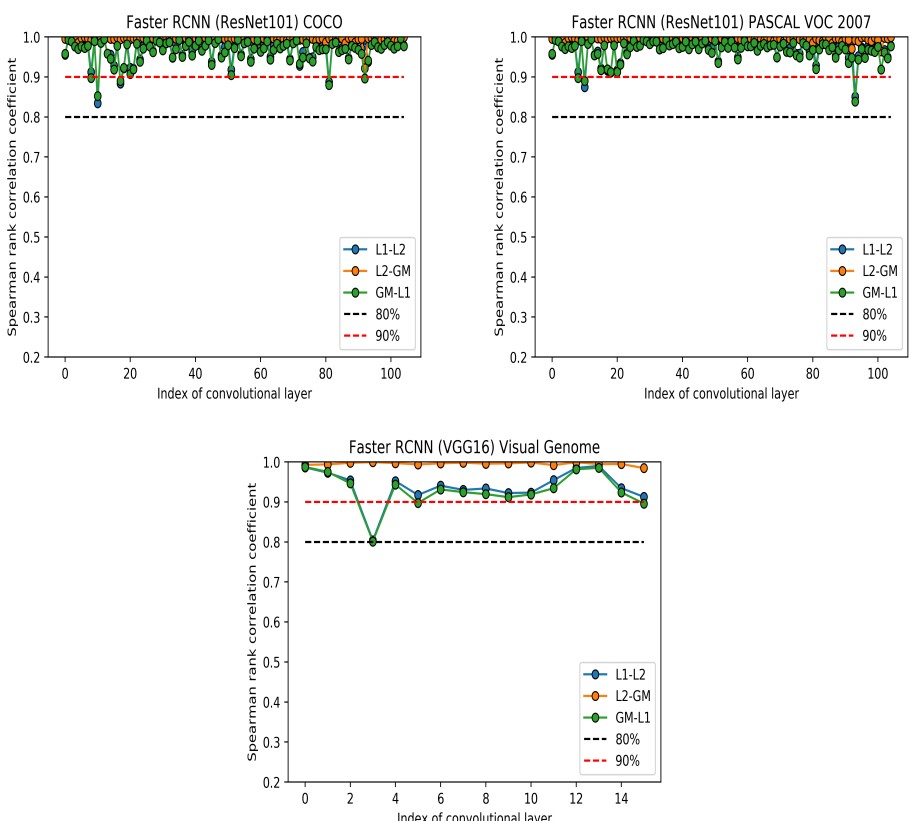

Figure 25: Other task: Faster RCNN

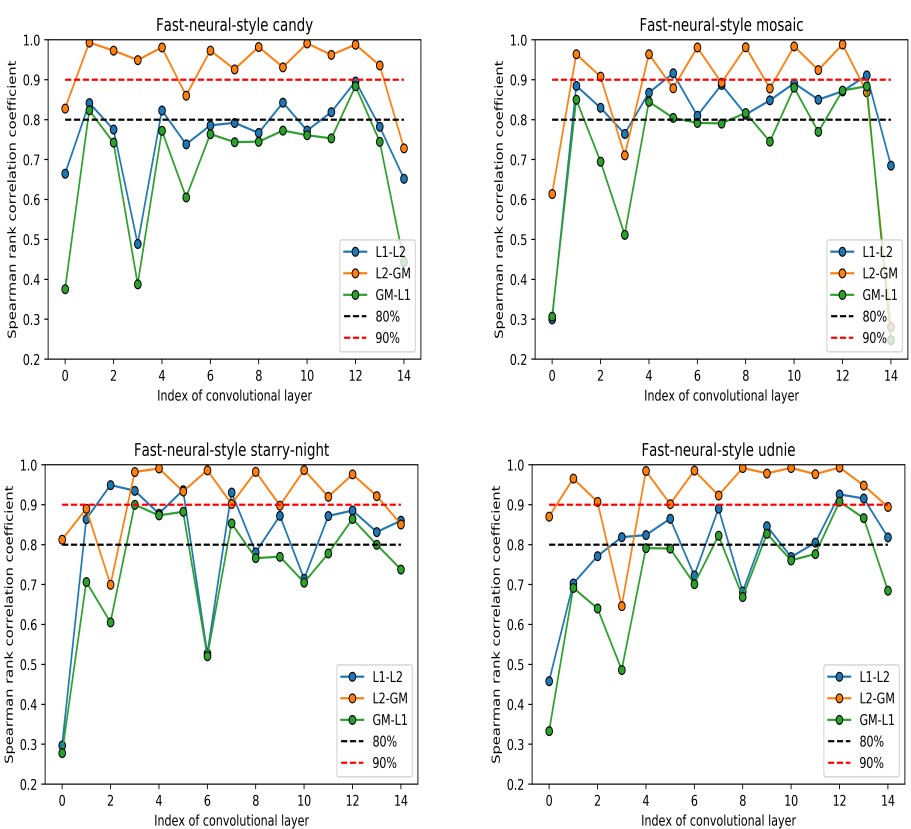

Figure 26: Other task: style transfer

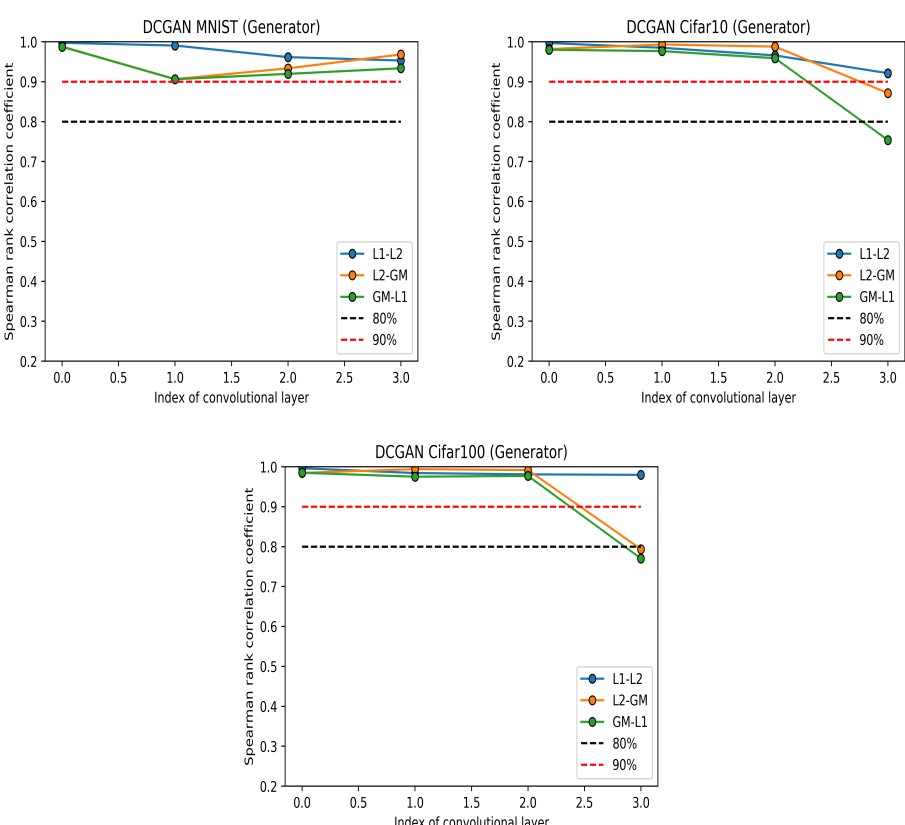

Figure 27: Other task: GAN

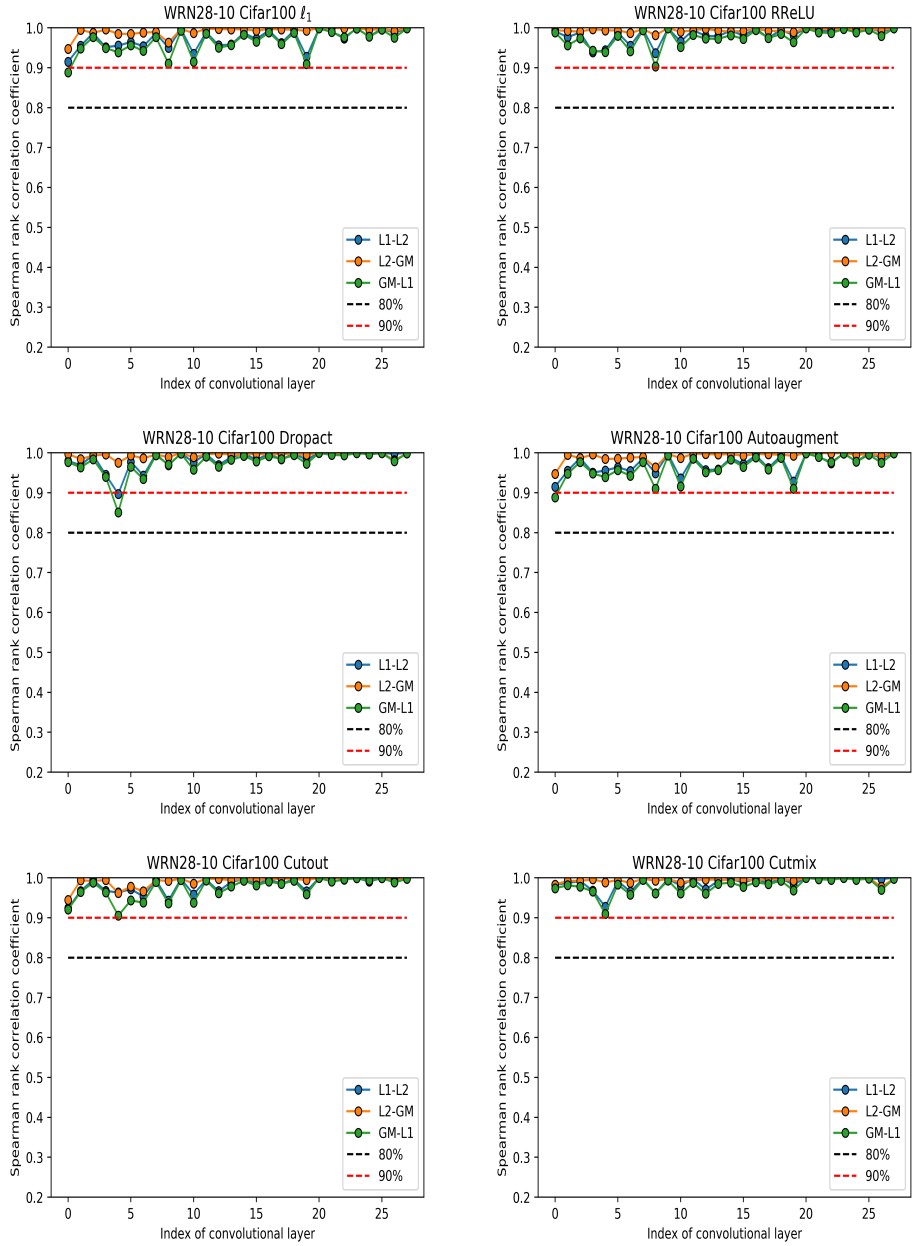

Figure 28: Other task: Regularization

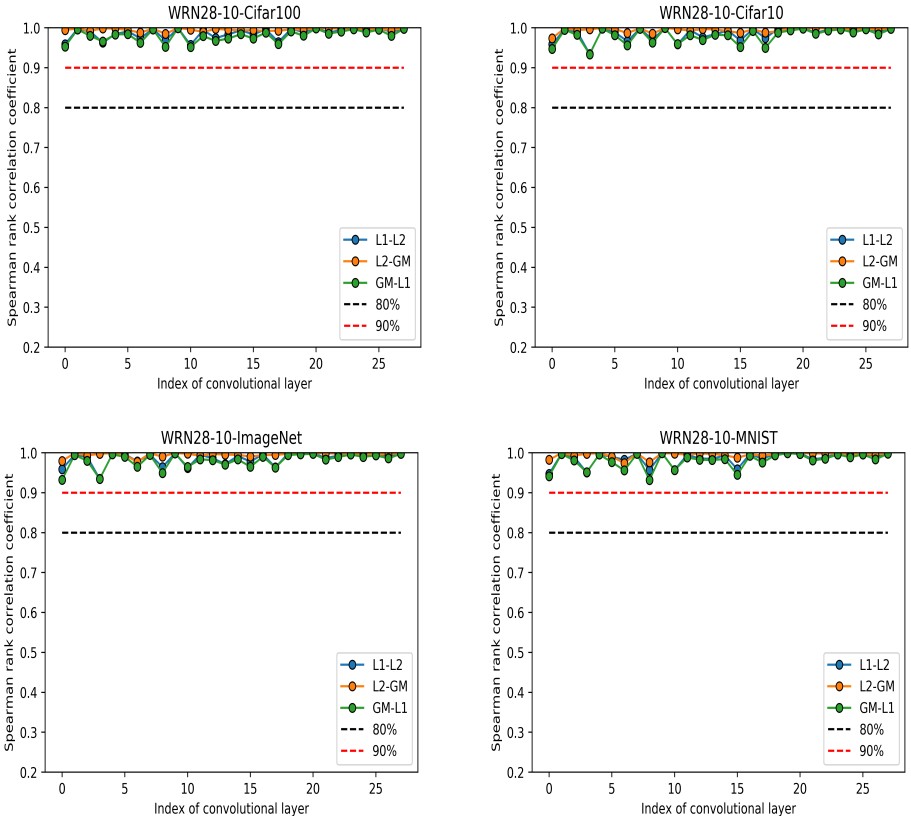

Figure 29: Dataset

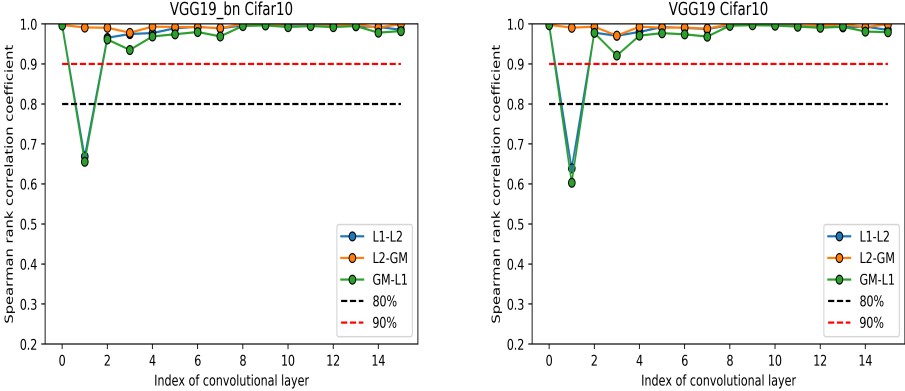

Figure 30: Batch normalization

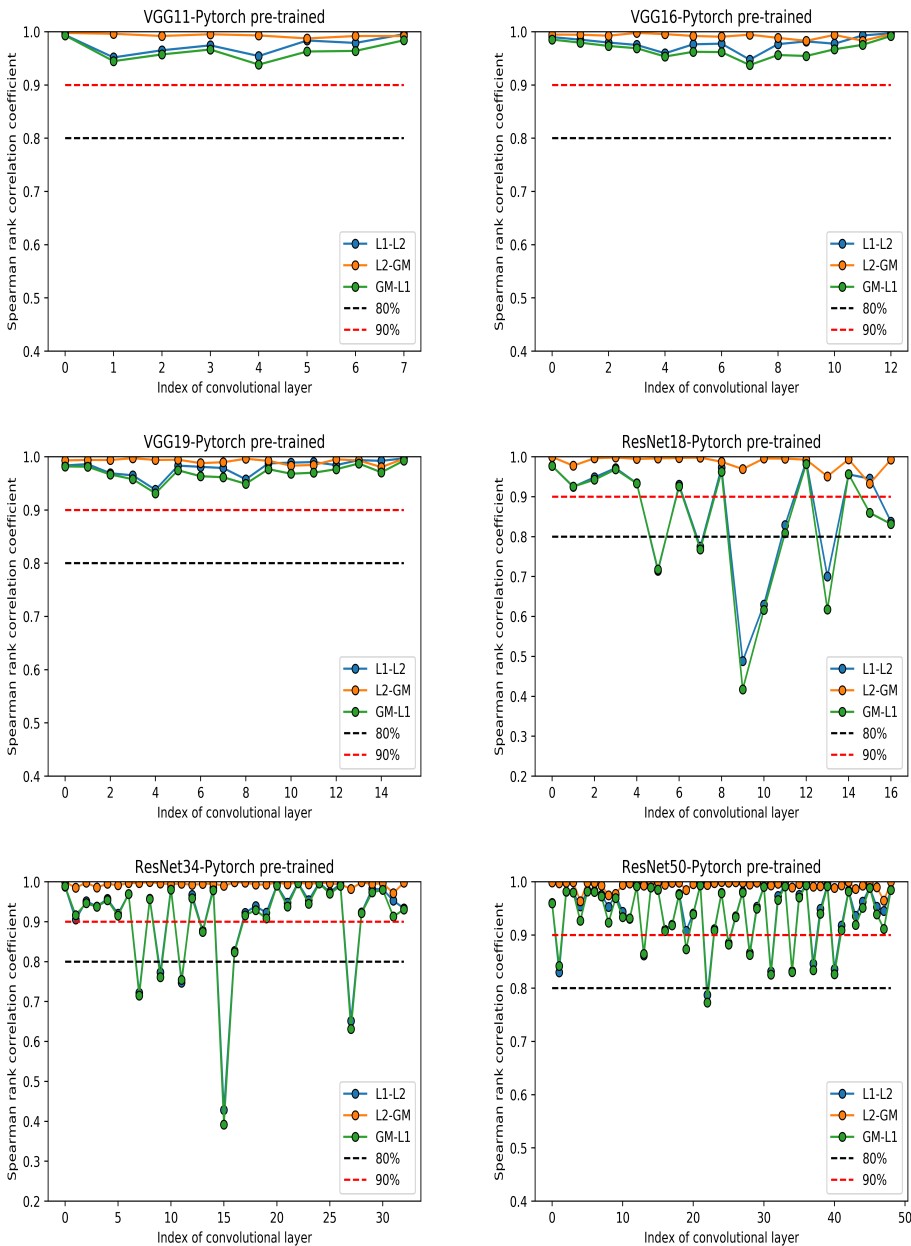

Figure 31: Pytorch pre-trained Model

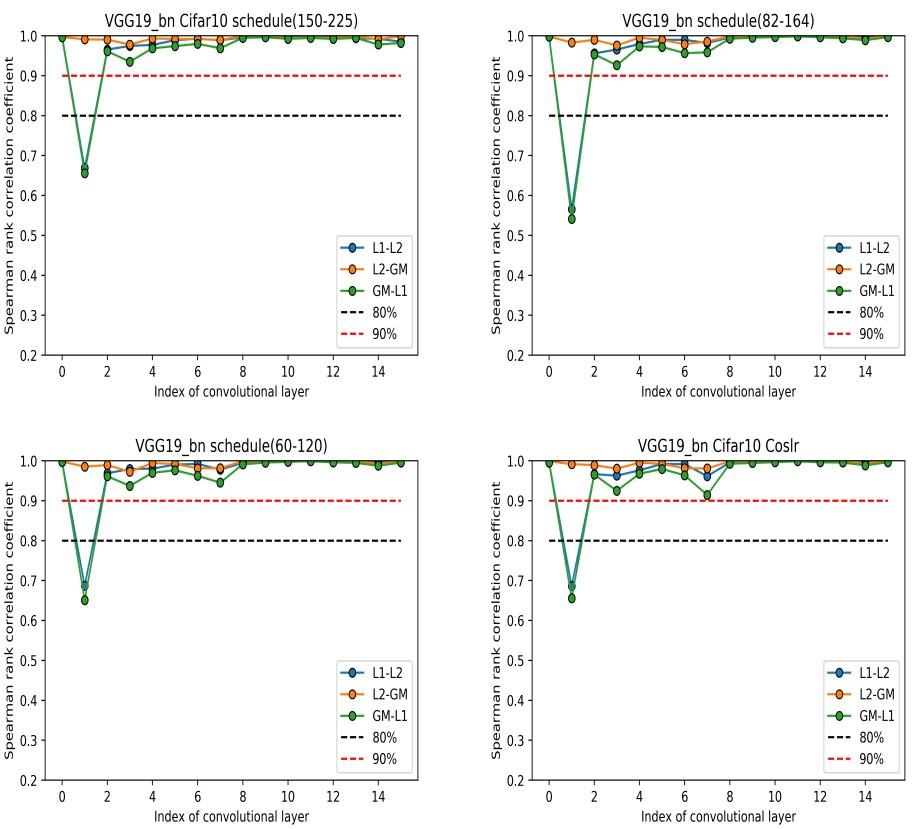

Figure 32: Learning rate