# OpenReview forum: "Rethinking the Pruning Criteria for Convolutional Neural Network"
_NeurIPS.cc/2021/Conference — NeurIPS 2021 Poster_

### Official Review · Reviewer_EQps · 2021-07-07

**Rating:** 4
**Confidence:** 5

**Summary:**

This paper studies 1) the similarity between different pruning criteria for channel pruning, 2) the applicability of different pruning criteria, where applicability refers to the confidence in the importance ranking of filters.  The authors empirically show that for some common pruning strategies such as l1 and l2 pruning and other norm-based pruning, the importance ranking of the filters is almost identical. Moreover, they claim with theoretical and experimental support that the importance score of different filters is very close to each other under certain conditions.  The main contribution of this paper is to analyze these two phenomena for better pruning criteria in the future.

**Limitations And Societal Impact:**

The authors discuss some limitations in Section 6. I suggest the authors discuss also their iid assumption and explain why they had to make that assumption although it is unrealistic. The authors do mention how their findings would impact the research community. They do not address the potential negative societal impact of their work.

**Main Review:**

I like the broad topic of the paper -- understanding the pruning criteria beter. However, the current version of the paper leaves many questions unanswered and needs substantial improvement. I list my concerns below:

- Writing: I think the writing can be improved significantly in terms of clarity, flow, and grammar. Here are some of the easy-to-fix typos I noticed:
    - line 62: unnecessary column at the end of the sentence.
    - line 93: 'Taking a convolutional layers'.
    - line 101: missing space after the comma.
    - line 145: missing space after the comma.
    - line 146: unclear sentence?
    - line 151: consider about.
    - line 162: n-dimensional.
    - line 191: 'Since Proposition 1,'.
    - line 194: 'closed to 0'.
    - line 208: 'in more types of pruning criteria'.

- Overall, I am not convinced with the importance of the problem at hand. It is not surprising that norm-based pruning criteria give almost the same ranking (Similarity). It is also not surprising that for some layers, the importance scores are very similar (Applicability). It would be interesting to compare the effect of pruning filters with similar importance scores on the performance. Perhaps they all perform well. In that case, the pruning criterion is reliable. If some of them do not perform well, then the pruning criterion does not produce a reliable ranking of filters' importance.  However, no such result is provided. Another question to ask is whether very similar importance scores relate to the incompressibility of that layer. Maybe the conclusion we should get from this observation is if importance scores are very similar, it is better not to prune any filters. However, none of these interesting points are discussed in the paper. So, I am having difficulty in understanding the authors' goal in analyzing 'similarity' and 'applicability' phenomena. Can the authors provide a stronger motivation for their study?

- I am confused about the setup in CWDA (lines 84-98). In line 85, $F_{ij}$ are assumed to be iid. However, in line 96, it is said that parameters in the same channel of $F_{ij}$ are observed to be linearly correlated. According to this observation, iid assumption is unrealistic. Did the authors make this iid assumption to simplify the problem?

- I also have concerns about the use of expectation and variance operators. As far as I understand, $\mathbb{E}[X]$ refers to the average of the importance scores in one layer. Do the authors think of the importance scores as different realizations of the same random variable X? If so, 1) why is $X=(x_1,...,x_k)$ defined as a deterministic sequence? 2) the authors should make it clear that they are using an empirical approximation to $\mathbb{E}[X]$. I feel like the authors only wanted to find the average of $X$ and misused expectation operator. Can the authors clarify their notation here?

- There is a recent paper on layer-wise sparsity [1], where a new pruning criterion and strategy are proposed to avoid layer collapse.  The authors may want to have a discussion on that in the paper.

- Minor: I would avoid abbreviations like 'can't' in an academic paper.

[1] Lee, Jaeho, et al. "Layer-adaptive Sparsity for the Magnitude-based Pruning." International Conference on Learning Representations. 2020.

**Time Spent Reviewing:**

6 hours

---

> ### Author Response · Authors · 2021-08-10
> **Response to Reviewer EQps (part 1):**
>
> Thank you for your detailed and meaningful review!
>
>
> **Q1: The questions about writing and typo.**
>
> A1: Thanks for your suggestions, we will update them in revision.
>
> .
>
> .
>
> **Q2: It would be interesting to compare the effect of pruning filters with similar importance scores on the performance.**
>
> A2: Similar to the setting in Fig 5, we can explore the effect of pruning filters with similar importance scores on the performance. First, we find that the criteria ($\ell_2$,Taylor $\ell_1$, Taylor $\ell_2$, BN $\gamma$ and BN$_{\beta}$) for VGGNet can cause the applicability problem in most layers (Fig.5). As such, we randomly select 10% or 20% filters to be pruned by the uniform distribution $U[0,1]$ in each layer, and the selective filters will be in similar importance scores. Finally, we finetune the pruned model (there are 20 random repeated experiments). $\Delta$ denotes the difference between max acc. and min acc. (i.e. max acc. - min acc.) . Since their importance scores are very similar, when the network is pruned and finetuned, it can be expected that the performance should be similar in these repeated experiments.
>
>
> VGG16 on CIFAR100 (Unpruned baseline acc: 72.99):
>
> | criterion      | min acc. (ratio = 10%) | max acc. (ratio = 10%) | mean acc. (ratio = 10%) | $\Delta$ (ratio = 10%) | min acc. (ratio = 20%) | max acc. (ratio = 20%) | mean acc. (ratio = 20%) | $\Delta$ (ratio = 20%) |
> |----------------|------------------------|------------------------|-------------------------|------------------------|------------------------|------------------------|-------------------------|------------------------|
> | $\ell_2$       | 71.41                  | 72.65                  | 71.75                   | **1.24**               | 71.01                  | 72.47                  | 71.32                   | **1.46**               |
> | Taylor$\ell_1$ | 71.67                  | 72.34                  | 71.89                   | **0.67**               | 71.32                  | 72.32                  | 71.45                   | **1.00**               |
> | Taylor$\ell_2$ | 71.87                  | 72.37                  | 71.91                   | **0.50**               | 71.66                  | 72.27                  | 71.65                   | **0.61**               |
> | BN$_\gamma$    | 71.09                  | 71.66                  | 71.36                   | **0.57**               | 71.02                  | 71.57                  | 71.12                   | **0.55**               |
> | BN$_\beta$     | 71.15                  | 72.58                  | 71.43                   | **1.43**               | 71.06                  | 72.11                  | 71.87                   | **1.05**               |
>
>
> VGG19 on CIFAR100 (Unpruned baseline acc: 73.42):
>
> | criterion      | min acc. (ratio = 10%) | max acc. (ratio = 10%) | mean acc. (ratio = 10%) | $\Delta$ (ratio = 10%) | min acc. (ratio = 20%) | max acc. (ratio = 20%) | mean acc. (ratio = 20%) | $\Delta$ (ratio = 20%) |
> |----------------|------------------------|------------------------|-------------------------|------------------------|------------------------|------------------------|-------------------------|------------------------|
> | $\ell_2$       | 71.99                  | 73.15                  | 72.26                   | **1.16**               | 71.11                  | 73.02                  | 72.15                   | **1.91**               |
> | Taylor$\ell_1$ | 71.67                  | 73.04                  | 72.23                   | **1.37**               | 71.60                  | 72.98                  | 72.24                   | **1.38**               |
> | Taylor$\ell_2$ | 72.12                  | 72.99                  | 72.28                   | **0.87**               | 72.04                  | 72.83                  | 72.54                   | **0.79**               |
> | BN$_\gamma$    | 72.01                  | 73.23                  | 72.25                   | **1.22**               | 71.98                  | 72.32                  | 72.12                   | **0.34**               |
> | BN$_\beta$     | 72.25                  | 73.23                  | 72.40                   | **0.98**               | 72.04                  | 72.65                  | 72.33                   | **0.61**               |
>
>
> However, from the results in the above table, although the score of the pruned filters is very close, we can still get pruning results with very different results (e.g. the $\Delta$ of VGG16 on $\ell_ 2 $ are more than 1). It means that these criteria may not really represent the importance of convolutional filters.
>
>
> In fact, we have questioned (line 219-223) whether these criteria can really estimate the importance of convolutional filters:
> - (i) In Fig.5, the Sp between most different pruning criteria are not large in these layers (in fact, most of them are less than 0.2), which indicates that these criteria have great differences in the redundancy measurement of convolutional filters. This may lead to a phenomenon that one criterion considers a convolutional filter to be important, while another considers it to be redundant.
> - (ii) We have taken an example (line 222-223 and Appendix J) to show that there are two criteria, and their measurement of "redundancy" is very different. That is, for some convolutional filters, one criterion considers that to be the most redundant and the other considers it to be the least redundant.
>
>
> .
>
> .
>
> .
>
>
> **Q3: Another question to ask is whether very similar importance scores relate to the incompressibility of that layer.**
>
> A3:The key premise of studying this incompressibility is that the criterion's estimation of "redundancy" is relatively accurate. From the results of experiments in Q2 / A2 and (line 222-223 and appendix J), the estimation of importance in these criteria may be inaccurate. Therefore, we do not discuss this incompressibility.
>
> The reason for the inaccurate estimation of these criteria may be that the design of these criteria is mostly intuitive and heuristic. The design of them may not guarantee the good performance of pruning results. This is different from another network compression method, i.e. neural network structure search (NAS). NAS methods are generally directly related to loss or accuracy, so the compressed models they obtain often have better performance, but the criteria designed by intuition can not guarantee good performance.

---

> ### Author Response · Authors · 2021-08-10
> **Response to Reviewer EQps (part 2):**
>
> **Q4:So, I am having  difficulty in understanding the authors' goal in analyzing 'similarity'  and 'applicability' phenomena. Can the authors provide a stronger  motivation for their study?**
>
> A4:'similarity'  and 'applicability' can help us re-understand these pruning criteria and rethink what the ideal pruning criteria should be:
>
> - (1) For similarity, different types of criteria may have very strong similarity in some layers (Fig. 5), and such strong similarity is NOT an **accident** (refer to Reviewer 6Th5 Q1/A1(1) ). This phenomenon may bring inspiration to the community, e.g. it can be used to blend different pruning criteria in different layers to construct better pruning criteria [1][2];
> - (2) For the applicability problem, we find that the important scores of convolutioal filters are very similar under some criteria. This naturally leads to the question of whether the pruning criteria are reasonable (line 172-177), i.e., whether these importance criteria can represent the importance of convolutional filters.
> - (3) From the perspective of similarity and applicability, it can be considered that the ideal pruning criterion should have:
>   + a. The estimation of the redundancy of convolutional filters should be relatively accurate;
>   + b. On the basis that the estimation of redundancy is correct, the ideal criterion should be distinguishable (unless there is incompressibility);
>   + c. The similarity among the accurate criteria (satisfy (3)a. and (3)b.) should be appropriate:
>     - i. Their similarity should not be too small (which means that there may be a contradiction like appendix J. If the similarity is very low, it indicates that at least one pruning criterion needs to be reviewed again),
>     - ii. Their similarity cannot be too large (Sp cannot be too large, which means that these criteria are the same, as shown in Fig 4, which is generally greater than 0.9)
>
> .
>
> .
>
> **Q5：About the iid assumption.**
>
> A5:This assumption about iid does not have problem:
> - (1) In line 85-86, we show that "$i^ {\rm th} $layer, $F_{ ij}(j = 1,2,...N_{i+1}) $ are iid, i.e. $F_{ i1},F_{ i2},...,F_{ iN_{i+1}} $ are iid"  . In other words, we assume that the filters in one convolutional layer are iid.
> - (2) In line 96-97, for each convolutional filter $F_{ij}$, its own parameters are not completely independent. This means that the matrix $\Sigma_{\mathrm{diag}}+\epsilon \cdot \Sigma_{\text {block }}$ in Eq (1) is not a diagonal matrix. As shown in line87 and Fig 2 (a), it is a block diagonal matrix.
> To conclude, we did not make this iid assumption to simplify the problem and the assumption is realistic.
>
>
>
> .
>
> .
>
> **Q6:Can the authors clarify their notation $\mathbb{E}(X)$**.
>
> A6: Yes, we want to represent the average of X, which can indicate the magnitude of $x_1,x_2,...x_k$ (line 152).
>
> .
>
> .
>
>
> **Q7:There is a recent  paper on layer-wise sparsity [3], where a new pruning criterion and  strategy are proposed to avoid layer collapse.  The authors may want to  have a discussion on that in the paper.**
>
> A7:  Thank you for your recommendation. This may provide inspiration for Section 5 of our paper.
>
> .
>
> .
>
> .
>
> **Q8:The authors do mention how their findings would impact the research community. They do not address the potential negative societal impact of their work.**
>
> A8: Although we have verified CWDA (line115-117) under various settings in table5 (Appendix Q), it still can not completely cover all the training settings of convolutional neural networks. Therefore, if we want to use CWDA under the new setting in the future, we still need to do some verifications. In the future, we can try to prove CWDA theoretically, which may be a better direction.
>
> .
>
> .
>
> [1]    Learning filter pruning criteria for deep convolutional neural networks acceleration.  CVPR2020.
>
> [2]  Blending Pruning Criteria for Convolutional Neural Networks. ICANN2021.
>
> [3] Lee, Jaeho, et al. "Layer-adaptive Sparsity for the Magnitude-based Pruning." International Conference on Learning Representations. 2020.

---

> > ### Comment · Reviewer_EQps · 2021-08-24
> > **An Early Response to the Rebuttal**
> >
> > I would like to thank the authors for the detailed response to my review. I have some additional questions and comments before I finalize my review.
> >
> > **About the iid assumption:** After I read the authors' response and check the paper again, now I see that the paper is actually consistent on using the iid assumption. I think adding a superscript $i$ to the $\Sigma$'s in (1) would make things more clear for the readers.
> >
> > **About expectation and variance operators:** If the authors want to keep the expectation and variance operations in the paper, they need to change the way how they define $X$ (it should be a random variable) and make a clear transition from "the expectation of a random variable" to "an average of samples of a random variable". There is nothing wrong with Monte Carlo sampling. The only problem is the notation is wrong without this transition. And it could be very easily fixed. However, the authors did not acknowledge that the notation is wrong. So, I am assuming they do not plan to fix it in the revised version. Is that right?
> >
> > **Overall Comments:** I would like to thank the authors for the additional experiments. Will those results be added to the revised version? I think the new results support that (at least) some pruning criteria fail to rank the importance scores. Is the take-away message "some of the pruning criteria do not find the importance scores optimally"? Is there another contribution I am missing?
> >
> > About the applicability, I do not think Reviewer 6Th5 Q1/A1(1) addresses my concerns. I mentioned that it is not surprising to see that norm-based pruning criteria output similar importance scores. New comparisons with random pruning are not very relevant to my comment. And I still cannot see: 1) what is the interpretation/conclusion/message here?; 2) what is the contribution here given that this observation is not surprising?
> >
> > Is the goal of the paper to list the properties of an ideal pruning criterion as in the response A4(3)? Or is there another contribution I am missing?

---

> > > ### Author Response · Authors · 2021-08-28
> > > **Response to An Early Response to the Rebuttal of Reviewer EQps**
> > >
> > > Thanks for your meaningful comments.
> > >
> > >
> > > .
> > >
> > > .
> > >
> > >
> > > Q:**About the iid assumption: After I read the authors'  response and check the paper again, now I see that the paper is actually  consistent on using the iid assumption. I think adding a superscript to the 's in (1) would make things more clear for the readers.**
> > >
> > > A:Thanks for your comments, we will add a superscript to the 's in Eq(1) in the revised version.
> > >
> > >
> > > .
> > >
> > > .
> > >
> > > .
> > >
> > >
> > > Q:**About expectation and variance operators: If the authors want to keep the expectation and variance operations in the paper, they need to change the way how they define (it should be a random variable) and make a clear transition from "the  expectation of a random variable" to "an average of samples of a random  variable". There is nothing wrong with Monte Carlo sampling. The only  problem is the notation is wrong without this transition. And it could  be very easily fixed. However, the authors did not acknowledge that the  notation is wrong. So, I am assuming they do not plan to fix it in the  revised version. Is that right?**
> > >
> > > A:Our notation is indeed wrong, we will fix it in the revised version, including the main paper and appendix.
> > >
> > > .
> > >
> > > .
> > >
> > > .
> > >
> > > **About the overall comments**
> > >
> > > **Overall Comments Q1: I would like to thank the authors  for the additional experiments. Will those results be added to the  revised version?**
> > >
> > > Overall Comments A1: Yes, all those results will be added to the revised version.
> > >
> > >
> > > .
> > >
> > > .
> > >
> > > .
> > >
> > > **Overall Comments Q2: Is the take-away  message "some of the pruning criteria do not find the importance scores optimally"?Is there another contribution I am missing?**
> > >
> > > Overall Comments A2:  Yes, "some of the pruning criteria do not find the importance scores optimally" is our take-away message. Moreover, these pruning criteria are well-known and widely used.
> > >
> > > .
> > >
> > > .
> > >
> > > .
> > > **Overall Comments Q3: About the applicability, I do not think Reviewer 6Th5 Q1/A1(1)  addresses my concerns.New comparisons with random pruning are not very relevant to my comment**
> > >
> > >
> > > Overall Comments A3:
> > > - (1) "Refer to reviewer 6th5 Q1/A1 (1)" mentioned in EQps (part 2) Q4/A4, i.e. the new comparisons with random pruning, is used to respond to similarity problems, **NOT** applicability. The reasons why the similarity problem can be surprising is:
> > > 	- a. This experiment reveals that the similarity between the criteria is not an accident, because the probability of high similarity is extremely low.
> > > 	- b. For example, as shown in our paper, norm-based criteria on layer-wise pruning have stable high similarity and some seemingly unrelated criteria are strong similar (SP > 0.95) in some layers in Fig 5
> > > - (2) About the applicability, in "response to reviewer EQps (Part 1)", we use EQps (Part 1) Q2/A2 and EQps (Part 1) Q3/A3 to answer the questions about applicability in your comment, that is:
> > > 	- a. **Q2/A2** for :  "It would be interesting to compare the effect of pruning filters with similar importance scores on the performance"
> > > 	- b. **Q3/A3** for : "Another question to ask is whether very similar importance scores relate to the incompressibility of that layer. "
> > >
> > > In Overall Comment A4, we will further summary these experiments and their results.
> > >
> > > .
> > >
> > > .
> > >
> > > .
> > >
> > > **Overall Comments Q4: And  I still cannot see: 1) what is the interpretation/conclusion/message  here?;**
> > >
> > > Overall Comments A4: We summarize the interpretation/conclusion/message on applicability as follow:
> > > - (1) In short, the conclusions of Q2/A2 and Q3/A3 on applicability in Overall Comments A3 (2) are as follows:
> > > 	- a. Some widely used pruning criteria cannot estimate the importance scores accurately;
> > > 	- b. Some neural networks do not have incompressibility under some given settings；
> > > 		- i. In EQps (Part 1) Q3/A3, we explained why we do not discuss incompressibility. In fact, from the view of pruning results (i.e. acc. after pruning and finetuning); In the EQps (Part 1) Q2/A2 experiments, we can see that VGG does not have incompressibility under the given setting. Specifically, in the experiments of EQps (Part 1) Q2/A2, even if the importance scores are very similar, there exist at least one pruned network (i.e. the pruned network which can have max acc.), whose acc. can be closed to the performance of the Unpruned baseline after finetuning. It reveals that these networks can be further compressed.
> > > - (2) Should we be surprised by applicability? If the scores of only a few layers are very close under one criterion, this phenomenon is indeed not surprising. However, as shown in Fig 5, **what we should be surprised is that**, taking VGG as an example, there are applicability problems in **many** layers on **different** types of pruning criteria . In fact, applicability problems are common, as shown in Fig1 and Fig5, and most of these pruning criteria are well-known and widely used. Therefore, if researchers can consider the applicability in the future, it will be possible to design better and more reasonable pruning criteria.
> > > - (3) From an experimental point of view, some neural networks (such as the VGGNet in Q2/A2  and Q3/A3 experiments) are compressible, that is, there are redundant convolutional filters in these networks. the scores of these filters should be small.  Therefore, the accurate importance scores should not be so close (i.e. there should be no applicability problem, the pruning criteria should be able to distinguish redundant and non-redundant filters)
> > >
> > > In conclusion, through the study of applicability, we can know:
> > >   - (a) “Do not have applicability problem" is a necessary condition for the ideal pruning criteria.
> > >   - (b) Some widely used pruning criteria cannot estimate the importance scores accurately;
> > >   - (c) If researchers can consider the applicability in the future, it will be possible to design better and more reasonable pruning criteria.
> > >
> > > Therefore, the applicability problem we provided is surprising and worth studying.
> > >
> > > .
> > >
> > > .
> > >
> > > .
> > >
> > > **Overall Comments Q5: 2) what is the contribution here given that this observation is  not surprising?**
> > >
> > >
> > > Overall Comments A5: Combined with Overall Comments A4, we organize the contributions here given that this observation seems not surprising:
> > > - (1) According to Overall Comments Q4(4), from the applicability problem, we can know that：
> > > 	- a. “Do not have applicability problem" is a necessary condition for the ideal pruning criteria;
> > > 	- b. Some widely used pruning criteria cannot estimate the importance scores accurately;
> > > 	- c. If researchers can consider the applicability in the future, it will be possible to design better and more reasonable pruning criteria.
> > > - (2) According to Overall Comments Q4(2), we can know that： what we should be surprised is that, taking VGG as an example, there are applicability problems in many layers even when we are using different types of pruning criteria.
> > > - (3) Through the analysis of applicability and similarity, this paper can provide the requirements of the ideal criterion, that is EQps (part 2) A4 (3).
> > > - (4) From the perspective of originality, the previous works rarely studied the quality of pruning criteria from the perspective of this paper (similarity and applicability), which are blind spots in the field of network pruning. As shown in line 45-48, the main concerns in the field of network pruning usually are:
> > >   - (a) How much the model was compressed;
> > >   - (b) How much performance was restored;
> > >   - (c) The inference efficiency of the pruned network；
> > >   - (d) The cost of finding the pruned network.
> > >
> > >
> > > We hope the community can pay attention to these blind spots to design better pruning criteria by EQps (part 2) A4 (3) .
> > >
> > >
> > > .
> > >
> > > .
> > >
> > > .
> > >
> > > **Overall Comments Q6: Is the goal of the paper to list the properties of an ideal pruning  criterion as in the response A4(3)? Or is there another contribution I  am missing?**
> > >
> > >
> > > Overall Comments A6:  Yes, as shown in our paper (line 75-76), our main contribution is to introspect what the ideal pruning criterion should be like from the perspective of similarity and applicability,  i.e., A4 (3). They can influence future criteria development and guide the researchers to design more reasonable criteria. (Refer to reviewer 6Th5 and reviewer nJ3h)
> > >
> > > In addition, there are other contributions worth paying attention to：
> > > - (1) CWDA has successfully explained the different phenomena in many similarity and applicability problems in theory. Since CWDA generally holds for convolutional neural networks (according to the statistical tests in our paper), it can be used in other fields related to convolutional neural networks and provide analytical tools for the researchers;
> > > - (2) We also break some stereotypes, such as that the results of $\ell_1$ and $\ell_2$ pruning. Most people think that their pruning properties are almost the same, however:
> > > 	- a. in section3.2, we find that the applicability of $\ell_1$ pruning and $\ell_2$ pruning on layer-wise pruning are different. Compared with $\ell_1$ pruning, $\ell_2$ pruning is more prone to applicability problem.
> > > 	- b. in section 5, we find that the similarity between $\ell_1$ and $\ell_2$ pruning is not always large on global pruning.
> > >
> > >
> > > these phenomena can be explained by CWDA theoretically.
> > >
> > > .
> > >
> > > .
> > >
> > > Thanks for your meaningful comments again.

---

> > > > ### Comment · Reviewer_EQps · 2021-09-01
> > > > **Final comment**
> > > >
> > > > I would like to thank the authors for their time and effort in answering all of my questions. I agree that the paper has a novel approach to evaluating various pruning criteria. The authors address most of my concerns although I am still not fully convinced that the applicability is surprising for norm-based pruning criteria. Therefore, I increase my score to 4.

---

### Official Review · Reviewer_nJ3h · 2021-07-07

**Rating:** 6
**Confidence:** 4

**Summary:**

The authors propose and verify an assumption about the distribution of convolutional weight called CWDA, which reveals that the well-trained convolutional filters in each layer approximately follow a Gaussian-alike distribution. Based on this assumption, the authors analyze the similarity and the applicability problem among different pruning criteria. The point of this analysis is that norm-based metrics, particularly L1 and L2, behave quite similarly. Experimental results on various network structures, tasks, and datasets demonstrate the proposed analysis. The resulting conclusions are interesting. However, there are some issues in the paper. Detailed comments are as follows.

**Limitations And Societal Impact:**

Please discuss the limitations of the proposed method.

**Main Review:**

**Contributions**:
1.	The authors propose and verify an assumption called CWDA, which reveals that the well-trained convolutional filters approximately follow a Gaussian-alike distribution.

2.	Based on the proposed CWDA, the authors analyze the similarity and the applicability problem of different types of pruning criteria, which is an interesting perspective. These two blind spots are able to guide the researchers to design more reasonable criteria.

**Questions and points needed to be improved**:
1.	The title “Rethinking the pruning criteria…” is over-claim. In fact, the main point of the paper is to show that norm-based metrics behave quite similarly. The authors also analyze the similarity and applicability of other pruning criteria. However, the conclusion here is not clear.

2.	In Figure 3 and Table 4, the authors only show the performance of Norm-based pruning criteria. It would be better for the authors to compare the performance between Activation-based [1-2], Importance-based [3-7], and BN-based criteria [8] on layer-wised pruning settings. These results would help the community to better understand different pruning criteria.

3.	There are some repetitions in this paper. (1) References 11 and 26 are repeated. (2) References 12 and 34 are repeated. (3) Sections I and P are repeated in the supplemental material.

4.	Some papers in the references list have been published in a conference/journal. For example, “Wide residual networks” have been published in BMVC 2014. Please cite the conference/journal version instead of the arXiv one.

5.	The authors do not reference the correct Lamma or Equation in Line 740, 762, 780, and 782 of the supplemental material.

**References**

[1]	Network trimming: A data-driven neuron pruning approach towards efficient deep architectures. arXiv 2016.

[2]	An entropy-based pruning method for cnn compression. arXiv 2017.

[3]	Pruning convolutional neural networks for resource efficient inference. ICLR 2017.

[4]	Importance Estimation for Neural Network Pruning. CVPR 2019.

[5]	SNIP: Single-shot Network Pruning based on Connection Sensitivity. ICLR 2019.

[6]	HRank: Filter Pruning using High-Rank Feature Map. CVPR 2020.

[7]	Discrimination-aware Network Pruning for Deep Model Compression. TPAMI 2021.

[8]	Learning Efficient Convolutional Networks through Network Slimming. ICCV 2017.


**Time Spent Reviewing:**

30

---

> ### Author Response · Authors · 2021-08-10
> **Response to Reviewer nJ3h:**
>
> **Q1：The questions about title, citation, reference and typo.**
>
> A1：Thanks for your suggestions, we will update them in revision.
>
> .
>
> .
>
>
> **Q2：In Figure 3 and Table  4, the authors only show the performance of Norm-based pruning  criteria. It would be better for the authors to compare the performance  between Activation-based [1-2], Importance-based [3-7], and BN-based  criteria [8] on layer-wised pruning settings. These results would help  the community to better understand different pruning criteria. The authors also analyze the similarity and applicability of other pruning criteria. However, the conclusion here is not clear.**
>
> A2：We give the pruning results of paper [1-6] according to the setting in Fig 3. The pruning paradigms of paper [7] and paper [8] are different from others, so they are not compared in the following table. In addition, the discussion of slimming [8] can be found in the footnote (page6) or Appendix R (page56).
>
> Experiments on CIFAR10:
>
> | pruned/tuned      | ratio=0.1   | ratio=0.3   | ratio=0.5   | ratio=0.7   | ratio=0.9   |
> |-------------------|-------------|-------------|-------------|-------------|-------------|
> | APoZ[1]           | 91.50/92.40 | 87.56/91.67 | 69.76/91.36 | 30.50/91.40 |  4.32/91.01 |
> | Entropy[2]        | 91.55/92.95 | 85.65/90.76 | 71.65/91.06 | 24.78/91.14 | 10.67/91.61 |
> | Taylor[3]         | 92.43/92.77 | 90.11/91.98 | 75.34/92.01 | 34.67/91.73 |  8.25/92.01 |
> | Taylor$\ell_1$[4] | 92.46/92.64 | 90.43/92.00 | 80.33/91.99 | 38.87/90.96 |  7.98/90.31 |
> | Taylor$\ell_2$[4] | 91.83/92.44 | 90.11/91.43 | 79.89/91.32 | 33.65/91.76 | 10.11/91.03 |
> | SNIP[5]           | 91.65/93.01 | 88.14/92.73 | 75.87/91.54 | 27.88/91.98 |  9.94/91.65 |
> | HRank[6]          | 92.43/93.00 | 88.87/91.90 | 77.11/92.02 | 35.76/92.00 | 10.32/91.87 |
>
> Experiments on CIFAR100:
>
> | pruned/tuned      | ratio=0.1   | ratio=0.3   | ratio=0.5   | ratio=0.7  | ratio=0.9  |
> |-------------------|-------------|-------------|-------------|------------|------------|
> | APoZ[1]           | 66.22/69.94 | 39.75/69.99 | 14.67/68.65 | 3.11/65.86 | 1.07/63.06 |
> | Entropy[2]        | 64.54/70.54 | 43.67/70.43 | 20.54/69.93 | 3.32/67.89 | 1.24/63.46 |
> | Taylor[3]         | 62.71/70.45 | 43.11/70.46 | 17.47/66.03 | 1.37/65.92 | 1.14/64.08 |
> | Taylor$\ell_1$[4] | 60.84/70.98 | 40.72/69.67 | 20.11/68.57 | 2.87/65.84 | 2.06/64.07 |
> | Taylor$\ell_2$[4] | 66.52/70.14 | 42.68/69.23 | 16.56/68.76 | 1.94/65.06 | 1.16/63.09 |
> | SNIP[5]           | 64.02/70.93 | 45.82/68.44 | 13.76/67.98 | 3.32/66.90 | 1.23/65.69 |
> | HRank[6]          | 65.29/70.87 | 44.89/69.63 | 16.76/66.97 | 1.17/66.87 | 1.02/63.75 |
>
> **Conclusions**：
>
>
> (1) According to the results of acc. (pruned/tuned), there is no strong similarity between these pruning criteria in the paper [1-6]. In fact, such results can be expected from the results in Fig. 5 (Page6). There is no strong similarity between these criteria in the global network, but only on some convolutional layers.
>
>
> (2) Although the conclusion here is not clear for other types of criteria (except norm-based), our observations can still provide inspiration for the community. As shown in Fig 5 (Page6), some seemingly unrelated criteria are strong similar (Sp>0.95) in some layers:
> - a. On the one hand, such a strong similarity is NOT an **accident** (refer to Reviewer 6Th5 part1 Q1/A1(1) ) . This phenomenon may bring inspiration to the community;
> - b. on the other hand, the difference in the similarity of various criteria can be used to construct better pruning criteria. For example, [10] uses this property to blend different pruning criteria in different layers to construct better pruning criteria (combining effective criteria with large differences in similarity is the key to ensemble learning ). Based on this observation, another pruning method using ensemble learning [9]  can also be further strengthened.
>
>
> .
>
> .
>
> **Q3：Please discuss the limitations of the proposed method.**
>
> A3：Although we have verified CWDA (line115-117) under various settings in table5 (Appendix q), it still can not completely cover all the training settings of convolutional neural networks. Therefore, if we want to use CWDA under the new setting in the future, we still need to do some verifications. In the future, we can try to prove CWDA theoretically, which may be a better direction.
>
>
> .
>
> .
>
> .
>
>
> [1]    Network trimming: A data-driven neuron pruning approach towards efficient deep architectures. arXiv 2016.
>
> [2]    An entropy-based pruning method for cnn compression. arXiv 2017.
>
> [3]    Pruning convolutional neural networks for resource efficient inference. ICLR 2017.
>
> [4]    Importance Estimation for Neural Network Pruning. CVPR 2019.
>
> [5]    SNIP: Single-shot Network Pruning based on Connection Sensitivity. ICLR 2019.
>
> [6]    HRank: Filter Pruning using High-Rank Feature Map. CVPR 2020.
>
> [7]    Discrimination-aware Network Pruning for Deep Model Compression. TPAMI 2021.
>
> [8]    Learning Efficient Convolutional Networks through Network Slimming. ICCV 2017.
>
> [9]    Learning filter pruning criteria for deep convolutional neural networks acceleration.  CVPR2020.
>
> [10]  Blending Pruning Criteria for Convolutional Neural Networks. ICANN2021.

---

> > ### Comment · Reviewer_nJ3h · 2021-08-26
> > **Response to the Rebuttal**
> >
> > I would like to thank the author for their replies. The authors have addressed my concerns. I tend to keep my initial rating.

---

### Official Review · Reviewer_6Th5 · 2021-07-14

**Rating:** 7
**Confidence:** 5

**Summary:**

Paper studies the problem of selecting a criterion for ranking based structured pruning. Particularly, authors observe that norm based criteria show quite similar behavior. Authors proposed a similarity and an applicability properties of 8 considered criteria. Additionally paper proposes a Convolutional Weight Distribution Assumption. Observations made in the paper will be helpful to the future work on structured pruning.

**Ethics Review Area:**

["I don’t know"]

**Limitations And Societal Impact:**

The main limitation seem to be in the criteria during pruning and do not reflect (or not shown) any outcome of the fine-tuning these models. For example out of 3 metrics we want to see what should be maximized/min to develop new criteria.

**Main Review:**

Strengths:

1) Paper studies an important problem of comparing and understanding various pruning criteria.
2) Selected number of criteria is quite inclusive and contains norm based criteria, gradient based criteria, and activation based.
3) Proposed properties of the pruning criterion is quite interesting and novel. For example Similarity is a good starting point for developing new pruning criterion in the future. Applicability attempts to estimate certainty of different criteria.
4) Comparison between layer-wise and global pruning is on point and provides different observations. Many pruning papers consider only one setting and it makes hard to compare them, but authors used the same framework and provide insights that same criteria behave differently.

Weaknesses:
1) Paper lacks some conclusions about how to interpret Similarity, Applicability and CWDA can help to understand what criteria is the best out of all considered. It might also seem that a criterion with random score assignment might score the highest in similarity and applicability (can authors comment on the Random pruning as it is an important baseline for any pruning method?). Would be interesting to see Fig 5 for that and CWDA.
2) CWDA depends on regularization techniques applied during training. For example weight decay will push weights to be Gaussian, Batch Normalization will de-correlate filters, Drop-out will correlate them more. Could authors comment on how those settings will affect the study and if their settings were varied during the study.
3) Applicability assumes that ideal criterion will clearly distribute filters with ranks (deviation of the scores will be high). However, the hypothesis might be wrong as regularization will significantly affect it. For example if drop-out is used then it is known that the layer will be more robust to pruning (doesn't matter what we remove because it prevents co-adaptation). In this case the ideal criterion will have a low Applicability but will be still relevant. Because of this fact, the Applicability property might be not correct even for the graph in Figure 5.

For rebuttal I am looking in understanding the following:
Answer to points 1-3. Analysis of random pruning will be very interesting.
Can authors finetune some of the model they get from pruning to verify that provided analysis doesn't break after the model is trained (for CWDA at least). Also we will see if similarity is important, as a different choice for only 1 neuron might make a huge difference.
Comment on the limitations (what authors see as a limitation for this work) as well.

Originality - Novel, didn't see the same study in other works
Quality - well written
Clarity - great
Significance - will be interesting to benchmark new applications


**Time Spent Reviewing:**

3

---

> ### Author Response · Authors · 2021-08-10
> **Response to Reviewer 6Th5:(part 1)**
>
> **Q1：Can authors comment on the Random pruning  as it is an important baseline for any pruning method?**
>
> A1：For random pruning，we give the following comments：
>
> (1) **For similarity**:
> - a. If we use random score as the importance score for pruning, we cannot show the experimental results in the form of Fig. 5 because the scores (random) are not unique. We design the following experiments to discuss the similarity problem under random pruning. Take the uniform distribution $U [0,1]$ as the random importance score of the filters. Under the setting in Fig 5, we can get the similarity between the random score and eight different types of pruning criteria.
> - b. notations:
>   + (i) "Sp$ \pm$ std" means that we sample random scores from $U[0,1]$, and calculate the sp & their standard deviation std (in 50000 random repeated experiments) between 8 different types of criteria respectively.
>   + (ii) “>0.8 ratio” and ">0.5 ratio" represent the percentage of the number of Sp greater than 0.8 or 0.5 in 50000 random repeated experiments, respectively.
> - c. conclusions:
>   + (i) From the perspective of Sp, the similarity between the random score criterion and eight different types of criteria is generally very small and the variance is large;
>   + (ii) From the results in the table below, we can find that the criteria of random score can achieve high similarity (i.e. Sp > 0.8), but the probability is extremely low (almost zero). Hense, a criterion with random score assignment is hard to score the highest in similarity. It means that the similarity between the criteria described in our paper is NOT an **accident**, but a phenomenon of great concern.
>
> | Layer-wise              |            | $\ell_1$           | $\ell_2$        | Taylor $\ell_1$    | Taylor $\ell_2$ | BN$_{\gamma}$  | BN$_{\beta}$       | Entropy         | APoZ            |
> |-------------------------|------------|--------------------|-----------------|--------------------|-----------------|----------------|--------------------|-----------------|-----------------|
> | VGG16 5$^{th}$ Conv     | sp$\pm$std | 0.002$\pm$0.12     | 0.000$\pm$0.21  | 0.001$\pm$0.14     | -0.011$\pm$0.22 | 0.000$\pm$0.13 | 0.006$\pm$0.06     | -0.002$\pm$0.11 | -0.013$\pm$0.13 |
> |                         | >0.8 ratio | 6$\times 10^{-3}$% | 0%              | 0%                 | 0%              | 0%             | 0%                 | 0%              | 0%              |
> |                         | >0.5 ratio | 6$\times 10^{-3}$% | 0%              | 8$\times 10^{-3}$% | 0%              | 0%             | 2$\times 10^{-3}$% | 0%              | 0%              |
> | VGG16 10$^{th}$ Conv    | sp$\pm$std | 0.001$\pm$0.21     | 0.000$\pm$0.10  | 0.001$\pm$0.09     | 0.021$\pm$0.12  | 0.001$\pm$0.22 | -0.006$\pm$0.11    | -0.002$\pm$0.14 | 0.006$\pm$0.20  |
> |                         | >0.8 ratio | 0%                 | 0%              | 0%                 | 0%              | 0%             | 0%                 | 0%              | 0%              |
> |                         | >0.5 ratio | 0%                 | 0%              | 0%                 | 0%              | 0%             | 0%                 | 0%              | 0%              |
> | ResNet56 8$^{th}$ Conv  | sp$\pm$std | 0.001$\pm$0.04     | 0.001$\pm$0.11  | -0.001$\pm$0.12    | -0.031$\pm$0.11 | 0.001$\pm$0.12 | 0.003$\pm$0.10     | 0.003$\pm$0.12  | -0.012$\pm$0.17 |
> |                         | >0.8 ratio | 0%                 | 0%              | 0%                 | 0%              | 0%             | 0%                 | 0%              | 0%              |
> |                         | >0.5 ratio | 0%                 | 0%              | 2$\times 10^{-3}$% | 0%              | 0%             | 2$\times 10^{-3}$% | 0%              | 0%              |
> | ResNet56 14$^{th}$ Conv | sp$\pm$std | 0.009$\pm$0.13     | -0.000$\pm$0.11 | -0.001$\pm$0.12    | -0.011$\pm$0.26 | 0.000$\pm$0.11 | -0.001$\pm$0.11    | -0.012$\pm$0.11 | -0.015$\pm$0.15 |
> |                         | >0.8 ratio | 0%                 | 0%              | 0%                 | 0%              | 0%             | 2$\times 10^{-3}$% | 0%              | 0%              |
> |                         | >0.5 ratio | 0%                 | 0%              | 0%                 | 0%              | 0%             | 2$\times 10^{-3}$% | 0%              | 0%              |
>
>
> | Global-wise |            | $\ell_1$                    | $\ell_2$                    | Taylor $\ell_1$             | Taylor $\ell_2$            | BN$_{\gamma}$              | BN$_{\beta}$                | Entropy                     | APoZ                       |
> |-------------|------------|-----------------------------|-----------------------------|-----------------------------|----------------------------|----------------------------|-----------------------------|-----------------------------|----------------------------|
> | VGG16       | sp$\pm$std | -1.0$\times 10^{-5}\pm$0.02 | -2.4$\times 10^{-5}\pm$0.01 | -1.0$\times 10^{-4}\pm$0.04 | 1.3$\times 10^{-5}\pm$0.02 | 1.2$\times 10^{-4}\pm$0.01 | 1.2$\times 10^{-5}\pm$0.01  | -1.7$\times 10^{-6}\pm$0.01 | 1.0$\times 10^{-6}\pm$0.01 |
> |             | >0.8 ratio | 0%                          | 0%                          | 0%                          | 0%                         | 0%                         | 0%                          | 0%                          | 0%                         |
> |             | >0.5 ratio | 0%                          | 0%                          | 0%                          | 0%                         | 0%                         | 0%                          | 0%                          | 0%                         |
> | ResNet56    | sp$\pm$std | 1.1$\times 10^{-4}\pm$0.02  | -1.4$\times 10^{-4}\pm$0.01 | -1.2$\times 10^{-4}\pm$0.04 | 1.0$\times 10^{-5}\pm$0.02 | 1.0$\times 10^{-5}\pm$0.01 | -1.3$\times 10^{-5}\pm$0.01 | 1.3$\times 10^{-5}\pm$0.01  | 1.0$\times 10^{-5}\pm$0.02 |
> |             | >0.8 ratio | 0%                          | 0%                          | 0%                          | 0%                         | 0%                         | 0%                          | 0%                          | 0%                         |
> |             | >0.5 ratio | 0%                          | 0%                          | 0%                          | 0%                         | 0%                         | 0%                          | 0%                          | 0%                         |
>
>
>
> (2) **For applicability**:
> - a. When using random score, its applicability problem depends entirely on the distribution of random score. Hence, the applicability problem of random prune should not be discussed.

---

> > ### Comment · Reviewer_6Th5 · 2021-08-25
> > **response**
> >
> > I would like to thank authors for providing feedback.
> >
> > For Q4 I am satisfied with the answer, it seems CWDA holds under pruning settings.
> >
> > For Q5, I like that the current work can influence future criteria development. However, I wish that would be done in this paper, then besides the analysis we will also have practical outcome. I think this point will strongly improve the paper.
> >
> > For Q3, it seems over parametrizing training will lead to deviation of scores from the ideal scenario. I think this is expected, but also shows the limitation of current work.
> >
> > For random score (Q1) similarity criterion looks to be small with respect to others as expected. However, there is a chance that Applicability will score high for random criteria. Could authors comment on it?
> >
> > Reading other reviews I tend to agree with reviewer nJ3h. Particularly, title should be less confusing. The paper does analysis of different score(criteria) based pruning rather than proposing something different what can be called "Rethinking".  Reviewer EQps points to some significant issues in the paper and, in my opinion, are addressed.
> >
> > Overall my score is between 6 and 7. I like the motivation and direction this paper takes. Newly introduced metrics will help to better motivate future pruning criteria. The description of CWDA is quite confusion in the beginning and requires time to understand it, probably authors should work on this part and make it more intuitive.

---

> > > ### Author Response · Authors · 2021-08-28
> > > **Response to response of Reviewer 6Th5**
> > >
> > > Thanks for your meaningful comments again.
> > >
> > > **Response Q1: For Q5, I like that the current work can influence future criteria development. However, I wish that would be done in this paper, then besides the analysis we will also have practical outcome. I think this point will strongly improve the paper.**
> > >
> > > **Response A1:** Thanks for your suggestions. Besides the analysis, in EQps (part 2) Q4/A4(3), from the perspective of similarity and applicability, we can provide the requirements of the ideal criterion, which can affect the development of the criterion in the future.
> > > - (a).The estimation of the redundancy of convolutional filters should be relatively accurate;
> > > - (b).On the basis that the estimation of redundancy is correct, the  ideal criterion should be distinguishable;
> > > - (c). The similarity among the accurate criteria (satisfy a. and b.) should be appropriate:
> > >   - (i) Their similarity should not be too small (which means that there  may be a contradiction like appendix J. If the similarity is very low,  it indicates that at least one pruning criterion needs to be reviewed  again),
> > >   - (ii) Their similarity cannot be too large (Sp cannot be too large,  which means that these criteria are the same, as shown in Fig 4, which  is generally greater than 0.9)
> > >
> > > .
> > >
> > > .
> > >
> > > .
> > >
> > >
> > >
> > > **Response Q2: For random score (Q1) similarity criterion looks to be small with  respect to others as expected. However, there is a chance that  Applicability will score high for random criteria. Could authors comment  on it?**
> > >
> > > **Response A2:** Yes, we can comment “there is a chance that  Applicability will score high for random criteria” as follow:
> > > - (1) We can indeed easily make Applicability score high or low for random criteria.
> > > - (2) However, as mentioned in "6th5: (Part 1) Q1/A1 (2) for applicability", we believe that the applicability problem of random prune should not be discussed. Because under this setting, the applicability problem completely depends on the distribution of the random score we selected.
> > > - (3) For example, Let's consider the scheme of the random score with normal distribution $N(x, y)$: (a) scheme 1: $x$ = 1 and $y$ = 0.000001; (b) scheme 2: $x$ = 1 and $y$ =10000; According to Eq (3), we can know that the applicability of scheme 1 is more serious than that of scheme 2. However, this result can not give any conclusion, because the selection of distribution is artificial and has nothing to do with the redundancy of the network. In other words, this random score cannot represent the importance of convolutional filters (as mentioned in EQps (part 1) Q3/A3, accurate importance evaluation is very important).
> > > - (4) According to Response A1 b and Overall Comments Q4/A4 for Reviewer EQps. "Importance scores can be distinguished" is a necessary condition for an ideal pruning criterion, but it does not mean that a pruning criterion with "importance scores can be distinguished" is an ideal one. Although we can easily construct special random scores so that these scores have no applicability problems  (like Response A2(3)), it does not mean that such a criterion is ideal.
> > >
> > > In conclusion, we believe that even for “there is a chance that  Applicability will score high for random criteria” . There is no need to worry about this phenomenon, and it has no impact on the conclusions of our paper.
> > >
> > >
> > > .
> > >
> > > .
> > >
> > > .
> > >
> > >
> > > **Response Q3:  Reading other reviews I tend to agree with reviewer nJ3h.  Particularly, title should be less confusing.**
> > >
> > >
> > > Response A3:  Thanks for the suggestions from reviewer nJ3h and you about the title. We will fix it in the revised version.
> > >
> > > .
> > >
> > > .
> > >
> > > .
> > >
> > >
> > > **Response Q4: Newly introduced metrics will help to better  motivate future pruning criteria**
> > >
> > >
> > > Response A4: Thanks for your suggestion and we list the requirements about "help to better motivate furure pruning criteria" in Response A1 or EQps (part 2) Q4/A4.
> > >
> > > .
> > >
> > > .
> > >
> > > .
> > >
> > >
> > >
> > > **Response Q5: The description of CWDA is quite confusing in the beginning and requires time to understand it, probably  authors should work on this part and make it more intuitive.**
> > >
> > > Response A5: Thanks for your suggestion. And we will improve it in the revised version.

---

> ### Author Response · Authors · 2021-08-10
> **Response to Reviewer 6Th5:(part 2)**
>
> **Q2：Could authors comment on how those settings ( dropout/bn/weight decay ) will affect the study and if their settings were varied during the study.**
>
> A2：For these settings (dropout / BN / weight decay)，we give the following comments：
>
> (1) For these settings (BN / weight decay), we have given them in the appendix, and they are: (i) weight decay: page52-table8 and Appendix M; (ii) BN: page64-figure 33；
>
> (2) About dropout: Due to the limited space, we take the setting of appendix Q table6 (the statistical test on different Network structures) as an example to study how dropout affects the CWDA. The prune ratio is set as 20% and 30%. The results are shown below. Moreover, the analysis about another similar reg method, called Dropact, can be found in page52-table8.
>
> **drop ratio = 0.2**:
>
> | Model(dropout)     | Remark   | Gaussion | Variance | Mean   | Magnitude | in the front of network? |
> |--------------------|----------|----------|----------|--------|-----------|--------------------------|
> | ResNet164          | CIFAR100 | 98.77%   | 97.55%   | 100%   | 100%      | $\checkmark$             |
> | VGG16              | CIFAR100 | 100%     | 93.75%   | 93.75% | 93.75%    | $\checkmark$             |
> | AlexNet            | CIFAR100 | 100%     | 100%     | 100%   | 100%      | $\checkmark$             |
> | DenseNet-BC-100-12 | CIFAR100 | 100%     | 98.99%   | 98.99% | 98.99%    | $\checkmark$             |
> | PreResNet110       | CIFAR100 | 99.08%   | 99.08%   | 100%   | 100%      | $\checkmark$             |
> | WRN28-10           | CIFAR100 | 100%     | 100%     | 100%   | 100%      | $\checkmark$             |
> | ResNext-16x64d     | CIFAR100 | 100%     | 100%     | 100%   | 100%      | $\checkmark$             |
> | ResNet164          | CIFAR10  | 100%     | 100%     | 100%   | 100%      | $\checkmark$             |
> | VGG16              | CIFAR10  | 100%     | 93.75%   | 100%   | 93.75%    | $\checkmark$             |
> | AlexNet            | CIFAR10  | 100%     | 100%     | 100%   | 100%      | $\checkmark$             |
> | DenseNet-BC-100-12 | CIFAR10  | 100%     | 98.99%   | 98.99% | 100%      | $\checkmark$             |
> | PreResNet110       | CIFAR10  | 99.08%   | 99.08%   | 100%   | 100%      | $\checkmark$             |
> | WRN28-10           | CIFAR10  | 100%     | 100%     | 100%   | 100%      | $\checkmark$             |
> | ResNext-16x64d     | CIFAR10  | 100%     | 100%     | 100%   | 100%      | $\checkmark$             |
>
> **drop ratio = 0.3**:
>
> | Model(dropout)     | Remark   | Gaussion | Variance | Mean   | Magnitude | in the front of network? |
> |--------------------|----------|----------|----------|--------|-----------|--------------------------|
> | ResNet164          | CIFAR100 | 97.55%   | 97.55%   | 100%   | 97.55%    | $\checkmark$             |
> | VGG16              | CIFAR100 | 93.75%   | 93.75%   | 93.75% | 93.75%    | $\checkmark$             |
> | AlexNet            | CIFAR100 | 100%     | 100%     | 100%   | 100%      | $\checkmark$             |
> | DenseNet-BC-100-12 | CIFAR100 | 100%     | 98.99%   | 100%   | 98.99%    | $\checkmark$             |
> | PreResNet110       | CIFAR100 | 99.08%   | 99.08%   | 100%   | 100%      | $\checkmark$             |
> | WRN28-10           | CIFAR100 | 100%     | 100%     | 100%   | 100%      | $\checkmark$             |
> | ResNext-16x64d     | CIFAR100 | 100%     | 100%     | 100%   | 100%      | $\checkmark$             |
> | ResNet164          | CIFAR10  | 100%     | 100%     | 100%   | 100%      | $\checkmark$             |
> | VGG16              | CIFAR10  | 93.75%   | 93.75%   | 93.75% | 93.75%    | $\checkmark$             |
> | AlexNet            | CIFAR10  | 100%     | 100%     | 100%   | 100%      | $\checkmark$             |
> | DenseNet-BC-100-12 | CIFAR10  | 100%     | 98.99%   | 98.99% | 100%      | $\checkmark$             |
> | PreResNet110       | CIFAR10  | 99.08%   | 99.08%   | 100%   | 99.08%    | $\checkmark$             |
> | WRN28-10           | CIFAR10  | 96.43%   | 96.43%   | 100%   | 96.43%    | $\checkmark$             |
> | ResNext-16x64d     | CIFAR10  | 100%     | 100%     | 100%   | 100%      | $\checkmark$             |
>
> (3) According to the statistical test (Appendix Q) and Q2/A2(1), CWDA holds under different training method (including the settings about dropout / BN / weight decay). The distribution of convolutional filters is Gaussian-alike distribution, and their difference are the matrix $\Sigma_{\mathrm{diag}}+\epsilon \cdot \Sigma_{\text {block }}$ in Eq(1).
>
> (4) To specifically study the effect of them on CWDA, the more detailed experiments about different weight decay (wd) have been shown in Appendix M. The difference wd does cause different normality, but it’s not the key reason for the normality.
>
>
>
> .
>
>
> .
>
> .
>
> .
>
> .
>
>
>
>
> **Q3:Applicability assumes that ideal criterion will clearly distribute filters with ranks (deviation of the scores will be high). However, the hypothesis might be wrong as regularization will significantly affect it.**
>
> A3:
> (1) The experiments shown in the following table are to explore the impact of dropout on Applicability ( take $\ell_2$ criterion as example) using Eq (3). From the results in the table below, it seems that dropout does not have a great impact on Applicability.
>
> |                    | dropout p=0.1       | dropout p=0.3       | dropout p=0.5       |
> |--------------------|---------------------|---------------------|---------------------|
> | VGG16 5$^{th}$     | 4.5$\times 10^{-3}$ | 4.2$\times 10^{-3}$ | 4.6$\times 10^{-3}$ |
> | VGG16 10$^{th}$    | 7.9$\times 10^{-3}$ | 7.7$\times 10^{-3}$ | 7.8$\times 10^{-3}$ |
> | ResNet56 8$^{th}$  | 6.5$\times 10^{-2}$ | 6.1$\times 10^{-2}$ | 6.2$\times 10^{-2}$ |
> | ResNet56 14$^{th}$ | 3.7$\times 10^{-3}$ | 4.0$\times 10^{-3}$ | 3.5$\times 10^{-3}$ |
>
>
> (2) In the following table, we give the unpruned Acc. and Applicability (Appli.) under different weight decay and different neural networks.
> Regularization does affect the Applicability, such as Fig.4 in [1] and the results shown in the table below. However, it should be noted that, as shown in the Section5.2 from [3], strong reg can significantly affect the applicability, but it also affects the training of the model (for example, the performance is changed from 77.57miou to 63.63miou, when changing the reg penalty coefficient from 1e-4 to 1e-3), which is consistent with the results in the following table.
> Therefore, if we want to discuss the impact of reg, we must discuss it within the appropriate reg range, that is, the model should be well-trained (line84-85, line322-326).  As shown in the following table, when the model is well-trained (like wd=1e-4~1e-3), the change in applicability is not obvious.
>
> | CIFAR100           |               | wd=1e-5             | wd=5e-5             | wd=1e-4             | wd=5e-4             | wd=1e-3             | wd=5e-3             | wd=1e-2             |
> |--------------------|---------------|---------------------|---------------------|---------------------|---------------------|---------------------|---------------------|---------------------|
> | VGG16 5$^{th}$     | Unpruned Acc. | 69.02               | 69.89               | 71.64               | 72.82               | 72.56               | 70.34               | 63.54               |
> |                    | Appli.        | 6.1$\times 10^{-3}$ | 5.5$\times 10^{-3}$ | 4.5$\times 10^{-3}$ | 4.5$\times 10^{-3}$ | 4.0$\times 10^{-3}$ | 2.8$\times 10^{-3}$ | 2.1$\times 10^{-3}$ |
> | VGG16 10$^{th}$    | Unpruned Acc. | 69.02               | 69.89               | 71.64               | 72.82               | 72.56               | 70.34               | 63.54               |
> |                    | Appli.        | 8.5$\times 10^{-3}$ | 8.1$\times 10^{-3}$ | 7.9$\times 10^{-3}$ | 7.5$\times 10^{-3}$ | 7.0$\times 10^{-3}$ | 5.6$\times 10^{-3}$ | 5.7$\times 10^{-3}$ |
> | ResNet56 8$^{th}$  | Unpruned Acc. | 66.87               | 67.88               | 71.11               | 70.84               | 69.36               | 66.84               | 64.84               |
> |                    | Appli.        | 7.1$\times 10^{-2}$ | 6.9$\times 10^{-2}$ | 6.3$\times 10^{-2}$ | 6.2$\times 10^{-2}$ | 5.6$\times 10^{-2}$ | 6.3$\times 10^{-2}$ | 6.0$\times 10^{-2}$ |
> | ResNet56 14$^{th}$ | Unpruned Acc. | 66.87               | 67.88               | 71.11               | 70.84               | 69.36               | 66.84               | 64.84               |
> |                    | Appli.        | 4.9$\times 10^{-3}$ | 4.3$\times 10^{-3}$ | 4.2$\times 10^{-3}$ | 3.9$\times 10^{-3}$ | 3.6$\times 10^{-3}$ | 2.9$\times 10^{-3}$ | 3.1$\times 10^{-3}$ |
>
> .
>
> .
>
> [1] Learning Efficient Convolutional Networks through Network Slimming. ICCV 2017.
>
> [2] Learning filter pruning criteria for deep convolutional neural networks acceleration.  CVPR2020.
>
> [3] CAP: Context-Aware Pruning for Semantic Segmentation. WACV2021.
>
> [4] Blending Pruning Criteria for Convolutional Neural Networks. ICANN2021.

---

> ### Author Response · Authors · 2021-08-10
> **Response to Reviewer 6Th5:(part 3)**
>
> **Q4：Can authors finetune  some of the model they get from pruning to verify that provided analysis  doesn't break after the model is trained (for CWDA at least)**
>
> A4：Due to the limited space, we take the setting of appendix Q table6 (the statistical test on different Network structures) as an example to study whether the pruned model after finetuning still meets CWDA. The prune ratio is set as 10% and 20%.
>
> **pruning ratio = 10%**:
>
>
> | Model(pruned-tuned) | Remark   | Gaussion | Variance | Mean   | Magnitude | in the front of network? |
> |---------------------|----------|----------|----------|--------|-----------|--------------------------|
> | ResNet164           | CIFAR100 | 97.55%   | 97.55%   | 100%   | 100%      | $\checkmark$             |
> | VGG16               | CIFAR100 | 100%     | 84.62%   | 93.75% | 93.75%    | $\checkmark$             |
> | AlexNet             | CIFAR100 | 100%     | 100%     | 100%   | 100%      | $\checkmark$             |
> | PreResNet110        | CIFAR100 | 99.08%   | 99.08%   | 100%   | 100%      | $\checkmark$             |
> | WRN28-10            | CIFAR100 | 100%     | 96.43%   | 100%   | 96.43%    | $\checkmark$             |
> | ResNet164           | CIFAR10  | 100%     | 100%     | 100%   | 100%      | $\checkmark$             |
> | VGG16               | CIFAR10  | 100%     | 93.75%   | 100%   | 93.75%    | $\checkmark$             |
> | AlexNet             | CIFAR10  | 100%     | 100%     | 100%   | 100%      | $\checkmark$             |
> | PreResNet110        | CIFAR10  | 99.08%   | 99.08%   | 100%   | 100%      | $\checkmark$             |
> | WRN28-10            | CIFAR10  | 100%     | 96.43%   | 100%   | 96.43%    | $\checkmark$             |
>
>
>
> **pruning ratio = 20%**:
>
> | Model(pruned-tuned) | Remark   | Gaussion | Variance | Mean   | Magnitude | in the front of network? |
> |---------------------|----------|----------|----------|--------|-----------|--------------------------|
> | ResNet164           | CIFAR100 | 96.32%   | 96.32%   | 100%   | 100%      | $\checkmark$             |
> | VGG16               | CIFAR100 | 93.75%   | 84.62%   | 93.75% | 84.62%    | $\checkmark$             |
> | AlexNet             | CIFAR100 | 100%     | 100%     | 100%   | 100%      | $\checkmark$             |
> | PreResNet110        | CIFAR100 | 99.08%   | 99.08%   | 100%   | 100%      | $\checkmark$             |
> | WRN28-10            | CIFAR100 | 100%     | 96.43%   | 96.43% | 96.43%    | $\checkmark$             |
> | ResNet164           | CIFAR10  | 100%     | 100%     | 100%   | 100%      | $\checkmark$             |
> | VGG16               | CIFAR10  | 84.62%   | 93.75%   | 84.62% | 93.75%    | $\checkmark$             |
> | AlexNet             | CIFAR10  | 100%     | 100%     | 100%   | 100%      | $\checkmark$             |
> | PreResNet110        | CIFAR10  | 99.08%   | 99.08%   | 100%   | 100%      | $\checkmark$             |
> | WRN28-10            | CIFAR10  | 96.43%   | 96.43%   | 100%   | 96.43%    | $\checkmark$             |
>
>
> It can be seen from the results in the above table that CWDA still holds. Moreover, It should be noted that,  with the reduction of parameters (after pruning),  the estimation of statistics may be affected because the accuracy of the estimation of statistics is related to the number of samples (line101-106).
>
> .
>
> .
>
> .
>
>
> **Q5：The main limitation  seem to be in the criteria during pruning and do not reflect (or not  shown) any outcome of the fine-tuning these models. For example out of 3  metrics we want to see what should be maximized/min to develop new criteria.**
>
> A5：If we use ensemble learning [2] [4] to develop new criteria (line346-347),  our paper can inspire that the differences in candidate criteria should be maximized for powerful criteria [4]. This may be an interesting direction in the future.
>
>
> [1] Learning Efficient Convolutional Networks through Network Slimming. ICCV 2017.
>
> [2] Learning filter pruning criteria for deep convolutional neural networks acceleration.  CVPR2020.
>
> [3] CAP: Context-Aware Pruning for Semantic Segmentation. WACV2021.
>
> [4] Blending Pruning Criteria for Convolutional Neural Networks. ICANN2021.

---

### Official Review · Reviewer_Y2y2 · 2021-07-17

**Rating:** 4
**Confidence:** 4

**Summary:**

The paper attempts to study and compare different pruning "criteria".
The paper addresses an interesting question, trying to evaluate if there is a need for a more complex pruning criteria. However the paper can be better organized. And some of the concerts are hard to follow.

**Main Review:**

Originality: The work is original and provides an attempt to evaluate the validity of new and more complex pruning criteria
Quality: The paper attempts to prove empirical study along with their theoretical/mathematical proof which adds more significance to work. However, the paper organization needs additional work. Especially the introduction part, where the main concepts and problems are introduced.
Clarity: The paper is hard to follow. The main concepts are purely introduced and paper is hard to follow. "Convolution Weight Distribution Assumption" were not well introduced and hard to follow. It would be helpful if more details be provided on the existing pruning criteria. Provided grouping is too generic and not well organized.
Significance: The work attempts to provide a tool for more unified evaluation of the pruning methods, which was missing in the current methodology.

**Time Spent Reviewing:**

3 hours

---

> ### Author Response · Authors · 2021-08-10
> **Response to Reviewer Y2y2:**
>
> **Q1 : This paper is hard to follow.**
>
> A1: In fact, our paper is not hard to follow and there are several facts as follows:
>
> - (1) We find two blind spots, called Similarity and Applicability problem, in network pruning (page1-2 Introduction);
> - (2) In order to study these spots, we propose CWDA (Section 2)
> - (3) We analyze these spots through empirical experiments and theoretical analysis by CWDA (Section 3-5)
> - (4) we discuss our findings of these spots (Section 6)
>
> **Q2 : It would be helpful if more details be provided on the existing pruning criteria.**
>
> A2: We had provided the details on the existing pruning criteria in Table2 (line 42) and Appendix K (line213).

---

### Decision · Program_Chairs · 2021-09-27

**Decision:**

Accept (Poster)

**Comment:**

The paper provides a study of pruning criteria by evaluating different methods and defining metrics that help understand the different properties of the various criteria out there. The topic of DNN pruning is quite rich today and I personally find it refreshing (as do some of the reviewers) to see a paper that tries to sort out the different methods. The paper is far from a complete solution to the problem, but it provides a good start. Given the importance and magnitude of the problem, I believe the paper is worthy of being published at NeurIPS and will likely motivate follow up papers that will improve the way we currently measure pruning criteria.